# AUTO-ENCODING GOODNESS OF FIT

**Aaron Palmer[1], Zhiyi Chi[2], Derek Aguiar[1], Jinbo Bi[1]**
[1]Department of Computer Science, [2]Department of Statistics
`{aaron.palmer,zhiyi.chi,derek.aguiar,jinbo.bi}@uconn.edu`
University of Connecticut, Storrs, CT, USA

## ABSTRACT

We develop a new type of generative autoencoder called the Goodness-of-Fit Autoencoder (GoFAE), which incorporates GoF tests at two levels. At the minibatch level, it uses GoF test statistics as regularization objectives. At a more global level, it selects a regularization coefficient based on higher criticism, i.e., a test on the uniformity of the local GoF p-values. We justify the use of GoF tests by providing a relaxed $L_2$-Wasserstein bound on the distance between the latent distribution and a distribution class. We prove that optimization based on these tests can be done with stochastic gradient descent on a compact Riemannian manifold. Empirically, we show that our higher criticism parameter selection procedure balances reconstruction and generation using mutual information and uniformity of p-values respectively. Finally, we show that GoFAE achieves comparable FID scores and mean squared errors with competing deep generative models while retaining statistical indistinguishability from Gaussian in the latent space based on a variety of hypothesis tests.

## 1 INTRODUCTION

Generative autoencoders (GAEs) aim to achieve unsupervised, implicit generative modeling via learning a latent representation of the data (Bousquet et al., 2017). A generative model, known as the decoder, maps elements in a latent space, called codes, to the data space. These codes are sampled from a pre-specified distribution, or *prior*. GAEs also learn an encoder that maps the data space into the latent space, by controlling the probability distribution of the transformed data, or *posterior*. As an important type of GAEs, variational autoencoders (VAEs) maximize a lower bound on the data log-likelihood, which consists of a reconstruction term and a Kullback-Leibler (KL) divergence between the approximate posterior and prior distributions (Kingma & Welling, 2013; Rezende et al., 2014). Another class of GAEs seek to minimize the optimal transport cost (Villani, 2008) between the true data distribution and the generative model. This objective can be simplified into an objective minimizing a reconstruction error and subject to matching the aggregated posterior to the prior distribution (Bousquet et al., 2017). This constraint is relaxed in the Wasserstein autoencoder (WAE) (Tolstikhin et al., 2017) via a penalty on the divergence between the aggregated posterior and the prior, allowing for a variety of discrepancies (Patrini et al., 2018; Kolouri et al., 2018).

Regardless of the criterion, training a GAE requires balancing low reconstruction error with a regularization loss that encourages the latent representation to be meaningful for data generation (Hinton & Salakhutdinov, 2006; Ruthotto & Haber, 2021). Overly emphasizing minimization of the divergence metric between the data derived posterior and the prior in GAEs is problematic. In VAEs, this can manifest as posterior collapse (Higgins et al., 2017; Alemi et al., 2018; Takida et al., 2021) resulting in the latent space containing little information about the data. Meanwhile, WAE can suffer from over-regularization when the prior distribution is too simple, e.g. isotropic Gaussian (Rubenstein et al., 2018; Dai & Wipf, 2019). Generally, it is difficult to decide when the posterior is *close enough* to the prior but not to a degree that is problematic. The difficulty is rooted in several issues: (a) an absence of tight constraints on the statistical distances; (b) distributions across minibatches used in the training; and (c) difference in scale between reconstruction and regularization objectives.

Unlike statistical distances, goodness-of-fit (GoF) tests are statistical hypothesis tests that assess the indistinguishability between a given (empirical) distribution and a *distribution class* (Stephens, 2017). In recent years, GAEs based on GoF tests have been proposed to address some of the aforementioned

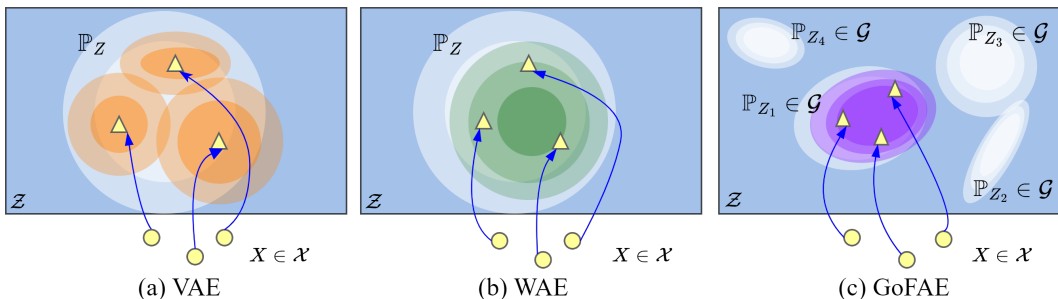

Figure 1: Latent behaviors for VAE, WAE, and GoFAE inspired from Figure 1 of Tolstikhin et al. (2017). (a) The VAE requires the approximate posterior distribution (orange contours) to match the prior $\mathbb{P}_Z$ (white contours) for each example. (b) The WAE forces the encoded distribution (green contours) to match prior $\mathbb{P}_Z$. (c) The GoFAE forces the encoded distribution (purple contours) to match some $\mathbb{P}_Z$ in the *class* prior $\mathcal{G}$. For illustration, several $\mathbb{P}_{Z_i} \in \mathcal{G}$ are visualized (white contours).

issues of VAEs and WAEs (Ridgeway & Mozer, 2018; Palmer et al., 2018; Ding et al., 2019). However, this emerging GAE approach still has some outstanding issues. In GAEs, GoF test statistics are optimized *locally* in minibatches. The issue of balancing reconstruction error and meaningful latent representation is manifested as the calibration of GoF test p-values. If GoF test p-values are too small (i.e., minibatches are distinguishable from the prior), then sampling quality is poor; conversely, an abundance of large GoF p-values may result in poor reconstruction as the posterior matches too closely to the prior at the minibatch level. In addition, there currently does not exist a stochastic gradient descent (SGD) algorithm that is applicable to GoF tests due to identifiability issues, unbounded domains, and gradient singularities.

**Our Contributions:** We study the GoFAE a framework for parametric test statistic optimization, resulting in a novel GAE that optimizes GoF tests for normality, and an algorithm for regularization coefficient selection. Note that the GoF tests are not only for Gaussians with nonsingular covariance matrices, but *all* Gaussians, so they can handle situations where the data distribution is concentrated on or closely around a manifold with dimension smaller than the ambient space. The framework uses Gaussian priors as it is a standard option (Doersch, 2016; Bousquet et al., 2017; Kingma et al., 2019), and also because normality tests are better understood than GoF tests for other distributions, with more tests and more accurate calculation of p-values available. The framework can be modified to use other priors in a straightforward way provided that the same level of understanding can be gained on GoF tests for those priors as for normality tests. See Fig.(1) for latent space behavior of VAE, WAE and our GoFAE. Proofs are deferred to the appendix. Our contributions are summarized as follows.

- We propose a framework (Sec. 2) for bounding the statistical distance between the posterior and a prior distribution *class* $\mathcal{G}$ in GAEs, which forms the theoretical foundation for a deterministic GAE - Goodness of Fit Autoencoder (GoFAE), that directly optimizes GoF hypothesis test statistics.
- We examine four GoF tests of normality based on correlation and empirical distribution functions (Sec. 3). Each GoF test focuses on a different aspect of Gaussianity, e.g., moments or quantiles.
- A model selection method using higher criticism of p-values is proposed (Sec. 3), which enables global normality testing and is test-based instead of performance-based. This method helps determine the range of the regularization coefficient that well balances reconstruction and generation using uniformity of p-values respectively (Fig. 3b).
- We show that gradient based optimization of test statistics for normality can be complicated by identifiability issues, unbounded domains, and gradient singularities; we propose a SGD that optimizes over a Riemannian manifold (Stiefel manifold in our case) that effectively solves our GAE formulation with convergence analysis (Sec. 4).
- We show that GoFAE achieves comparable FID scores and mean squared error on three datasets while retaining statistical indistinguishability from Gaussian in the latent space (Sec. 5).

## 2   PRELIMINARIES FOR GOODNESS OF FIT AUTOENCODING

**Background.** Let $(\mathcal{X}, \mathbb{P}_X)$ and $(\mathcal{Z}, \mathbb{P}_Z)$ be two probability spaces. In our setup, $\mathbb{P}_X$ is the true, but unknown, non-atomic distribution on the data space $\mathcal{X}$, while $\mathbb{P}_Z$ is a prior distribution on the latent space $\mathcal{Z}$. An implicit generative model is defined by sampling a code $Z \sim \mathbb{P}_Z$ and

applying a mapping $G$, called the decoder, to produce $G(Z) \in \mathcal{X}$. The distribution of $G(Z)$ is given by the pushforward of $\mathbb{P}_Z$ under $G$, commonly denoted by $G_\# \mathbb{P}_Z$. Our concern is finding $G$ such that $G_\# \mathbb{P}_Z$ is close to $\mathbb{P}_X$. One approach is based on optimal transport. If $\mu$ and $\nu$ are two distributions respectively on spaces $\mathcal{E}_1$ and $\mathcal{E}_2$, and $c(u, v)$ is a cost function on $\mathcal{E}_1 \times \mathcal{E}_2$, then the optimal transport cost to transfer $\mu$ to $\nu$ is $\mathcal{T}_c(\mu, \nu) = \inf_{\pi \in \Pi(\mu, \nu)} \mathbb{E}_{(U,V) \sim \pi}[c(U, V)]$, where $\Pi(\mu, \nu)$ is the set of all joint probability distributions with marginals $\mu$ and $\nu$. If $\mathcal{E}_1 = \mathcal{E}_2$ and is endowed with a metric $d$, and $c(u, v) = d(u, v)^p$ with $p > 0$, then $d_{W_p}(\mu, \nu) = \mathcal{T}_c(\mu, \nu)^{1/p}$ is known as the $L_p$-Wasserstein distance. In principle, $G$ can be learned by solving $\min_G \mathcal{T}_c(\mathbb{P}_X, G_\# \mathbb{P}_Z)$. Unfortunately, the minimization is intractable. Instead, the WAE approach entails finding a mapping $F$, known as the encoder, such that the pushforward distribution $\mathbb{P}_Y := F_\# \mathbb{P}_X$ matches $\mathbb{P}_Z$. We will only consider *non-random* encoders and decoders, that is, a fully deterministic autoencoder. This is theoretically supported by the Monge-Kantorovich equivalence (Villani, 2008; Patrini et al., 2018). In this setting, $F : \mathcal{X} \to \mathcal{Z}$ encodes $X \sim \mathbb{P}_X$ to produce the code $Y = F(X)$, and $G : \mathcal{Z} \to \mathcal{X}$ decodes $Y$ to produce the reconstruction $G(Y)$. To emphasize the encode-decode process we can write the reconstruction as $(G \circ F)(X) := G(F(X)) = G(Y)$ and the reconstruction distribution as the pushforward distribution $(G \circ F)_\# \mathbb{P}_X = G_\# F_\# \mathbb{P}_X = G_\# \mathbb{P}_Y$. We can define the WAE objective

$$\mathrm{WAE}_c^\lambda(\mathbb{P}_X, G) = \inf_F \left[ \int_{\mathcal{X}} c(x, G(F(x))) \mathrm{d} \mathbb{P}_X(x) + \lambda D(F_\# \mathbb{P}_X, \mathbb{P}_Z) \right], \tag{1}$$

where $\lambda > 0$ is a regularization coefficient and $D$ is a penalty on statistical discrepancy.

**Wasserstein Distance: Bounds.** When $\mathcal{X}$ and $\mathcal{Z}$ are Euclidean spaces, a common choice for $c$ is the squared Euclidean distance, leading to the $L_2$-Wasserstein distance $d_{W_2}$ and the following bounds.

**Proposition 1.** *If $\mathcal{X}$ and $\mathcal{Z}$ are Euclidean spaces and the decoder $G$ is differentiable, then (a) $d_{W_2}(G_\# \mathbb{P}_Y, G_\# \mathbb{P}_Z) \leq \|\nabla G\|_\infty d_{W_2}(\mathbb{P}_Y, \mathbb{P}_Z)$, (b) $d_{W_2}(\mathbb{P}_X, G_\# \mathbb{P}_Z) \leq [\mathbb{E} \|X - G(Y)\|^2]^{1/2} + \|\nabla G\|_\infty d_{W_2}(\mathbb{P}_Y, \mathbb{P}_Z)$.*

Combined with the triangle inequality for $d_{W_2}$, (a) and (b) in Proposition 1 imply that when the reconstruction error $[\mathbb{E} \|X - G(Y)\|^2]^{1/2}$ is small, proximity between the latent distribution $\mathbb{P}_Y$ and the prior $\mathbb{P}_Z$ is sufficient to ensure proximity of the generated distribution $G_\# \mathbb{P}_Z$ and the data distribution $\mathbb{P}_X$. Both (a) and (b) are similar to Patrini et al. (2018), though here a less tight bound – the square-error – is used in (b) as it is an easier objective to optimize. Given a sample of data, $\{X_i\}_{i=1}^n$, the $L_2$-Wasserstein distance between the empirical distribution of $\{Y_i = F(X_i)\}_{i=1}^n$ and $\mathbb{P}_Z$ is a natural approximation of $d_{W_2}(\mathbb{P}_Y, \mathbb{P}_Z)$ because it is a consistent estimator of the latter as $n \to \infty$, although in general is biased (Xu et al., 2020).

**Proposition 2.** *Let $\hat{\mathbb{P}}_{X,n}$ be the empirical distribution of samples $\{X_i\}_{i=1}^n$, and $\hat{\mathbb{P}}_{Y,n}$ be the empirical distribution of $\{Y_i = F(X_i)\}_{i=1}^n$. Assume that $F(X)$ is differentiable with respect to $X$ with bounded gradients $\nabla F(X)$. Then, (a) $d_{W_2}(\hat{\mathbb{P}}_{Y,n}, \mathbb{P}_Y) \leq \|\nabla F\|_\infty d_{W_2}(\hat{\mathbb{P}}_{X,n}, \mathbb{P}_X)$, (b) $d_{W_2}(\mathbb{P}_Y, \mathbb{P}_Z) \leq d_{W_2}(\hat{\mathbb{P}}_{Y,n}, \mathbb{P}_Z) + \|\nabla F\|_\infty d_{W_2}(\hat{\mathbb{P}}_{X,n}, \mathbb{P}_X)$.*

For large $n$, $d_{W_2}(\hat{\mathbb{P}}_{X,n}, \mathbb{P}_X)$ is small, so from (b), the proximity of the latent distribution and the prior is mainly controlled by the proximity between the empirical distribution $\hat{\mathbb{P}}_{Y,n}$ and $\mathbb{P}_Z$. Together with Proposition 1, this shows that in order to achieve close proximity between $G_\# \mathbb{P}_Z$ and $\mathbb{P}_X$, we need to have a strong control on the proximity between $\hat{\mathbb{P}}_{Y,n}$ and $\mathbb{P}_Z$ in addition to a good reconstruction.

**Extending to a Class.** So far we have focused on a *single* completely specified $\mathbb{P}_Z$. However, as we will argue in the next section, it can be beneficial to reduce the specificity to allow for a *class* $\mathcal{G}$ of priors. Letting $d_{W_2}(\cdot, \mathcal{G}) = \inf_{\mathbb{P} \in \mathcal{G}} d_{W_2}(\cdot, \mathbb{P})$, all the above results can be easily extended. For example, Proposition 1 (a) gives $\inf_{\mathbb{P}_Z \in \mathcal{G}} d_{W_2}(G_\# \mathbb{P}_Y, G_\# \mathbb{P}_Z) \leq \|G\|_\infty d_{W_2}(\mathbb{P}_Y, \mathcal{G})$. Once $d_{W_2}(\mathbb{P}_Y, \mathcal{G})$ is controlled, we can then identify $\mathbb{P}_Z \in \mathcal{G}$ that satisfies $d_{W_2}(\mathbb{P}_Y, \mathbb{P}_Z) = d_{W_2}(\mathbb{P}_Y, \mathcal{G})$ and use it as the prior. Clearly, if $\mathcal{G}$ only contains one distribution, this is reduced to the case where the prior is completely specified. Thus, the setting that allows for a class of priors is more general.

We propose to use hypothesis tests associated with the Wasserstein distance. This does not preclude tests associated with other statistical distances, which may even be used in conjunction. Many standard statistical distances are not proper metrics, but aim to control statistical distances dominated by the Wasserstein distance. At the *population* level, the Wasserstein distance provides stronger separation between different distributions; at the *sample* level, it is useful to use different GoF tests since each GoF test is sensitive to different characteristics of the data.

## 3 THE NEED FOR A HIGHER CRITICISM: FROM LOCAL TO GLOBAL TESTING

GAEs are typically trained using size $m$ minibatches, or subsamples, of data. A natural starting point will be to push for matching the prior at the minibatch level. However, Section 3.2 discusses how *only* focusing on the minibatch level runs the risk of overfitting and how it can be mitigated with higher criticism. GoF tests assess whether a sample of observed data, $\{y_i\}_{i=1}^m$, is drawn from a distribution class $\mathcal{G}$. The most general opposing hypotheses are $\mathcal{H}_0 : \mathbb{P}_Y \in \mathcal{G}$ vs $\mathcal{H}_1 : \mathbb{P}_Y \notin \mathcal{G}$. A test is specified by a test statistic $T$ and a significance level $\alpha \in (0,1)$. If the observed value of $T$ is $T^\star$ when it is applied to $\{y_i\}_{i=1}^m$, i.e., $T(\{y_i\}_{i=1}^m) = T^\star$, then the (lower tail) p-value is $P(T(\{Y_i\}_{i=1}^m) \leq T^\star \mid \mathcal{H}_0)$, where $\{Y_i\}_{i=1}^m \sim \mathbb{P}_Y \in \mathcal{G}$. $\mathcal{H}_0$ is rejected in favor of $\mathcal{H}_1$ at level $\alpha$ if the p-value is less than $\alpha$, or equivalently, if $T^\star \leq T_\alpha$, where $T_\alpha$ is the $\alpha$-th quantile of $T$ under $\mathcal{H}_0$. The probability integral transform states that if $T$ has a continuous distribution under $\mathcal{H}_0$, the p-values are uniformly distributed on $(0,1)$ (Murdoch et al., 2008).

**Normality.** Multivariate normality (MVN) has received the most attention in the study on multivariate GoF tests. If $Y \in \mathbb{R}^d$ has a normal distribution, then for every $\mathbf{u} \in \mathcal{S}$, $Y^T \mathbf{u}$, the projection of $Y$ on $\mathbf{u}$, follows a UVN distribution (Rao et al., 1973), where $\mathcal{S} = \{\mathbf{u} \in \mathbb{R}^d \mid \|\mathbf{u}\| = 1\}$ is the unit sphere in $\mathbb{R}^d$. Conversely, if $Y$ has a non-normal distribution, the set of $\mathbf{u} \in \mathcal{S}$ with $Y^T\mathbf{u}$ being normal has Lebesgue measure 0 (Shao & Zhou, 2010). Thus, one way to test MVN is to apply GoF tests of univariate normality (UVN) to a set of (random) projections of $Y$ and then make a decision based on the resulting collection of test statistics. Many UVN tests exist (Looney, 1995; Mecklin & Mundfrom, 2005), each focusing on different distributional characteristics. We next briefly describe correlation based (CB) tests because they are directly related to the Wasserstein distance. For other test types, see appendix § B. In essence, CB tests on UVN are based on assessment of the correlation between empirical and UVN quantiles (Dufour et al., 1998). Two common CB tests are Shapiro-Wilk (SW) (Shapiro & Wilk, 1965) and Shapiro-Francia (SF) (Shapiro & Francia, 1972). Their test statistics will be denoted by $T_{SW}$ and $T_{SF}$ respectively. Both are closely related to $d_{W_2}$. For example, for $m \gg 1$, $T_{SW}$ is a strictly decreasing function of $d_{W_2}(\hat{\mathbb{P}}_{Y,m}, \mathcal{N}(0,1))$ (del Barrio et al., 1999), i.e., large $T_{SW}$ values correspond to small $d_{W_2}$, justifying the rule that $T_{SW}^\star > T_\alpha$ in order *not* to reject $\mathcal{H}_0$.

**The Benefits of GoF Testing in Autoencoders.** GoF tests on MVN inspect if $\mathbb{P}_Y$ is equal to some $\mathbb{P}_Z \in \mathcal{G}$ (Fig. 1), where $\mathcal{G}$ is the class of all Gaussians, i.e., MVN distributions. The primary justification for preferring $\mathbb{P}_Z$ as a Gaussian to the more commonly specified isotropic Gaussian (Makhzani et al., 2015; Kingma & Welling, 2013; Higgins et al., 2017) is that its covariance $\mathbf{\Sigma}$ is allowed to implicitly adapt during training. For example, $\mathbf{\Sigma}$ can be of full rank, allowing for correlations between the latent variables, or singular (i.e., a degenerate MVN, Figs. 5b,5c). The ability to adapt to degenerate MVN is of particular benefit. If $F$ is

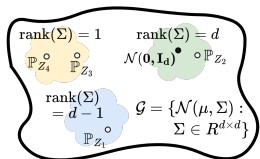

Figure 2: Illustration of distribution family $\mathcal{G}$.

differentiable and the dimension of the latent space is greater than the intrinsic dimension of the data distribution, then the latent variable $Y = F(X)$ only takes values in a manifold whose dimension is less than that of the latent space (Rubenstein et al., 2018). If $\mathbb{P}_Z$ is only allowed to be an isotropic Gaussian, or more generally, a Gaussian with a covariance matrix of full rank, the regularizer in Equation 1 will promote $F$ that can fill the latent space with $F(X)$, leading to poor sample quality and wrong proportions of generated images (Rubenstein et al., 2018). Since GoF tests use the class $\mathcal{G}$ of all Gaussians, they may help avoid such a situation. Note that this precludes the use of whitening, i.e. the transformation $\mathbf{\Sigma}^{-1/2}[\mathbf{X} - \mathbb{E}(\mathbf{X})]$ to get to $\mathcal{N}(\mathbf{0}, \mathbf{I})$, as the covariance matrix $\mathbf{\Sigma}$ may be singular. In other words, whitening confines available Gaussians to a subset much smaller than $\mathcal{G}$ (Fig. 2). Notably, a similar benefit was exemplified in the 2-Stage VAE, where decoupling manifold learning from learning the probability measure stabilized FID scores and sharpened generated samples when the intrinsic dimension is less than the dimension of the latent space (Dai & Wipf, 2019).

There are several additional benefits of GoF testing. First, it allows the use of higher criticism (HC), discussed in Section 3.2. GoF tests act at the local level (minibatch), while HC applies to the global level (training set). Second, many GoF tests produce closed-form test statistics, and thus do not require tuning and provide easily understood output. Lastly, testing for MVN via projections is unaffected by rank deficiency. This is *not* the case with the majority of multivariate GoF tests, e.g., the Henze–Zirkler test (Henze & Zirkler, 1990). In fact, any affine invariant test statistic for MVN must be a function of Mahalanobis distance, consequently requiring non-singularity (Henze, 2002).

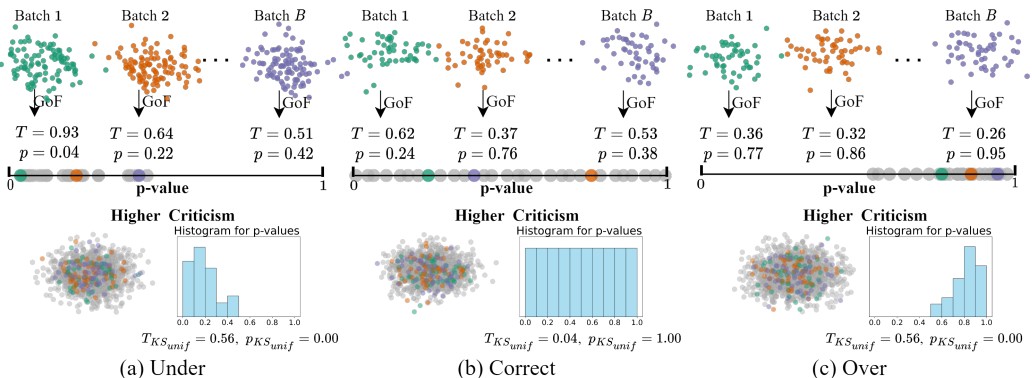

Figure 3: Model selection demonstration. If the regularization coefficient $\lambda$ is too small (a) or too big (c), the model is under (over) regularized and the minibatch p-values ($p$) are skewed right (left). As a result, global normality is rejected through the higher criticism principle for p-value uniformity ($p_{KS_{unif}} < \alpha$). (b) An appropriately chosen $\lambda$ fails to reject global normality ($p_{KS_{unif}} > \alpha$).

### 3.1 A LOCAL PERSPECTIVE: GOODNESS OF FIT FOR NORMALITY

In light of Section 2, the aim is to ensure proximity of $\mathbb{P}_X$ to $G_{\#}\mathbb{P}_Z$ by finding $F, G$ such that the reconstruction loss, $d$, is small and $F_{\#}\mathbb{P}_X$ is sufficiently close to prior $\mathbb{P}_Z$. As discussed above, $\mathbb{P}_Z$ is not completely specified but required to belong to $\mathcal{G}$. We can formulate the problem as

$$\min_{F,G} \mathbb{E}[d(X, G(F(X)))] \text{ subject to } F_{\#}\mathbb{P}_X \in \mathcal{G}.$$

To enforce the constraint in the minimization, a hypothesis test can be used to decide whether or not to reject the null $\mathcal{H}_0 : F_{\#}\mathbb{P}_X \in \mathcal{G}$. If $\mathcal{H}_0$ is rejected for *small* values of the test statistic, we can include this rejection criterion as a constraint to the optimization problem

$$\min_{F,G} \mathbb{E}[d(X, G(F(X)))] \text{ subject to } \mathbb{E}[T(\{F(X_i)\}_{i=1}^m)] > T_{\alpha}.$$

Rewriting with regularization coefficient $\lambda$ leads to the Lagrangian $\min_{F,G} \mathbb{E}[d(X, G(F(X)))] + \lambda(T_{\alpha} - \mathbb{E}[T(\{F(X_i)\}_{i=1}^m)])$. Since $\lambda \geq 0$, the final objective can be simplified to

$$\min_{F,G} \mathbb{E}[d(X, G(F(X))) - \lambda T(\{F(X_i)\}_{i=1}^m)]. \tag{2}$$

When the network is trained using a *single* minibatch of size $m$, making $Y$ *less* statistically distinguishable from $\mathcal{G}$ amounts to increasing $\mathbb{E}[T(\{F(X_i)\}_{i=1}^m)]$, where $X_i$ are i.i.d. $\sim \mathbb{P}_X$. However, if the training results in none of the minibatches yielding small $T$ values to reject $\mathcal{H}_0$ at a given $\alpha$ level, it also indicates a mismatch between the latent and prior distributions. This type of mismatch cannot be detected at the minibatch level; a more global view is detailed in the next section.

### 3.2 A GLOBAL PERSPECTIVE: GOODNESS OF FIT FOR UNIFORMITY - HIGHER CRITICISM

Neural networks are powerful universal function approximators (Hornik et al., 1989; Cybenko, 1989). With sufficient capacity it may be possible to train $F$ to *overfit*, i.e. to produce *too* many minibatches with large p-values. Under $\mathcal{H}_0$, the probability integral transform posits p-value uniformity. Therefore, it is expected that after observing many minibatches, approximately a fraction $\alpha$ of them will have p-values that are less than $\alpha$. This idea of a more encompassing test is known as Tukey's higher criticism (HC) (Donoho et al., 2004). While *each* minibatch GoF test is concerned with optimizing for indistinguishability from the prior distribution class $\mathcal{G}$, the HC test

---

**Algorithm 1** Evaluating Higher Criticism

**Require:** Trained encoder $F_{\theta}$, $\{\mathbf{x}_i\}_{i=1}^N$, GoF test $T$
1: **for** $i = 1 : \lfloor N/m \rfloor$ **do**
2:    Randomly sample minibatch $\mathbf{X}$ of size $m$
3:    $\mathbf{Y} = F_{\theta}(\mathbf{X})$
4:    **if** projection required **then**
5:      $T^{\star} = T(\mathbf{Yu})$, where $\mathbf{u} \in \mathcal{S}$
6:    **else**
7:      $T^{\star} = T(\mathbf{Y})$
8:    Calculate p-value of $T^{\star}$ and store it
9: Use $KS_{unif}$ to evaluate p-value set.

---

is concerned with testing whether the *collection* of p-values is uniformly distributed on $(0, 1)$, which may be accomplished through the Kolmogorov–Smirnov uniformity ($KS_{unif}$) test. See Algorithm 1 for a pseudo-code and Fig. 3 for an illustration of the HC process. HC and GoF form a symbiotic relationship; HC cannot exist by itself, and GoF by itself may over or under fit, producing p-values in incorrect proportions.

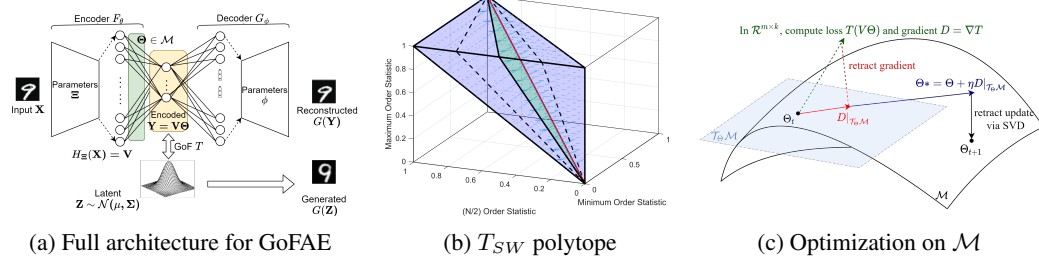

(a) Full architecture for GoFAE      (b) $T_{SW}$ polytope      (c) Optimization on $\mathcal{M}$

Figure 4: (a) GoFAE Architecture. (b) Example of singularity. Blue region is a polytope created as $\mathcal{U} = \{\mathbf{x} = [x_1, x_2, x_3] \in \mathbb{R}^3 : 0 \leq x_1 \leq x_2 \leq x_3 \leq 1\}$. For $\forall \mathbf{x} \in \mathcal{U}$, the coordinates of $\mathbf{x}$ are also its order statistics, the min, median and max coordinates corresponds to the $x, y$ and $z$-axis. The green simplex is where $T_{SW}(\{x_1, x_2, x_3\}) = 1$, and the region built with the dotted lines and the red line creates an acceptance region, outside of which is the rejection region. The light blue arrows are the derivatives at corresponding points and the red boundary line corresponds to the singularity where the gradient blows up. (c) Visualization for GoF optimization on a Stiefel manifold $\mathcal{M}$.

## 4   RIEMANNIAN SGD FOR OPTIMIZING A GOODNESS OF FIT AUTOENCODER

In practice $F$ and $G$ are neural networks parameterized by $\boldsymbol{\theta}$ and $\boldsymbol{\phi}$ respectively. The empirical GoFAE minibatch loss function based on Equation 2 is written as

$$\mathcal{L}(\boldsymbol{\theta}, \boldsymbol{\phi}; \{x_i\}_{i=1}^m) = \frac{1}{m} \sum_{i=1}^m d(x_i, G_{\boldsymbol{\phi}}(F_{\boldsymbol{\theta}}(x_i))) \pm \lambda T(\{F_{\boldsymbol{\theta}}(x_i)\}_{i=1}^m), \qquad (3)$$

A GoF test statistic $T$ is not merely an evaluation mechanism. Its gradient will impact how the model learns what characteristics of a sample indicate strong deviation from normality, carrying over to what the $\mathbb{P}_Y$ becomes. The following desiderata may help when selecting $T$. We denote a collection of sample observations by bold $\mathbf{X}, \mathbf{V}$, and $\mathbf{Y}$. We suppose that with probability one under $\mathbb{P}_X$ the following two conditions are satisfied.

- **GoF-Trainable:** $T(F_{\boldsymbol{\theta}}(\mathbf{X}))$ is almost everywhere continuously differentiable in feasible region $\Omega$.
- **GoF-Consistent:** There exists a $\boldsymbol{\theta}^\star$ in $\Omega$ such that $F_{\boldsymbol{\theta}^\star}(\mathbf{X})$ is consistent with the assumption that $F_{\boldsymbol{\theta}^\star}(\mathbf{X})$ are i.i.d. samples from the target distribution.

GoF-Trainable is needed if gradient based methods are used to optimize the network parameters. Consider a common encoder architecture $F_{\boldsymbol{\theta}}$ that consists of multiple feature extraction layers (forming a mapping $H_{\boldsymbol{\Xi}}(\mathbf{X})$) followed by a fully connected layer with parameter $\boldsymbol{\Theta}$. Thus, $F_{\boldsymbol{\theta}}(\mathbf{X}) = H_{\boldsymbol{\Xi}}(\mathbf{X})\boldsymbol{\Theta} = \mathbf{V}\boldsymbol{\Theta} = \mathbf{Y}$, and $\boldsymbol{\theta} = \{\boldsymbol{\Xi}, \boldsymbol{\Theta}\}$. The last layer is linear with no activation function as shown in Fig. 4a. With this design, normality can be optimized with respect to the last layer $\boldsymbol{\Theta}$ as discussed below. Given an GoF-Consistent statistic $T$, it is natural to seek a solution, $\boldsymbol{\theta}^\star$.

**Theorem 3.** *Suppose $\mathbf{V} \in \mathbb{R}^{m \times k}$ is of full row rank. For $\boldsymbol{\Theta} \in \mathbb{R}^{k \times d}$, define $\mathbf{Y} = \mathbf{V}\boldsymbol{\Theta}$. Denote $T_{SW} = T_{SW}(\mathbf{Yu})$ where $\mathbf{u} \in \mathbb{R}^d$ is a unit vector. Then, $T_{SW}$ is differentiable with respect to $\boldsymbol{\Theta}$ almost everywhere, and $\nabla_{\boldsymbol{\Theta}} T_{SW} = \mathbf{0}$ if and only if $T_{SW} = 1$.*

This theorem justifies the use of $T_{SW}$ as an objective function according to GoF desiderata. The largest possible value of $T_{SW}$ is 1, corresponding to an inability to detect deviation from normality no matter what level $\alpha$ is specified. See the appendix for other test choices.

**Identifiability, Singularities, and Stiefel Manifold.** If $\mathbf{Y} = \mathbf{V}\boldsymbol{\Theta}$ is Gaussian for some $\boldsymbol{\Theta}$, then $\mathbf{V}\boldsymbol{\Theta}\mathbf{M}$ is also Gaussian for any nonsingular matrix $\mathbf{M}$. Thus, any matrix of the form $\boldsymbol{\Theta}\mathbf{M}$ is a solution. This leads to several problems when optimizing an objective function containing $\boldsymbol{\Theta}$ as a variable and the test statistic as part of the objective. First, there is an identifiability issue since, without any restrictions, the set of solutions is infinite. Here, all that matters is the direction of the projection, so restricting the space to the unit sphere is reasonable. Second, almost all GoF tests for normality in the current literature are affine invariant. However, the gradient and Hessian of the test statistics will not be bounded. Fig. 4b illustrates an example of such a singularity issue.

**Theorem 4.** *Let $T(\{\mathbf{y}_i\}_{i=1}^m)$, $m \geq 3$ be any affine invariant test statistic that is non-constant and differentiable wherever $\mathbf{y}_i$ are not all equal. Then, for any $\mathbf{b}$, as $(\mathbf{y}_1, \ldots, \mathbf{y}_m) \to (\mathbf{b}, \ldots, \mathbf{b})$, $\sup \|\nabla T(\{\mathbf{y}_i\}_{i=1}^m)\| \to \infty$, where $\| \cdot \|$ is the Frobenius norm.*

If $\mathbf{\Theta}$ is searched without being kept away from zero or the diagonal line, traditional SGD results will not be applicable to yield convergence of $\mathbf{\Theta}$. A common strategy might be to lower bound the smallest singular value of $\mathbf{\Theta}^T\mathbf{\Theta}$. However, it does not solve the issue that any re-scaling of $\mathbf{\Theta}$ leads to another solution, so $\mathbf{\Theta}$ must be also upper bounded. It is thus desirable to restrict $\mathbf{\Theta}$ in such a way that avoids getting close to singular by both upper and lower bounding it in terms of its singular values. We propose to restrict $\mathbf{\Theta}$ to the compact Riemannian manifold of orthonormal matrices, i.e. $\mathbf{\Theta} \in \mathcal{M} = \{\mathbf{\Theta} \in \mathbb{R}^{k \times d} : \mathbf{\Theta}^T\mathbf{\Theta} = \mathbf{I}_d\}$ which is also known as the Stiefel manifold. This imposes a feasible region for $\boldsymbol{\theta} \in \Omega = \{\{\mathbf{\Xi}, \mathbf{\Theta}\} : \mathbf{\Theta}^T\mathbf{\Theta} = \mathbf{I}_d\}$. Optimization over a Stiefel manifold has been studied previously (Nishimori & Akaho, 2005; Absil et al., 2009; Cho & Lee, 2017; Bécigneul & Ganea, 2018; Huang et al., 2018; 2020; Li et al., 2020).

## 4.1 OPTIMIZING GOODNESS OF FIT TEST STATISTICS

A Riemannian metric provides a way to measure lengths and angles of tangent vectors of a smooth manifold. For the Stiefel manifold $\mathcal{M}$, in the Euclidean metric, we have $\langle \mathbf{\Gamma}_1, \mathbf{\Gamma}_2 \rangle = \text{trace}(\mathbf{\Gamma}_1^T\mathbf{\Gamma}_2)$ for any $\mathbf{\Gamma}_1, \mathbf{\Gamma}_2$ in the tangent space of $\mathcal{M}$ at $\mathbf{\Theta}$, $\mathcal{T}_{\mathbf{\Theta}}\mathcal{M}$. It is known that $\mathcal{T}_{\mathbf{\Theta}}\mathcal{M} = \{\mathbf{\Gamma} : \mathbf{\Gamma}^T\mathbf{\Theta} + \mathbf{\Theta}^T\mathbf{\Gamma} = 0, \mathbf{\Theta} \in \mathcal{M}\}$ (Li et al., 2020). For arbitrary matrix $\mathbf{\Theta} \in \mathbb{R}^{k \times d}$, the retraction back to $\mathcal{M}$, denoted $\mathbf{\Theta}\big|_{\mathcal{M}}$, can be performed via a singular value decomposition, $\mathbf{\Theta}\big|_{\mathcal{M}} = \mathbf{R}\mathbf{S}^T$, where $\mathbf{\Theta} = \mathbf{R}\mathbf{\Lambda}\mathbf{S}^T$ and $\mathbf{\Lambda}$ is the $d \times d$ diagonal matrix of singular values, $\mathbf{R} \in \mathbb{R}^{k \times d}$ has orthonormal columns, i.e. $\mathbf{R} \in \mathcal{M}$, and $\mathbf{S} \in \mathbb{R}^{d \times d}$ is an orthogonal matrix. The retraction of the gradient $\mathbf{D} = \nabla_{\mathbf{\Theta}}T_\star$ to $\mathcal{T}_{\mathbf{\Theta}}\mathcal{M}$, denoted by $\mathbf{D}\big|_{\mathcal{T}_{\mathbf{\Theta}}\mathcal{M}}$, can be accomplished by $\mathbf{D}\big|_{\mathcal{T}_{\mathbf{\Theta}}\mathcal{M}} = \mathbf{D} - \mathbf{\Theta}(\mathbf{\Theta}^T\mathbf{D} + \mathbf{D}^T\mathbf{\Theta})/2$ (Fig. 4c). The process of mapping Euclidean gradients to $\mathcal{T}_{\mathbf{\Theta}}\mathcal{M}$, updating $\mathbf{\Theta}$, and retracting back to $\mathcal{M}$ is well known (Nishimori & Akaho, 2005). The next result states that the Riemannian gradient for $T_{SW}$ has a finite second moment, which is needed for our convergence theorem in § 4.2.

**Theorem 5.** *Let $T = T_{SW}$ be as in Theorem 3. Denote by $\nabla_{\boldsymbol{\theta}}$, $\nabla_{\mathbf{\Theta}}$, the Riemannian gradient w.r.t. $\boldsymbol{\theta}$, $\mathbf{\Theta}$, respectively, and $\|\cdot\|$ the Frobenius norm. Let $\mathbf{B} = (\mathbf{I} - \mathbf{J}/m)\mathbf{V}$, i.e., $\mathbf{B}$ is obtained by subtracting from each row of $\mathbf{V}$ the mean of the rows. Suppose $(a)$ $\sup_{\mathbf{\Xi}}(\mathbb{E}[\|\mathbf{B}\|^4] + \mathbb{E}[\|\nabla_{\mathbf{\Xi}}\mathbf{B}\|^4]) < \infty$, and $(b)$ $\sup_{\|\mathbf{x}\|=1,\mathbf{\Xi}} \mathbb{E}[\|\mathbf{B}\mathbf{x}\|^{-4}] < \infty$. Then, $\sup_{\boldsymbol{\theta}} \mathbb{E}[\|\nabla_{\boldsymbol{\theta}}T\|^2] < \infty$.*

## 4.2 CONVERGENCE OF THE GOODNESS OF FIT AUTOENCODER

Since $\mathbf{\Theta} \in \mathcal{M}$, model training requires Riemannian SGD. The proof of Bonnabel (2013) for the convergence of SGD on a Riemannian manifold, which was extended from the Euclidean case (Bottou, 1998), requires conditions on step size, differentiability, and a uniform bound on the stochastic gradient. We show that this result holds under a much weaker condition, eliminating the need for a uniform bound. While we apply the resulting theorem to our specific problem with a Stiefel manifold, we emphasize that Theorem 6 is applicable to other suitable Riemannian manifolds.

**Theorem 6.** *Let $\mathcal{M}$ be a connected Riemannian manifold with injectivity radius uniformly bounded from below by $I > 0$. Let $C \in C^{(3)}(\mathcal{M})$ and $R_w$ be a twice continuously differentiable retraction. Let $z_0, z_1, \ldots$ be i.i.d. $\sim \zeta$ taking values in $\mathcal{Z}$. Let $H : \mathcal{Z} \times \mathcal{M} \to \mathcal{T}\mathcal{M}$ be a measurable function such that $\mathbb{E}[H(\zeta, w)] = \nabla C(w)$ for all $w \in \mathcal{M}$, where $\mathcal{T}\mathcal{M}$ is the tangent bundle of $\mathcal{M}$. Consider the SGD update $w_{t+1} = R_{w_t}(-\gamma_t H(z_t, w_t))$ with step size $\gamma_t > 0$ satisfying $\sum \gamma_t^2 < +\infty$ and $\sum \gamma_t = +\infty$. Suppose there exists a compact set $K$ such that all $w_t \in K$ and $\sup_{w \in K} \mathbb{E}[\|H(\zeta, w)\|^2] \leq A^2$ for some $A > 0$. Then, $C(w_t)$ converges almost surely and $\nabla C(w_t) \to 0$ almost surely.*

In our context, $C(\cdot)$ corresponds to Equation 3. The parameters of the GoFAE are $\{\boldsymbol{\theta}, \boldsymbol{\phi}\} = \{\mathbf{\Xi}, \mathbf{\Theta}, \boldsymbol{\phi}\}$ where $\mathbf{\Xi}, \boldsymbol{\phi}$ are defined in Euclidean space, a Riemannian manifold endowed with the Euclidean metric, and $\mathbf{\Theta}$ on the Stiefel manifold. Thus, $\{\mathbf{\Xi}, \mathbf{\Theta}, \boldsymbol{\phi}\}$ is a product manifold that is also Riemannian and $\{\mathbf{\Xi}, \mathbf{\Theta} \, \boldsymbol{\phi}\}$ are updated simultaneously. Convergence of the GoFAE holds from Proposition 5 and Theorem 6 provided that $\mathbf{\Xi}, \boldsymbol{\phi}$ stay within a compact set. Algorithms 2 and S1 give the pseudo-code for GoFAE optimization and the complete GoFAE and HC pipeline, respectively.

## 5 EXPERIMENTS

We evaluate generation and reconstruction performance, normality, and informativeness of GoFAE[1] using several GoF statistics on the MNIST (LeCun et al., 1998), CelebA (Liu et al., 2015) and

---

[1] Code can be found at `https://github.com/aripalmer/GoFAE`.

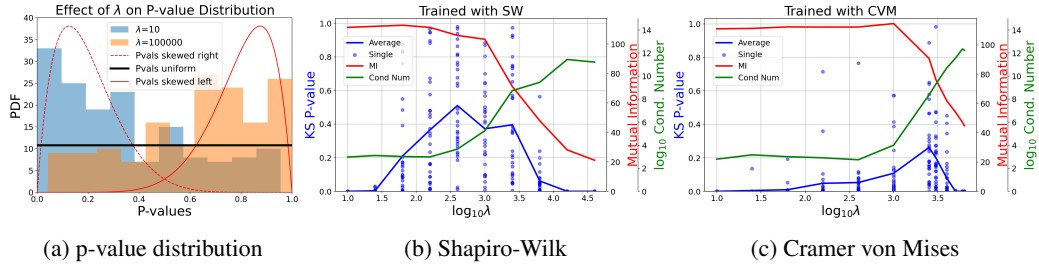

(a) p-value distribution    (b) Shapiro-Wilk    (c) Cramer von Mises

Figure 5: Effects of $\lambda$ on p-value distribution and mutual information for GoFAE models.

CIFAR10 (Krizhevsky et al., 2009) datasets and compare to several other GAE models. We emphasize that our goal is not to merely to produce competitive evaluation metrics, but to provide a principled way to balance the reconstruction versus prior matching trade-off. For MNIST and CelebA the architectures are based on Tolstikhin et al. (2017), while CIFAR10 is from Lippe (2022). The aim is to keep the architecture consistent across models with the exception of method specific components. See the appendix for complete architectures (§ C), training details (§ D), and additional results (§ E). The following results are from experiments on CelebA.

**Effect of $\lambda$ and Mutual Information (MI).** We investigated the balance between generation and reconstruction in the GoFAE by considering minibatch (local) GoF test p-value distributions, (global) normality, and MI as a function of $\lambda$ (Fig. 5). Global normality as assessed through the higher criticism (HC) principle for p-value uniformity is computed using Algorithm 1. We trained GoFAE on CelebA using $T_{SW}$ as the test statistic for $\lambda = 10$ (Fig. 5a, blue) and $\lambda = 100,000$ (Fig. 5a, yellow). For small $\lambda$, the model emphasizes reconstruction and the p-value distribution will be skewed right since the penalty for deviating from normality is small (Fig. 5a, red dashed line). As $\lambda$ increases, more emphasis is placed on normality and, for some $\lambda$, p-values are expected to be uniformly distributed (Fig. 5a, black line). If $\lambda$ is too large, the p-value distribution will be left-skewed (Fig. 5a, red solid line), which corresponds to overfitting the prior.

**Algorithm 2** GoFAE Optimization

**Require:** test $T$, learning rates: $\eta_1, \eta_2$, max iterations $J$, regularization coefficient $\lambda$
1: **Initialize:** $\boldsymbol{\Xi}, \boldsymbol{\Theta}, \boldsymbol{\phi}$
2: **while** $j < J$ **do**
3:  Sample minibatch of size $m$, $\mathbf{X}$
4:  $\mathbf{V} = F_{\boldsymbol{\Xi}_j}^{(1)}(\mathbf{X}), \mathbf{Y} = F_{\boldsymbol{\Theta}_j}^{(2)}(\mathbf{V}) = \mathbf{V}\boldsymbol{\Theta}_j$
5:  **if** test requires projection **then**
6:    $T^\star = T(\mathbf{Yu})$, where $\mathbf{u} \in \mathcal{S}$
7:  **else**
8:    $T^\star = T(\mathbf{Y})$
9:  $\mathcal{L} = d(\mathbf{X}, G_{\boldsymbol{\phi}}(\mathbf{Y})) \pm \lambda T^\star$
10:  $\boldsymbol{\Xi}_{j+1} = \boldsymbol{\Xi}_j - \eta_1 \nabla_{\boldsymbol{\Xi}}\mathcal{L}$ or other optim
11:  $\boldsymbol{\phi}_{j+1} = \boldsymbol{\phi}_j - \eta_1 \nabla_{\boldsymbol{\phi}}\mathcal{L}$ or other optim
12:  $\mathbf{D} = \nabla_{\boldsymbol{\Theta}}T^\star$
13:  $\boldsymbol{\Gamma} = \mathbf{D} - \boldsymbol{\Theta}_j(\boldsymbol{\Theta}_j^T\mathbf{D} + \mathbf{D}^T\boldsymbol{\Theta}_j)/2$
14:  $\boldsymbol{\Theta}_{j+1}' = \boldsymbol{\Theta}_j + \eta_2\boldsymbol{\Gamma}$
15:  Compute $\mathbf{R}\boldsymbol{\Lambda}\mathbf{S}^T = \text{SVD}(\boldsymbol{\Theta}_{j+1}')$
16:  $\boldsymbol{\Theta}_{j+1} = \boldsymbol{\Theta}_{j+1}'\big|_{\mathcal{M}} = \mathbf{R}\mathbf{S}^T$

We assessed the normality of GoFAE minibatch encodings using 30 repetitions of Algorithm 1 for different $\lambda$ (Figs. 5b-5c). The blue points and blue solid line represent the $KS_{unif}$ test p-values ($p_{KS_{unif}}$) and mean ($\bar{p}_{KS_{unif}} = \sum_{i=1}^{30} p_{KS_{unif_i}}$), respectively. HC rejects global normality ($\bar{p}_{KS_{unif}} < 0.05$) when the test statistic p-values are right-skewed (too many minibatches deviate from normality) or left-skewed (minibatches are *too* normal); this GoFAE property discourages the posterior from over-fitting the prior. The range of $\lambda$ values for which the distribution of GoF test p-values is indistinguishable from uniform is where the mean $\bar{p}_{KS_{unif}} \geq 0.05$ (Figs. 5b-5c).

Our method uses the class of Gaussians, $\mathcal{G} = \{\mathcal{N}(\boldsymbol{\mu}, \boldsymbol{\Sigma}) : \boldsymbol{\mu} \in \mathbb{R}^d, \boldsymbol{\Sigma} \in \mathbb{R}^{d \times d}\}$ as a prior, where $(\boldsymbol{\mu}, \boldsymbol{\Sigma})$ are the parameters of a Gaussian distribution denoting the mean and covariance matrix, respectively. When a model is finished training, we assume $F_\#\mathbb{P}_X \in \mathcal{G}$ and use estimates of the mean and covariance $(\hat{\boldsymbol{\mu}}, \hat{\boldsymbol{\Sigma}})$ for each GoFAE model to generate samples. Specifically, we drew $N = 10^4$ samples $\{\mathbf{z}_i\}_{i=1}^N$ to generate images $\{G_{\boldsymbol{\phi}}(\mathbf{z}_i)\}_{i=1}^N$. These images are then encoded, giving the set $\{\tilde{\mathbf{z}}_i\}_{i=1}^N$ with $\tilde{\mathbf{z}}_i = F_{\boldsymbol{\theta}}(G_{\boldsymbol{\phi}}(\mathbf{z}_i))$, resulting in a Markov chain $\{\mathbf{z}_i\}_{i=1}^N \to \{G_{\boldsymbol{\phi}}(\mathbf{z}_i)\}_{i=1}^N \to \{\tilde{\mathbf{z}}_i\}_{i=1}^N$. Assuming $\tilde{\mathbf{z}}_i$ are also normal, the data processing inequality gives a lower bound on the mutual information between $Z \sim \mathbb{P}_Z$ and $\tilde{Z} \sim F_\# G_\# \mathbb{P}_Z$, $I(Z, \tilde{Z})$ (Figs. 5b-5c, red line). As $\lambda$ increases the MI diminishes, suggesting the posterior overfits the prior as the encoding becomes independent of the data. The unimodality of $\bar{p}_{KS_{unif}}$ and monotonicity of MI suggests $\lambda$ can be selected solely based on $KS_{unif}$ without reference to a performance criterion.

**Gaussian Degeneracy.** Due to noise in the data, numerically $\mathbb{P}_Y$ can never be singular. Nevertheless, the experiments indicated the GoF test pushed $\mathbb{P}_Y$ to become "more and more singular" in the

Table 1: Evaluation of CelebA by MSE, FID scores, and samples with p-values from higher criticism.

| Algorithm | MSE ↓ | FID Score ↓ | | Kolmogorov-Smirnov Uniformity Test | | | | |
| | | Recon. | Gen. | SW | SF | CVM | KS | EP |
|---|---|---|---|---|---|---|---|---|
| AE-Baseline | 69.32 | 40.28 | 126.43 | 0.0(0.0) | 0.0(0.0) | 0.0(0.0) | 0.0(0.0) | 0.0(0.0) |
| VAE (fixed $\gamma$) | 101.78 | 49.29 | 56.78 | 0.01(0.05) | 0.0(0.01) | 0.10(0.19) | 0.30(0.30) | 0.23(0.27) |
| VAE (learned $\gamma$) | 68.09 | 34.54 | 58.15 | 0.0(0.0) | 0.0(0.0) | 0.0(0.0) | 0.0(0.0) | 0.0(0.0) |
| 2-Stage-VAE | 68.09 | 34.54 | 49.55 | 0.51(0.26) | 0.49(0.27) | 0.49(0.29) | 0.52(0.28) | 0.49(0.32) |
| $\beta(10)$-VAE | 235.37 | 99.59 | 101.30 | 0.56(0.31) | 0.46(0.35) | 0.47(0.28) | 0.38(0.25) | 0.44(0.27) |
| $\beta(2)$-VAE | 129.72 | 60.72 | 65.65 | 0.19(0.21) | 0.06(0.11) | 0.42(0.29) | 0.52(0.30) | 0.44(0.27) |
| WAE-GAN | 72.70 | 38.23 | 48.39 | 0.01(0.04) | 0.0(0.0) | 0.11(0.18) | 0.14(0.19) | 0.13(0.15) |
| GoFAE-SW(63) | **66.11** | 34.69 | 43.76 | 0.14(0.23) | 0.28(0.26) | 0.01(0.03) | 0.08(0.17) | 0.08(0.17) |
| GoFAE-SF(25) | 66.35 | **33.89** | **43.3** | 0.04(0.09) | 0.12(0.20) | 0.0(0.01) | 0.02(0.04) | 0.01(0.03) |
| GoFAE-CVM(398) | 67.43 | 37.50 | 46.74 | 0.19(0.22) | 0.34(0.29) | 0.05(0.08) | 0.16(0.27) | 0.19(0.28) |
| GoFAE-KS(63) | 66.62 | 35.14 | 43.93 | 0.0(0.00) | 0.0(0.00) | 0.0(0.00) | 0.00(0.00) | 0.00(0.00) |
| GoFAE-EP(10) | 67.03 | 39.71 | 48.53 | 0.03(0.08) | 0.06(0.10) | 0.00(0.00) | 0.00(0.00) | 0.02(0.04) |

sense that the condition number of its covariance matrix, $\kappa(\hat{\Sigma})$, became increasingly large. As $\lambda$ continues to increase, $\kappa(\hat{\Sigma})$ also increases (Figs. 5b-5c, green line), implying a Gaussian increasingly concentrated around a lower-dimensional linear manifold. We observed the same trend in the spectrum of singular values (SV) of $\hat{\Sigma}$ for each $\lambda$ after training with $T_{SW}$ (Fig. 6); while the spectrum did not exhibit large SVs, it had many small SVs. Notably, even when $\lambda$ was not too large, $\kappa(\hat{\Sigma})$ was relatively large, indicating a shift to a lower-dimension (Figs. 5b-5c). However, MI remained relatively large and the $KS_{unif}$ test suggests the $\mathbb{P}_Y$ is still indistinguishable from Gaussian. Together, these results are evidence that the GoFAE can adapt as needed to a reduced-dimension representation while maintaining the representation's informativeness.

**Reconstruction, Generation, and Normality.** We assessed the quality of the generated and test set reconstructed images using Fréchet Inception Distance (FID) (Heusel et al., 2017) based on $10^4$ samples and mean-square error (MSE) (Table 1). We compared the GoFAE with the AE (Bengio et al., 2013), VAE (Kingma & Welling, 2013), VAE with learned $\gamma$ (Dai & Wipf, 2019), 2-Stage VAE (Dai & Wipf, 2019), $\beta$-VAE (Higgins et al., 2017) and WAE-GAN (Tolstikhin et al., 2017); convergence was assessed by tracking the test set reconstruction error

Figure 6: SV of $\hat{\Sigma}$

over training epochs (Tables S3–S9 and Fig. S9). Figs. 7a, 7b are for models presented in Table 1. We selected the smallest $\lambda$ whose mean p-value of the HC uniformity test, $KS_{unif}$, was greater than $0.05$ for the GoFAE models. The correlation based GoF tests have the most competitive performance on FID and test set MSE. We assessed the normality of minibatch encodings across each method using several GoF tests for normality combined with $KS_{unif}$. We ran 30 repetitions of Algorithm 1 for each method and reported the mean (std) of the $KS_{unif}$ p-values in Table 1. In addition to superior MSE and FID scores, the GoFAE models obtained uniform p-values under varying GoF tests. The variability across GoF tests highlights the fact that different tests are sensitive to different distributional characteristics. Qualitative and quantitative assessments (§ E.3) and convergence plots (§ E.4) are given in the appendix for MNIST, CIFAR-10, and CelebA. Finally, an ablation study provided empirical justification for Riemannian SGD in the GoFAE (§ E.5).

# 6 CONCLUSION

We presented the GoFAE, a deterministic GAE based on optimal transport and Wasserstein distances that optimizes GoF test statistics. We showed that gradient based optimization of GoFAE induces identifiability issues, unbounded domains, and gradient singularities, which we resolve using Riemannian SGD. By using GoF statistics

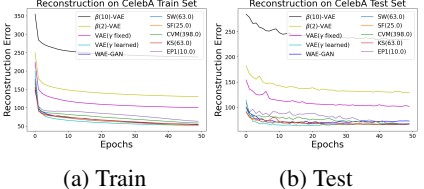

(a) Train       (b) Test

Figure 7: Reconstruction error on the CelebA training (a) and testing (b) sets.

to measure deviation from the prior *class* of Gaussians, our model is capable of implicitly adapting its covariance matrix during training from full-rank to singular, which we demonstrate empirically. We developed a performance agnostic model selection algorithm based on higher criticism of p-values for global normality testing. Collectively, empirical results show that GoFAE achieves comparable reconstruction and generation performance while retaining statistical indistinguishability from Gaussian in the latent space.

## ACKNOWLEDGEMENTS

The authors thank the anonymous reviewers for their comments which have improved this work. This work was partially supported by a National Science Foundation (NSF) grant: IIS-1718738, a National Institutes of Health (NIH) grant: K02DA043063, and a grant from US Department of Education: a Graduate Assistance in Areas of National Need (GAANN) program. J.Bi was also supported by NIH grants: 5R01MH119678-02 and 5R01DA051922-02.

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

# A PROOFS

## A.1 PROOF OF PROPOSITION 1.

Recall that in the setup of the result, $X$ is a random variable taking values in $\mathbb{R}^m$, $Y$ and $Z$ are random variables taking values in $\mathbb{R}^d$, $F : \mathbb{R}^m \to \mathbb{R}^d$ and $G : \mathbb{R}^d \to \mathbb{R}^m$ are two mappings, and $X$ and $Y$ are linked by $Y = F(X)$.

**Proposition 1.** *If $\mathcal{X}$ and $\mathcal{Z}$ are Euclidean spaces and the decoder $G$ is differentiable, then (a) $d_{W_2}(G_\# \mathbb{P}_Y, G_\# \mathbb{P}_Z) \leq \|\nabla G\|_\infty d_{W_2}(\mathbb{P}_Y, \mathbb{P}_Z)$, (b) $d_{W_2}(\mathbb{P}_X, G_\# \mathbb{P}_Z) \leq [\mathbb{E}\|X - G(Y)\|^2]^{1/2} + \|\nabla G\|_\infty d_{W_2}(\mathbb{P}_Y, \mathbb{P}_Z)$.*

*Proof.* (a) Recall that for any two probability measures $\mu$ and $\nu$ on the same Euclidean space,

$$d_{W_2}^2(\mu, \nu) = \inf_{\pi \in \Pi(\mu, \nu)} \int \|y - z\|^2 \pi(\mathrm{d}z, \mathrm{d}z).$$

Let $f(x, y) = (G(x), G(y))$. For any $Q \in \Pi(\mathbb{P}_Y, \mathbb{P}_Z)$, the measure $Q'(\cdot) = Q \circ f^{-1}(\cdot) \in \Pi(\mathbb{P}_{G(Y)}, \mathbb{P}_{G(Z)})$, so

$$d_W^2(\mathbb{P}_{G(Y)}, \mathbb{P}_{G(Z)}) \leq \int \|y' - z'\|^2 Q'(\mathrm{d}y', \mathrm{d}z') = \int \|G(y) - G(z)\|^2 Q(\mathrm{d}y, \mathrm{d}z)$$

$$\leq \|\nabla G\|_\infty^2 \int \|y - z\|^2 Q(\mathrm{d}y, \mathrm{d}z).$$

Taking infimum of the right hand side over all $Q \in \Pi(\mathbb{P}_Y, \mathbb{P}_Z)$ gives the result.

(b) By triangular inequality for $d_{W_2}$ and part (a),

$$d_{W_2}(\mathbb{P}_X, \mathbb{P}_{G(Z)}) \leq d_{W_2}(\mathbb{P}_X, \mathbb{P}_{G(Y)}) + d_{W_2}(\mathbb{P}_{G(y)}, \mathbb{P}_{G(Z)})$$

$$\leq [\mathbb{E}\|X - G(Y)\|^2]^{1/2} + \|\nabla G\|_\infty d_{W_2}(\mathbb{P}_Y, \mathbb{P}_Z). \qquad \square$$

## A.2 PROOF OF PROPOSITION 2.

**Proposition 2.** *Let $\hat{\mathbb{P}}_{X,n}$ be the empirical distribution of samples $\{X_i\}_{i=1}^n$, and $\hat{\mathbb{P}}_{Y,n}$ be the empirical distribution of $\{Y_i = F(X_i)\}_{i=1}^n$. Assume that $F(X)$ is differentiable with respect to $X$ with bounded gradients $\nabla F(X)$. Then, (a) $d_{W_2}(\hat{\mathbb{P}}_{Y,n}, \mathbb{P}_Y) \leq \|\nabla F\|_\infty d_{W_2}(\hat{\mathbb{P}}_{X,n}, \mathbb{P}_X)$, (b) $d_{W_2}(\mathbb{P}_Y, \mathbb{P}_Z) \leq d_{W_2}(\hat{\mathbb{P}}_{Y,n}, \mathbb{P}_Z) + \|\nabla F\|_\infty d_{W_2}(\hat{\mathbb{P}}_{X,n}, \mathbb{P}_X)$.*

*Proof.* (a) is a direct consequence of Proposition 1 (a), except that $G$ therein is replaced with $F$. (b) follows by combining (a) with $d_{W_2}(\mathbb{P}_Y, \mathbb{P}_Z) \leq d_{W_2}(\hat{\mathbb{P}}_{Y,m}, \mathbb{P}_Z) + d_{W_2}(\hat{\mathbb{P}}_{Y,m}, \mathbb{P}_Y)$. $\qquad \square$

## A.3 PROOF OF THEOREM 3.

As described in Section B.1, the Shapiro–Wilk statistic of a sample $X_1, \ldots, X_m$ is

$$T_{SW} = \frac{(\sum_{i=1}^m a_i X_{(i)})^2}{\sum_{i=1}^m (X_i - \bar{X})^2},$$

where $X_{(1)} \leq X_{(2)} \cdots \leq X_{(m)}$ are the order statistics and $a_i$ are certain constants that are all different. Let $\mathbf{a} = (a_1, \ldots, a_m)^T$. We only need the fact that $\|\mathbf{a}\| = 1$ and $\mathbf{1}^T \mathbf{a} = 0$, where $\mathbf{1}$ is the vector of $m$ 1's.

**Theorem 3.** *Suppose $\mathbf{V} \in \mathbb{R}^{m \times k}$ is of full row rank. For $\boldsymbol{\Theta} \in \mathbb{R}^{k \times d}$, define $\mathbf{Y} = \mathbf{V}\boldsymbol{\Theta}$. Denote $T_{SW} = T_{SW}(\mathbf{Yu})$ where $\mathbf{u} \in \mathbb{R}^d$ is a unit vector. Then, $T_{SW}$ is differentiable with respect to $\boldsymbol{\Theta}$ almost everywhere, and $\nabla_{\boldsymbol{\Theta}} T_{SW} = \mathbf{0}$ if and only if $T_{SW} = 1$.*

*Proof.* Let $\mathbf{V}_*$ be a row permutation of $\mathbf{V}$ such that the coordinates of $\mathbf{V}_* \boldsymbol{\Theta} \mathbf{u}$ are in increasing order. Denote by $\mathbf{I}$ the $m \times m$ identity matrix and $\mathbf{J} = \mathbf{11}^T$. Put $\mathbf{B} = (\mathbf{I} - \mathbf{J}/m)\mathbf{V}_*$ and note that $\mathbf{a}^T \mathbf{V}_* = \mathbf{a}^T \mathbf{B}$ as $\mathbf{a}^T \mathbf{J} = \mathbf{a}^T \mathbf{11}^T = (\mathbf{1}^T \mathbf{a})^T \mathbf{1} = \mathbf{0}$. Then

$$T_{SW} = \frac{(\mathbf{a}^T \mathbf{V}_* \boldsymbol{\Theta} \mathbf{u})^2}{\|(\mathbf{I} - \mathbf{J}/m)\mathbf{V}_* \boldsymbol{\Theta} \mathbf{u}\|^2} = \frac{(\mathbf{a}^T \mathbf{B} \boldsymbol{\Theta} \mathbf{u})^2}{\|\mathbf{B} \boldsymbol{\Theta} \mathbf{u}\|^2}. \tag{S4}$$

It is easy to see that $T_{SW}$ is differentiable at $\mathbf{\Theta}$ if all $\mathbf{v}_i^T\mathbf{\Theta}\mathbf{u}$ are different, where $\mathbf{v}_1^T, \ldots, \mathbf{v}_m^T$ are the rows of $\mathbf{V}$. Also, in this case, $\mathbf{V}_*$ is unique and $\mathbf{B}\mathbf{\Theta}\mathbf{u} \neq 0$. On the other hand, for $i \neq j$, since $\mathbf{u} \neq \mathbf{0}$ and $\mathbf{v}_i \neq \mathbf{v}_j$ as $\mathbf{V}$ is of full row rank, the set of $\mathbf{\Theta}$ with $(\mathbf{v}_i - \mathbf{v}_j)^T\mathbf{\Theta}\mathbf{u} = 0$ is a strict linear subspace of $\mathbb{R}^{k \times d}$, so has Lebesgue measure 0. Then $T_{SW}$ is differentiable in $\mathbf{\Theta}$ almost everywhere.

Fix $\mathbf{\Theta}$ with all the coordinates of $\mathbf{V}\mathbf{\Theta}\mathbf{u}$ being different. Then

$$\nabla_{\mathbf{\Theta}} T_{SW} = \frac{2(\mathbf{a}^T\mathbf{B}\mathbf{\Theta}\mathbf{u})}{\|\mathbf{B}\mathbf{\Theta}\mathbf{u}\|^2}\mathbf{B}^T\left[\mathbf{a} - \frac{(\mathbf{a}^T\mathbf{B}\mathbf{\Theta}\mathbf{u})(\mathbf{B}\mathbf{\Theta}\mathbf{u})}{\|\mathbf{B}\mathbf{\Theta}\mathbf{u}\|^2}\right]\mathbf{u}^T. \tag{S5}$$

Note that $\mathbf{a}^T\mathbf{B}\mathbf{\Theta}\mathbf{u}$ is a scalar. Also, $\mathbf{a}^T\mathbf{B}\mathbf{\Theta}\mathbf{u} \neq 0$, for otherwise from (S4) $T_{SW} = 0$, contradicting Lemma 3 (Shapiro & Wilk, 1965).

Suppose $\nabla_{\mathbf{\Theta}} T_{SW} = \mathbf{0}$. Then from (S5),

$$\mathbf{B}^T\left[\mathbf{a} - \frac{(\mathbf{a}^T\mathbf{B}\mathbf{\Theta}\mathbf{u})(\mathbf{B}\mathbf{\Theta}\mathbf{u})}{\|\mathbf{B}\mathbf{\Theta}\mathbf{u}\|^2}\right]\mathbf{u}^T = \mathbf{0}.$$

Let $c = \frac{\|\mathbf{B}\mathbf{\Theta}\mathbf{u}\|^2}{\mathbf{a}^T\mathbf{B}\mathbf{\Theta}\mathbf{u}}$ and $\mathbf{h} = \mathbf{B}\mathbf{\Theta}\mathbf{u} - c\mathbf{a}$. Then the above equation can be written as

$$\mathbf{B}^T\left[\mathbf{a} - \frac{\mathbf{B}\mathbf{\Theta}\mathbf{u}}{c}\right]\mathbf{u}^T = -c^{-1}\mathbf{B}^T\mathbf{h}\mathbf{u}^T = \mathbf{0},$$

giving $\mathbf{B}^T\mathbf{h}\mathbf{u}^T\mathbf{u} = \|\mathbf{u}\|^2\mathbf{B}^T\mathbf{h} = \mathbf{0}$. From $\mathbf{u} \neq \mathbf{0}$, $\mathbf{0} = \mathbf{h}^T\mathbf{B} = \mathbf{h}^T(\mathbf{I} - \mathbf{J}/m)\mathbf{V}_* = (\mathbf{h} - k\mathbf{1})^T\mathbf{V}_*$ with $k = (\mathbf{h}^T\mathbf{1})/m$. Because $\mathbf{V}_*$ is of full row rank, $\mathbf{h} = k\mathbf{1}$. However, $\mathbf{1}^T\mathbf{h} = \mathbf{1}^T\mathbf{B}\mathbf{\Theta}\mathbf{u} - c\mathbf{1}^T\mathbf{a}$. Since $\mathbf{1}^T\mathbf{B} = \mathbf{1}^T(\mathbf{I} - \mathbf{J}/m)\mathbf{V} = \mathbf{0}$ and as pointed earlier, $\mathbf{1}^T\mathbf{a} = 0$, then $\mathbf{1}^T\mathbf{h} = 0$. As a result $k = 0$, giving $\mathbf{h} = \mathbf{0}$, i.e. $\mathbf{B}\mathbf{\Theta}\mathbf{u} = c\mathbf{a}$. Plugging this into (S5) and noting that $\|\mathbf{a}\| = 1$, $T_{SW} = 1$.

On the other hand, by Lemma 2 of (Shapiro & Wilk, 1965), $T_{SW} = 1$ if and only if $\mathbf{B}\mathbf{\Theta}\mathbf{u} = s\mathbf{a}$ for some scalar $s \neq 0$. Clearly in this case all the coordinates of $\mathbf{B}\mathbf{\Theta}\mathbf{u}$ are different, so $T_{SW}$ is differentiable at $\mathbf{\Theta}$. Since 1 is the maximum value of $T_{SW}$, then $\nabla_{\mathbf{\Theta}} T_{SW} = \mathbf{0}$. □

## A.4 PROOF OF THEOREM 4.

**Theorem 4.** *Let $T(\{\mathbf{y}_i\}_{i=1}^m)$, $m \geq 3$ be any affine invariant test statistic that is non-constant and differentiable wherever $\mathbf{y}_i$ are not all equal. Then, for any $\mathbf{b}$, as $(\mathbf{y}_1, \ldots, \mathbf{y}_m) \to (\mathbf{b}, \ldots, \mathbf{b})$, $\sup \|\nabla T(\{\mathbf{y}_i\}_{i=1}^m)\| \to \infty$, where $\|\cdot\|$ is the Frobenius norm.*

*Proof.* First, $\forall s > 0$, $T(s\mathbf{y}_1 + \mathbf{b}, \ldots, s\mathbf{y}_m + \mathbf{b}) = T(\mathbf{y}_1, \ldots, \mathbf{y}_m)$ then $\nabla T(s\mathbf{y}_1 + \mathbf{b}, \ldots, s\mathbf{y}_m + \mathbf{b}) = \frac{1}{s}\nabla T(\mathbf{y}_1, \ldots, \mathbf{y}_m)$. By assumption $\nabla T(\mathbf{y}_1, \ldots, \mathbf{y}_m) \neq 0$ for some $(\mathbf{y}, \ldots, \mathbf{y}_m)$. Then let $s \to 0$, $\|\nabla T(s\mathbf{y}_1 + \mathbf{b}, \ldots, s\mathbf{y}_m + \mathbf{b})\| = \frac{1}{s}\|\nabla T(\mathbf{y}_1, \ldots, \mathbf{y}_m)\| \to \infty$. □

## A.5 PROOF OF THEOREM 5.

Recall $\mathbf{X} = (X_1, \ldots, X_m)$ and $\mathbf{V} = H_{\mathbf{\Xi}}(\mathbf{X})$, where the $X_i$'s are i.i.d. sample observations. Note that $\mathbf{V}$ is a function of $\mathbf{\Xi}$ and $\mathbf{X}$ but not a function of $\mathbf{\Theta}$.

**Theorem 5.** *Let $T = T_{SW}$ be as in Theorem 3. Denote by $\nabla_{\boldsymbol{\theta}}$, $\nabla_{\mathbf{\Theta}}$, the Riemannian gradient w.r.t. $\boldsymbol{\theta}$, $\mathbf{\Theta}$, respectively, and $\|\cdot\|$ the Frobenius norm. Let $\mathbf{B} = (\mathbf{I} - \mathbf{J}/m)\mathbf{V}$, i.e., $\mathbf{B}$ is obtained by subtracting from each row of $\mathbf{V}$ the mean of the rows. Suppose $(a)$ $\sup_{\mathbf{\Xi}}(\mathbb{E}[\|\mathbf{B}\|^4] + \mathbb{E}[\|\nabla_{\mathbf{\Xi}}\mathbf{B}\|^4]) < \infty$, and $(b)$ $\sup_{\|\mathbf{x}\|=1, \mathbf{\Xi}} \mathbb{E}[\|\mathbf{B}\mathbf{x}\|^{-4}] < \infty$. Then, $\sup_{\boldsymbol{\theta}} \mathbb{E}[\|\nabla_{\boldsymbol{\theta}}T\|^2] < \infty$.*

*Proof.* Since all the assumptions and assertion of Theorem 6 are invariant under any permutation of the rows of $\mathbf{V}$, for ease of notation and without loss of generality, suppose the coordinates of $\mathbf{V}\mathbf{\Theta}\mathbf{u}$ are in increasing order. By the construction, the coordinates of $\mathbf{B}\mathbf{\Theta}\mathbf{u}$ are also increasing and have mean $\mathbf{0}$. Then

$$T = \frac{(\mathbf{a}^T\mathbf{B}\mathbf{\Theta}\mathbf{u})^2}{\|\mathbf{B}\mathbf{\Theta}\mathbf{u}\|^2}.$$

By $\nabla_{\boldsymbol{\theta}}T = (\nabla_{\mathbf{\Theta}}T, \nabla_{\mathbf{\Xi}}T)$, it suffices to show $\sup_{\boldsymbol{\theta}} \mathbb{E}(\|\nabla_{\mathbf{\Theta}}T\|^2) < \infty$ and $\sup_{\boldsymbol{\theta}} \mathbb{E}(\|\nabla_{\mathbf{\Xi}}T\|^2) < \infty$.

Regard the Stiefel manifold $\mathcal{M}$ as a subset of $\mathbb{R}^{k \times d}$ equipped with the inner product $\langle \mathbf{\Gamma}_1, \mathbf{\Gamma}_2 \rangle = \text{tr}(\mathbf{\Gamma}_1^T \mathbf{\Gamma}_2)$, which is simply the Euclidean inner product of the vectorized $\mathbf{\Gamma}_1$ and $\mathbf{\Gamma}_2$. The Riemannian gradient $\nabla_{\mathbf{\Theta}}$ is obtained under this inner product and is equal to the orthogonal projection of $\bar{\nabla}_{\mathbf{\Theta}}$ onto $\mathcal{T}_{\mathbf{\Theta}}\mathcal{M}$, where $\bar{\nabla}_{\mathbf{\Theta}}$ denotes the Euclidean gradient. Then $\|\nabla_{\mathbf{\Theta}}T\| \leq \|\bar{\nabla}_{\mathbf{\Theta}}T\|$, so it is enough to show $\sup_{\boldsymbol{\theta}} \mathbb{E}(\|\bar{\nabla}_{\mathbf{\Theta}}T\|^2) < \infty$. From (S5),

$$\bar{\nabla}_{\mathbf{\Theta}}T = \frac{2(\mathbf{a}^T\mathbf{B}\mathbf{\Theta}\mathbf{u})}{\|\mathbf{B}\mathbf{\Theta}\mathbf{u}\|^2}\mathbf{B}^T\left[\mathbf{a} - \frac{(\mathbf{a}^T\mathbf{B}\mathbf{\Theta}\mathbf{u})(\mathbf{B}\mathbf{\Theta}\mathbf{u})}{\|\mathbf{B}\mathbf{\Theta}\mathbf{u}\|^2}\right]\mathbf{u}^T.$$

Then by $\|\mathbf{u}\| = 1$,

$$\|\bar{\nabla}_{\mathbf{\Theta}}T\| \leq \frac{4\|\mathbf{a}\|^2\|\mathbf{B}\|}{\|\mathbf{B}\mathbf{\Theta}\mathbf{u}\|},$$

so by Cauchy-Schwarz inequality

$$\mathbb{E}(\|\bar{\nabla}_{\mathbf{\Theta}}T\|^2) \leq 16\|\mathbf{a}\|^4\left[\mathbb{E}(\|\mathbf{B}\|^4)\mathbb{E}(\|\mathbf{B}\mathbf{\Theta}\mathbf{u}\|^{-4})\right]^{1/2}$$

$$\leq 16\|\mathbf{a}\|^4\left[\mathbb{E}(\|\mathbf{B}\|^4)\sup_{\|\mathbf{x}\|=1,\mathbf{\Xi}}\mathbb{E}(\|\mathbf{B}\mathbf{x}\|^{-4})\right]^{1/2},$$

where the second inequality is due to the independence of $\mathbf{u}$ and $\mathbf{B}$. Then by assumptions (b) and (c), $\sup_{\boldsymbol{\theta}} \mathbb{E}(\|\bar{\nabla}_{\mathbf{\Theta}}T\|^2) < \infty$.

Next, since $\mathbf{\Xi}$ lives in a Euclidean space, the Riemannian gradient of $T$ w.r.t. $\mathbf{\Xi}$ consists of the partial derivatives of $T$ w.r.t. of its coordinates. For each coordinate $\xi$ of $\mathbf{\Xi}$,

$$\partial_\xi T := \frac{\partial T}{\partial \xi} = 2\left[\frac{\mathbf{a}^T(\partial_\xi\mathbf{B})\mathbf{\Theta}\mathbf{u}(\mathbf{a}^T\mathbf{B}\mathbf{\Theta}\mathbf{u})}{\|\mathbf{B}\mathbf{\Theta}\mathbf{u}\|^2} - \frac{(\mathbf{a}^T\mathbf{B}\mathbf{\Theta}\mathbf{u})^2(\mathbf{B}\mathbf{\Theta}\mathbf{u})^T(\partial_\xi\mathbf{B})\mathbf{\Theta}\mathbf{u}}{\|\mathbf{B}\mathbf{\Theta}\mathbf{u}\|^4}\right].$$

Then by $\|\mathbf{\Theta}\mathbf{u}\| = 1$,

$$|\partial_\xi T| \leq \frac{4\|\mathbf{a}\|^2\|\partial_\xi\mathbf{B}\|}{\|\mathbf{B}\mathbf{\Theta}\mathbf{u}\|}.$$

Taking the sum of $(\partial_\xi T)^2$ over all the coordinates of $\mathbf{\Xi}$ then yields

$$\|\nabla_{\mathbf{\Xi}}T\|^2 \leq \frac{16\|\mathbf{a}\|^4\|\nabla_{\mathbf{\Xi}}\mathbf{B}\|^2}{\|\mathbf{B}\mathbf{\Theta}\mathbf{u}\|^2}.$$

So by Cauchy-Schwarz inequality

$$\mathbb{E}(\nabla_{\mathbf{\Xi}}T)^2 \leq 16\|\mathbf{a}\|^4\left[\mathbb{E}(\|\nabla_{\mathbf{\Xi}}\mathbf{B}\|^4)\mathbb{E}(\|\mathbf{B}\mathbf{\Theta}\mathbf{u}\|^{-4})\right]^{1/2}$$

$$\leq 16\|\mathbf{a}\|^4\left[\mathbb{E}(\|\nabla_{\mathbf{\Xi}}\mathbf{B}\|^4)\sup_{\|\mathbf{x}\|=1,\mathbf{\Xi}}\mathbb{E}(\|\mathbf{B}\mathbf{x}\|^{-4})\right]^{1/2},$$

where the second inequality is again due to the independence of $\mathbf{u}$ and $\mathbf{B}$. Then by assumptions (b) and (c), $\sup_{\boldsymbol{\theta}} \mathbb{E}(\|\nabla_{\mathbf{\Xi}}T\|^2) < \infty$. $\qquad\square$

## A.6 PROOF OF THEOREM 6.

The following result is a relaxation of Theorems 1 and 2 in (Bonnabel, 2013). Let $(\gamma_t)_{t\geq 0} = (\gamma_0, \gamma_1, \gamma_2, \ldots)$ be a sequence of step sizes. Suppose $C(\cdot)$ is a three times continuously differentiable cost function on a smooth connected Riemannian manifold $\mathcal{M}$, $(z_t)_{t\geq 0}$ is a sequence of i.i.d. random variables taking values in a measurable space $\mathcal{Z}$, and $H(\cdot, \cdot)$ is a measurable function on $\mathcal{Z} \times \mathcal{T}\mathcal{M}$ such that $\mathbb{E}_z H(z, w) = \nabla C(w)$, where $\mathcal{T}\mathcal{M}$ is the tangent bundle of $\mathcal{M}$ and $z \sim z_t$. Also suppose $w_0 \in \mathcal{M}$ is independent of $(z_t)_{t\geq 0}$.

**Theorem 6.** *Let $\mathcal{M}$ be a connected Riemannian manifold with injectivity radius uniformly bounded from below by $I > 0$. Let $C \in C^{(3)}(\mathcal{M})$ and $R_w$ be a twice continuously differentiable retraction. Let $z_0, z_1, \ldots$ be i.i.d. $\sim \zeta$ taking values in $\mathcal{Z}$. Let $H : \mathcal{Z} \times \mathcal{M} \to \mathcal{T}\mathcal{M}$ be a measurable function such that $\mathbb{E}[H(\zeta, w)] = \nabla C(w)$ for all $w \in \mathcal{M}$, where $\mathcal{T}\mathcal{M}$ is the tangent bundle of $\mathcal{M}$. Consider the SGD update $w_{t+1} = R_{w_t}(-\gamma_t H(z_t, w_t))$ with step size $\gamma_t > 0$ satisfying $\sum \gamma_t^2 < +\infty$ and $\sum \gamma_t = +\infty$. Suppose there exists a compact set $K$ such that all $w_t \in K$ and $\sup_{w\in K}\mathbb{E}[\|H(\zeta, w)\|^2] \leq A^2$ for some $A > 0$. Then, $C(w_t)$ converges almost surely and $\nabla C(w_t) \to 0$ almost surely.*

As in (Bonnabel, 2013), the proof of Theorem 6 starts with the the following result which is of interest in its own right.

**Proposition S1.** *Let $(\gamma_t)_{t\geq 0}$ and $\mathcal{M}$ be as in Theorem 7. Consider the update $w_{t+1} = \exp_{w_t}(-\gamma_t H(z_t, w_t))$, where $\exp_w$ is the exponential map at $w$. Suppose there exists a compact set $K$ such that $w_t \in K$ for all $t \geq 0$. We also suppose for some $A > 0$, $\mathbb{E}_z(\|H(z, w)\|^2) \leq A^2$ for all $w \in K$. Then, $C(w_t)$ converges a.s. and $\nabla C(w_t) \to 0$ a.s.*

*Proof of Proposition S1.* The proof builds upon the one for Theorem 1 (Bonnabel, 2013). Let $\mathcal{F}_t = \sigma(w_0, z_0, \ldots, z_{t-1})$. Then for $t \geq 1$, $w_t$ is $\mathcal{F}_t$ measurable. If $\gamma_t\|H(z_t, w_t)\| < I$, then $\exp_{w_t}\{sH(z_t, w_t)\}_{0 \leq s \leq \gamma_t}$ is the geodesic linking $w_{t+1}$ and $w_t$, so as in the proof of Theorem 1 in (Bonnabel, 2013), the Taylor formula yields

$$C(w_{t+1}) - C(w_t) \leq -\gamma_t\langle H(z_t, w_t), \nabla C(w_t)\rangle + \gamma_t^2\|H(z_t, w_t)\|^2 k_2,$$

where $\langle\,,\,\rangle$ is the Riemannian inner product at $\mathcal{T}_{w_t}\mathcal{M}$. The inequality has the same form as equation (5) in (Bonnabel, 2013) but with $k_2 = \sup_{w \in K_0} \|\nabla^2 C\|$, i.e., $\|(\nabla^2 C(w))v\| \leq k_2\|v\|$ for all $w \in K_0$ and $v \in \mathcal{T}_w\mathcal{M}$, where $K_0$ is the compact set of all points with distance at most $I$ from $K$. Define events

$$E_{-1} = \emptyset, \quad E_t = \{\|H(z_t, w_t)\| < I/\gamma_t\}, \quad t \geq 0. \tag{S6}$$

Denote by $\mathbf{1}_E$ the indicator function of an event $E$. Then by $C(w) \geq 0$,

$$C(w_{t+1})\mathbf{1}_{E_t} \leq C(w_t) - \gamma_t\langle H(z_t, w_t), \nabla C(w_t)\rangle\mathbf{1}_{E_t} + \gamma_t^2\|H(z_t, w_t)\|^2 k_2. \tag{S7}$$

Taking expectations conditional on $\mathcal{F}_t$ on both sides of the equality,

$$\mathbb{E}[C(w_{t+1})\mathbf{1}_{E_t} \,|\, \mathcal{F}_t] \leq C(w_t) - \gamma_t\mathbb{E}(\langle H(z_t, w_t), \nabla C(w_t)\rangle \,|\, \mathcal{F}_t)$$
$$+ \gamma_t\mathbb{E}(\langle H(z_t, w_t), \nabla C(w_t)\rangle\mathbf{1}_{E_t^c} \,|\, \mathcal{F}_t) + \gamma_t^2\mathbb{E}(\|H(z_t, w_t)\|^2 \,|\, \mathcal{F}_t)k_2. \tag{S8}$$

Since $z_t$ is independent from $\mathcal{F}_t$ while $w_t$ is $\mathcal{F}_t$-measurable,

$$\mathbb{E}(\langle H(z_t, w_t), \nabla C(w_t)\rangle \,|\, \mathcal{F}_t) = \mathbb{E}_z(\langle H(z, w_t), \nabla C(w_t)\rangle) = \|\nabla C(w_t)\|^2,$$
$$\mathbb{E}(\|H(z_t, w_t)\|^2 \,|\, \mathcal{F}_t) = \mathbb{E}_z(\|H(z, w_t)\|^2) \leq A^2, \tag{S9}$$

where in $\mathbb{E}_z(\cdot)$, $w_t$ is treated as a fixed value. On the other hand,

$$\mathbb{E}(\langle H(z_t, w_t), \nabla C(w_t)\rangle\mathbf{1}_{E_t^c} \,|\, \mathcal{F}_t) = \langle\mathbb{E}[H(z_t, w_t)\mathbf{1}_{E_t^c} \,|\, \mathcal{F}_t], \nabla C(w_t)\rangle$$
$$\leq \mathbb{E}[\|H(z_t, w_t)\|\mathbf{1}_{E_t^c} \,|\, \mathcal{F}_t]k_1,$$

where $k_1 = \sup_K \|\nabla C\|$. Since $E_t^c = \{\|H(z_t, w_t)\| \geq I/\gamma_t\}$, by Markov inequality,

$$\mathbb{E}[\|H(z_t, w_t)\|\mathbf{1}_{E_t^c} \,|\, \mathcal{F}_t] \leq \mathbb{E}[\|H(z_t, w_t)\|^2/(I/\gamma_t) \,|\, \mathcal{F}_t] \leq \gamma_t A^2/I. \tag{S10}$$

Then from (S8),

$$\mathbb{E}[C(w_{t+1})\mathbf{1}_{E_t} \,|\, \mathcal{F}_t] \leq C(w_t) + \gamma_t^2 A^2 k - \gamma_t\|\nabla C(w_t)\|^2 \tag{S11}$$

with $k = k_2 + k_1/I$. Let

$$N_t = C(w_t)\mathbf{1}_{E_{t-1}} + A^2 k\sum_{s \geq t}\gamma_s^2 - \sum_{s < t}C(w_s)\mathbf{1}_{E_{s-1}^c}.$$

Since $\sum_{s < t+1}C(w_s)\mathbf{1}_{E_{s-1}^c}$ is $\mathcal{F}_t$-measurable, then from (S11),

$$\mathbb{E}(N_{t+1} \,|\, \mathcal{F}_t) \leq N_t - \gamma_t\|\nabla C(w_t)\|^2. \tag{S12}$$

Therefore, $N_t$ is a supermartingale. Let

$$\xi = \sum_{s \geq 0}C(w_s)\mathbf{1}_{E_{s-1}^c}.$$

Let $k_0 = \sup_{w \in K} C(w)$. Then $|N_t| \leq k_0 + A^2 k\sum_{s \geq 0}\gamma_s^2 + \xi$. On the other hand, by Fubini's theorem followed by Markov inequality,

$$\mathbb{E}\xi \leq k_0\sum_{s \geq 0}\mathbb{P}\{\|H(z_s, w_s)\| \geq I/\gamma_t\} \leq k_0\sum_{s \geq 0}\frac{\gamma_s^2}{I^2}\sup_{w \in K}\mathbb{E}_z\|H(z, w)\|^2 < \infty.$$

Then $N_t$ is uniformly integrable, i.e., $\sup_t \mathbb{E}(|N_t|\mathbf{1}_{|N_t \geq c|}) \to 0$ as $c \to \infty$. Then by martingale convergence theorem, $N_t$ converges a.s. Moreover, from the display and Borel-Cantelli lemma, all but a finite number of $\mathbf{1}_{E_t^c}$ are 0, a.s. As a result, it then follows that $C(w_t)$ converges a.s.

To show that $\nabla C(w_t) \to 0$ a.s., by Doob's decomposition, $N_t = M_t - Z_t$, where $M_t$ is a martingale and $Z_t$ is increasing and $\mathcal{F}_{t-1}$-measurable with $Z_0 = 0$, and both $M_t$ and $Z_t$ are uniformly integrable, giving $Z_t \uparrow Z \geq 0$ with $\mathbb{E}Z < \infty$. Put $p = \|\nabla C\|^2$. From (S12),

$$\sum_t \gamma_t p(w_t) \leq \sum_t (Z_{t+1} - Z_t) = Z < \infty \quad \text{a.s.}$$

Then, by $\sum \gamma_t = \infty$, to show $\nabla C(w_t) \to 0$, it suffices to show that $p(w_t)$ converges a.s.

Similar to (S7),

$$p(w_{t+1})\mathbf{1}_{E_t} - p(w_t) \leq -2\gamma_t \langle \nabla C(w_t), (\nabla^2 C(w_t))H(z_t, w_t)\rangle \mathbf{1}_{E_t} + \gamma_t^2 \|H(z_t, w_t)\|^2 k_4,$$

where $k_4$ is an upper bound on $\|\nabla^2 p\|$ on $K_0$. For ease of notation, write $\langle \nabla C, (\nabla^2 C)H\rangle_t$ for $\langle \nabla C(w_t), (\nabla^2 C(w_t))H(z_t, w_t)\rangle$, $\|H\|_t$ for $\|H(z_t, w_t)\|$, and so on. Then

$$\mathbb{E}(p(w_{t+1})\mathbf{1}_{E_t} - p(w_t) \,|\, \mathcal{F}_t) \leq -2\gamma_t \mathbb{E}(\langle \nabla C, (\nabla^2 C)H\rangle_t \mathbf{1}_{E_t} \,|\, \mathcal{F}_t) + \gamma_t^2 k_4 \mathbb{E}(\|H\|_t^2 \,|\, \mathcal{F}_t)$$

$$\leq -2\gamma_t \mathbb{E}(\langle \nabla C, (\nabla^2 C)H\rangle_t \mathbf{1}_{E_t} \,|\, \mathcal{F}_t) + \gamma_t^2 k_4 A^2.$$

On the other hand, recalling that $k_1 = \sup_K \|\nabla C\|$ and $k_2 = \sup_{K_0} \|\nabla^2 C\|$,

$$|\mathbb{E}(\langle \nabla C, (\nabla^2 C)H\rangle_t \mathbf{1}_{E_t} \,|\, \mathcal{F}_t)| \leq |\mathbb{E}(\langle \nabla C, (\nabla^2 C)H\rangle_t \,|\, \mathcal{F}_t)| + |\mathbb{E}(\langle \nabla C, (\nabla^2 C)H\rangle_t \mathbf{1}_{E_t^c} \,|\, \mathcal{F}_t)|$$

$$\leq |\langle \nabla C, (\nabla^2 C)\nabla C\rangle_t| + \|\nabla C\|_t \cdot \mathbb{E}(\|(\nabla^2 C)H\|_t \mathbf{1}_{E_t^c} \,|\, \mathcal{F}_t)$$

$$\leq k_2 \|\nabla C\|_t^2 + k_2 \|\nabla C\|_t \cdot \mathbb{E}(\|H\|_t \mathbf{1}_{E_t^c} \,|\, \mathcal{F})$$

$$\leq k_2 \|\nabla C\|_t^2 + k_3 \|\nabla C\|_t \cdot \mathbb{E}(\|H\|_t \mathbf{1}_{E_t^c} \,|\, \mathcal{F})$$

$$\leq k_2 p(w_t) + (k_1 k_2/I)A^2 \gamma_t.$$

where the last line follows from (S10). Combining the above two displays,

$$\mathbb{E}(p(w_{t+1})\mathbf{1}_{E_t} - p(w_t)\mathbf{1}_{E_{t-1}} \,|\, \mathcal{F}_t) \leq q(w_t) := p(w_t)\mathbf{1}_{E_{t-1}^c} + 2k_2 \gamma_t p(w_t) + k_* \gamma_t^2 A^2, \quad \text{(S13)}$$

where $k_* = k_4 + 2k_1 k_2/I$. From the above proof, $\sum \mathbb{E}q(w_t) < \infty$. Then by a similar argument for the convergence of $C(w_t)$ based on submartingale convergence, $p(w_t)$ converges a.s. $\qquad \square$

*Proof of Theorem 6.* The proof builds upon the one for Theorem 2 (Bonnabel, 2013). There are constants $\varrho > 0$ and $0 < I_0 \leq I$, such that $d(R_w(v), \exp_w(v)) \leq \varrho\|v\|^2$ for all $w \in K$ and $v \in \mathcal{T}_w\mathcal{M}$ with $\|v\| \leq I_0$. Without loss of generality, suppose $\varrho > 1$ and $I_0 = I < 1/\varrho$, for otherwise we can decrease $I$.

Define the same events $E_t$ as in (S6) and let constants $k_0, \ldots, k_4$ be defined as in the proof of Proposition S1. Let $w_t^* = \exp_{w_t}(-\gamma_t H(z_t, w_t))$. Then

$$d(w_{t+1}^*, w_{t+1})\mathbf{1}_{E_t} \leq \gamma_t^2 \varrho \|H(z_t, w_t)\|^2.$$

By assumption, $w_{t+1} \in K$. Then on the event $E_t$, $d(w_{t+1}^*, w_{t+1}) \leq \varrho I^2 < I$, so $w_{t+1} \in K_0$, where $K_0$ is defined in the proof of Proposition S1. Then

$$C(w_{t+1})\mathbf{1}_{E_t} - C(w_t) \leq C(w_{t+1}^*)\mathbf{1}_{E_t} - C(w_t) + |C(w_{t+1}) - C(w_{t+1}^*)|\mathbf{1}_{E_t}$$

$$\leq C(w_{t+1}^*)\mathbf{1}_{E_t} - C(w_t) + d(w_{t+1}, w_{t+1}^*)k_1\mathbf{1}_{E_1}$$

$$\leq C(w_{t+1}^*)\mathbf{1}_{E_t} - C(w_t) + \gamma_t^2 \|H(z_t, w_t)\|^2 \varrho k_1.$$

Then by (S9) and (S11),

$$\mathbb{E}(C(w_{t+1})\mathbf{1}_{E_t} - C(w_t) \,|\, \mathcal{F}_t) \leq \gamma_t^2 A^2 k - \gamma_t \|\nabla C(w_t)\|^2,$$

where $k = k_2 + k_1/I + \varrho k_1$. Then, following the same argument as in the proof of Proposition S1, $C(w_t)$ converges a.s. and $\sum_{t \geq 0} \gamma_t \mathbb{E}(\|\nabla C(w_t)\|^2) < \infty$, and to show $p(w_t) := \|\nabla C(w_t)\|^2 \to 0$, it suffices to show $p(w_t)$ converges a.s. We have

$$p(w_{t+1})\mathbf{1}_{E_t} - p(w_t) \leq |p(w_{t+1}) - p(w_{t+1}^*)|\mathbf{1}_{E_t} + p(w_{t+1}^*)\mathbf{1}_{E_t} - p(w_t)$$

$$\leq k_4 \gamma_t^2 \varrho \|H(z_t, w_t)\|^2 + p(w_{t+1}^*)\mathbf{1}_{E_t} - p(w_t).$$

Then by (S9) and (S13),

$$\mathbb{E}(p(w_{t+1})\mathbf{1}_{E_t} - p(w_t)\mathbf{1}_{E_{t-1}} \,|\, \mathcal{F}_t) \leq p(w_t)\mathbf{1}_{E_{t-1}^c} + 2k_2 \gamma_t p(w_t) + (2k_4 + 2k_1 k_2/I)\gamma_t^2 A^2.$$

Then, with the same argument following (S13), $p(w_t)$ converges a.s. $\qquad \square$

## B  GOODNESS-OF-FIT TESTS

In this section, we present several commonly used GoF tests. They are grouped into three classes: tests based on correlation (CB), tests based on empirical distribution function (EDF), and tests based on empirical characteristic function (ECF). The CB GoF tests were first covered in Section 3. The latter two are covered here.

*Empirical Distribution Function (EDF) Tests:* UVN EDF tests are based on the discrepancy between the empirical and hypothesized distribution functions (D'Agostino, 2017), encompassing two broad classes: supremum tests, and quadratic tests. Kolmogorov–Smirnov (KS) is a supremum test, measuring the largest absolute distance. Two popular quadratic tests are Cramér–von Mises (CVM), which measures the integrated quadratic deviation weighted by a function $\Psi$, and Anderson–Darling (AD) (Anderson et al., 1952), which gives higher weight to distribution tails.

*Empirical Characteristic Function (ECF) Tests:* ECF tests are based on the weighted integral of the difference between the ECF and its pointwise limit, including Epps–Pulley (EP) for UVN (Epps & Pulley, 1983) and MVN (Baringhaus & Henze, 1988) and the Henze–Zirkler (HZ) test, a generalization of EP (Henze & Zirkler, 1990).

For consistency, $T_*$ and $d_*$ represent respectively the test statistic and corresponding statistical distance for each test. We will used $F_X$ to denote both the law of $X$ and its cumulative distribution function. For a random sample $X_1, X_2, \ldots X_m$, denote its sample mean, sample variance, and empirical cumulative distribution function by $\bar{X}$, $S_m$, and $\hat{F}_{X,m}$, respectively. If the $X_i$'s are univariate, we further sort them into order statistics $X_{(1)} \leq X_{(2)} \leq \cdots X_{(m)}$.

### B.1  CB CLASS: SHAPIRO–WILK, SHAPIRO–FRANCIA

Let $X_1, \ldots, X_m$ be univairate and i.i.d. $\sim F_X$. To test $\mathcal{H}_0 : F_X \in \mathcal{G}$, where $\mathcal{G}$ is the class of normal distributions, the Shapiro–Wilk (SW) test statistic is defined as

$$T_{SW} = \frac{\left(\sum_{i=1}^m a_i X_{(i)}\right)^2}{\sum_{i=1}^m \left(X_i - \bar{X}\right)^2},$$

where $\mathbf{a} = (a_1, a_2, \ldots, a_m)^T$ is obtained via

$$\mathbf{a} = \frac{\mathbf{M}^{-1}\mathbf{c}}{\|\mathbf{M}^{-1}\mathbf{c}\|}$$

with $\mathbf{c} = (c_1, \ldots, c_m)^T$ and $\mathbf{M}$ being the mean vector and covariance matrix, respectively, of the order statistics of $m$ independent $N(0, 1)$ random variables. The corresponding $L_2$-Wasserstein distance is

$$d_{W_2}^2(\hat{F}_{X,m}, \mathcal{G}) = \inf_{a,\sigma^2} d_{W_2}^2(\hat{F}_{X,m}, N(a, \sigma^2)) = S_m^2 - \left(\int_0^1 \hat{F}_{X,m}^{-1}(t)\Phi^{-1}(t)\,\mathrm{d}t\right)^2$$

with $\Phi(\cdot)$ being the distribution function of standard normal (del Barrio et al., 1999). Following the same notations for SW test, the Shapiro–Francis (SF) test on normality (Shapiro & Francia, 1972) is defined as

$$T_{SF} = \frac{\left(\sum_{i=1}^m b_i X_{(i)}\right)^2}{\sum_{i=1}^m \left(X_i - \bar{X}\right)^2},$$

where $\mathbf{b} = (b_1, b_2, \ldots, b_m)^T$ is obtained via

$$\mathbf{b} = \frac{\mathbf{c}}{\|\mathbf{c}\|}.$$

### B.2  EDF CLASS: CRAMER–VON MISES, KOLMOGOROV–SMIRNOV

Let $X_1, \ldots, X_m$ be univariate and i.i.d. $\sim F_X$. Let $F$ be a specified univariate distribution and suppose we whish to test $\mathcal{H}_0 : F_X = F$. The Cramer–Von Mises (CVM) test statistic is defined as

$$T_{CVM} = \frac{1}{12m} + \sum_{i=1}^m \left(F(X_{(i)}) - \frac{2i-1}{2m}\right)^2.$$

The CVM test statistic corresponds to the statistical distance

$$d_{CVM}(\hat{F}_{X,m}, F) = \int \left(\hat{F}_{X,m} - F\right)^2 \boldsymbol{\Psi}(F)\, \mathrm{d}F,$$

where $\Psi$ is a weight function. Note that for $T_{CVM}$, $\Psi \equiv 1$. On the other hand, the Kolmogorov–Smirnov (KS) test statistic and related statistical distance are defined respectively as

$$T_{KS} = \max_{1 \le i \le m} \left\{ F(X_{(i)}) - \frac{i-1}{m}, \frac{i}{m} - F(X_{(i)}) \right\}, \quad d_{KS}(\hat{F}_{X,m}, F) = \sup |\hat{F}_{X,m} - F|.$$

### B.3 ECF CLASS: HENZE–ZIRKLER

Suppose $X_i$ are $k$-dimensional and i.i.d. $\sim \mathbb{P}_X$. Following the similar notation in (Henze & Zirkler, 1990), define the scaled residuals $Y_j = S_m^{-1/2}(X_j - \bar{X}), j = 1, \ldots, m$. Let

$$\phi_m(\mathbf{t}) = \frac{1}{m} \sum_{j=1}^{m} \exp(i\mathbf{t}'Y_j)$$

denote the empirical characteristic function of $Y_j$. To test $\mathcal{H}_0 : F_X \in \mathcal{G}$, the Henze–Zirkler (HZ) test statistic is defined as

$$T_{HZ} = m\left(4I\{S_m \text{ is singular}\} + D_{m,\beta}I\{S_m \text{ is nonsingular}\}\right),$$

where $\beta > 0$ is a parameter,

$$D_{m,\beta} = \int_{\mathbb{R}^k} \left| \phi_m(\mathbf{t}) - \exp\left(-\frac{1}{2}\|\mathbf{t}\|^2\right) \right|^2 \psi_\beta(\mathbf{t})\, \mathrm{d}\mathbf{t}, \quad \psi_\beta(\mathbf{t}) = (2\pi\beta^2)^{-k/2} \exp\left(-\frac{\|\mathbf{t}\|^2}{2\beta^2}\right).$$

After some simplification Korkmaz et al. (2014), we have

$$D_{m,\beta} = \frac{1}{m^2} \sum_{i,j=1}^{m} e^{-\beta^2 \|Y_i - Y_j\|^2/2} - \frac{2}{(1+\beta)^{k/2} m} \sum_{i=1}^{m} e^{-\beta^2 \|Y_i\|^2/[2(1+\beta^2)]} + \frac{1}{(1+2\beta^2)^{k/2}}.$$

The optimal choice for $\beta$ is proposed to be

$$\frac{1}{\sqrt{2}} \left[ \frac{m(2k+1)}{4} \right]^{1/(k+4)}.$$

Denote the Fourier transform of a probability measure $\mu$ on $\mathbb{R}^k$ by $\tilde{\mu}(\mathbf{t}) = \int_{\mathbb{R}^k} e^{i\mathbf{t}'\mathbf{x}} \mu(\mathrm{d}\mathbf{x})$. Then

$$D_{m,\beta} = \int_{\mathbb{R}^k} |\tilde{F}_{Y,m}(\mathbf{t}) - \tilde{G}(\mathbf{t})|^2 \psi_\beta(\mathbf{t})\, \mathrm{d}\mathbf{t},$$

where $G$ denotes the $k$-variate standard normal distribution. Thus, HZ statistic corresponds to the following statistical distance

$$d_{HZ}(\mu, \nu) = \int_{\mathbb{R}^k} |\tilde{\mu}(\mathbf{t}) - \tilde{\nu}(\mathbf{t})|^2 \psi_\beta(\mathbf{t})\, \mathrm{d}\mathbf{t}.$$

### B.4 A COMPARISON BETWEEN THE ASSOCIATED STATISTICAL DISTANCES

**Proposition S2.** *Let $X$, $Y$ denote two $k$-variate random variables. Then, (a) $d_{HZ}(F_X, F_Y) \le C_1 d_{W_2}(F_X, F_Y)$, (b) $d_{KS}(F_X, F_Y) \le C_2\sqrt{d_{W_1}(F_X, F_Y)}$, (c) if $k = 1$ and the weight function $\Psi(\cdot)$ in the CVM test is bounded, then $d_{CVM}(F_X, F_Y) \le C_3 d_{KS}(F_X, F_Y)^2$. In these inequalities the constants $C_1, C_2$ may depend on $k$, while $C_3$ may depend on $\Psi(\cdot)$.*

*Proof.* (a) Put $\mu = F_X$ and $\nu = F_Y$. For any $\omega \in \Pi(\mu, \nu)$, by Cauchy–Schwarz inequality and Fubini's theorem

$$d_{HZ}(\mu, \nu)^2 = \int_{\mathbb{R}^d} \left| \int_{\mathbb{R}^d} e^{i\mathbf{t}'\mathbf{x}} \mu(\mathrm{d}\mathbf{x}) - \int_{\mathbb{R}^d} e^{i\mathbf{t}'\mathbf{y}} \nu(\mathrm{d}\mathbf{y}) \right|^2 \phi(\mathbf{t}) \, \mathrm{d}\mathbf{t}$$

$$= \int_{\mathbb{R}^d} \left| \int_{\mathbb{R}^d \times \mathbb{R}^d} (e^{i\mathbf{t}'\mathbf{x}} - e^{i\mathbf{t}'\mathbf{y}}) \omega(\mathrm{d}\mathbf{x}, \mathrm{d}\mathbf{y}) \right|^2 \phi(\mathbf{t}) \, \mathrm{d}\mathbf{t}$$

$$\leq \int_{\mathbb{R}^d} \left\{ \int_{\mathbb{R}^d \times \mathbb{R}^d} |e^{i\mathbf{t}'\mathbf{x}} - e^{i\mathbf{t}'\mathbf{y}}|^2 \omega(\mathrm{d}\mathbf{x}, \mathrm{d}\mathbf{y}) \right\} \phi(\mathbf{t}) \, \mathrm{d}\mathbf{t}$$

$$= \int_{\mathbb{R}^d \times \mathbb{R}^d} \left\{ \int_{\mathbb{R}^d} |e^{i\mathbf{t}'\mathbf{x} - i\mathbf{t}'\mathbf{y}}|^2 \phi(\mathbf{t}) \, \mathrm{d}\mathbf{t} \right\} \omega(\mathrm{d}\mathbf{x}, \mathrm{d}\mathbf{y}).$$

Using $|e^{ia} - e^{ib}|^2 \leq (a - b)^2$ for $a, b \in \mathbb{R}$,

$$d_{HZ}(\mu, \nu)^2 \leq \int_{\mathbb{R}^d \times \mathbb{R}^d} \left\{ \int_{\mathbb{R}^d} |\mathbf{t}'\mathbf{x} - \mathbf{t}'\mathbf{y}|^2 \phi(\mathbf{t}) \, \mathrm{d}\mathbf{t} \right\} \omega(\mathrm{d}\mathbf{x}, \mathrm{d}\mathbf{y})$$

$$\leq \int_{\mathbb{R}^d \times \mathbb{R}^d} \left\{ \int_{\mathbb{R}^d} \|\mathbf{t}\|^2 \|\mathbf{x} - \mathbf{y}\|^2 \phi(\mathbf{t}) \, \mathrm{d}\mathbf{t} \right\} \omega(\mathrm{d}\mathbf{x}, \mathrm{d}\mathbf{y})$$

$$= \int_{\mathbb{R}^d} \|\mathbf{t}\|^2 \phi(\mathbf{t}) \, \mathrm{d}\mathbf{t} \cdot \int_{\mathbb{R}^d \times \mathbb{R}^d} \|\mathbf{x} - \mathbf{y}\|^2 \omega(\mathrm{d}\mathbf{x}, \mathrm{d}\mathbf{y})$$

$$= C \int_{\mathbb{R}^d \times \mathbb{R}^d} \|\mathbf{x} - \mathbf{y}\|^2 \omega(\mathrm{d}\mathbf{x}, \mathrm{d}\mathbf{y})$$

with $C = \int_{\mathbb{R}^d} \|\mathbf{t}\|^2 \phi(\mathbf{t}) \, \mathrm{d}\mathbf{t} < \infty$. Since the above inequality holds for all $\omega \in \Pi(\mu, \nu)$, then $d_{HZ}(\mu, \nu) \leq \sqrt{C} d_{W_2}(\mu, \nu)$. Let $C_1 = \sqrt{C}$. Then (a) follows.

(b) The connection between Kolmogorov–Smirnov distance and $L_1$-Wasserstain distance under both of the univariate and multivariate cases are well studied, detailed proof can be found in Corollary 3.1 (Koike, 2019), Proposition 1.2 (Ross, 2011) etc.

(c) Letting $C_3 = \sup \Psi$,

$$d_{CVM}(F_X, F_Y) = \int (F_X - F_Y)^2 \Psi(F_Y) \, \mathrm{d}F_Y \leq C_3 \int |F_X - F_Y|^2 \, \mathrm{d}F_Y$$

$$\leq C_3 \int (\sup |F_X - F_Y|)^2 \, \mathrm{d}F_Y = C_3 d_{KS}(F_X, F_Y)^2. \qquad \square$$

## C   MODEL ARCHITECTURE

The architecture for the encoder and decoder of CelebA and MNIST is based in the WAE (Tolstikhin et al., 2017). The discriminator for the WAE-GAN followed Tolstikhin et al. (2017). The architecture for CIFAR10 is based on Lippe (2022). When modeling a specific dataset, the specified encoder and decoder components are the same for all models.

## D   TRAINING DETAILS

### D.1   TEST STATISTIC PROJECTIONS

Once a batch has been encoded, it is projected from 64D to 1D in order to be tested. Instead of randomly sampling directions from the unit sphere to implement the projections, we sample an orthonormal basis and project down each direction, calculating the test statistic along each. This is used in two ways: 1) selecting the most pessimistic direction to optimize, or 2) computing the average direction to optimize.

For example, the SW test statistic value should be large to fail to reject normality. Selecting the direction associated with the smallest statistic value can be used as a new statistic to optimize.

Table S1: Architectures used for modeleing the CelebA, MNIST and Cifar10 datasets. Conv $128 \times 4 \times 4$ represents a convolution with 128 filters which are $4 \times 4$, FC stands for fully connected layer, ReLU for the rectified linear units, and ConvT for convolution transpose.

| Dataset | Optimizer | | Architecture |
|---------|-----------|--------|--------------|
| CelebA | Adam | Input | 64x64x3 |
| | | Encoder | Conv 128x4x4 (stride 2), 256x4x4 (stride 2), 512x4x4 (stride 2), 1024x4x4 (stride 2), FC 16384, 256. ReLU activation |
| | RSGD | Input | 256 |
| | | Stiefel | FC 64 Orthogonal initialization |
| | Adam | Input | 64 |
| | | Decoder | FC 65536, ConvT 512x4x4 (stride 2), ConvT 256x4x4 (stride 2), ConvT 128x4x4 (stride 2), ConvT 3x1x1 (stride 1), Sigmoid activation. |
| MNIST | Adam | Input | 28x28x1 |
| | | Encoder | Conv 128x4x4 (stride 2), 256x4x4 (stride 2), 512x4x4 (stride 2), 1024x4x4 (stride 2), FC 16384, 256. ReLU activation |
| | RSGD | Input | 256 |
| | | Stiefel | FC 10 Orthogonal initialization |
| | Adam | Input | 10 |
| | | Decoder | FC 65536, ConvT 512x4x4 (stride 1), ConvT 256x4x4 (stride 1), ConvT 128x4x4 (stride 2), Sigmoid activation. |
| CIFAR10 | Adam | Input | 32x32x3 |
| | | Encoder | Conv 64x3x3 (stride 2, pad 1), Conv 64x3x3 (stride 1, pad 1) Conv 128x3x3 (stride 2, pad 1), Conv 128x3x3 (stride 1, pad 1) Conv 128x3x3 (stride 2, pad 1), FC 2048, 256, ReLU activation |
| | RSGD | Input | 256 |
| | | Stiefel | FC 64 Orthogonal initialization |
| | Adam | Input | 64 |
| | | Decoder | FC 2048, ConvT 128x3x3 (stride 2, pad 1, out pad 1) Conv 128x3x3 (stride 1, pad1), ConvT 64x3x3(stride 2, pad 1, out pad 1) Conv 64x3x3 (stride 1, pad 1), ConvT 3x3x3 (stride 2, pad 1, out pad 1), Sigmoid activation |

Similarly, for tests that fail to reject for small values, selecting the direction associated with the *largest* value can be used. We refer to both scenarios as selecting the most pessimistic direction. Alternatively, the mean of *all* the statistics could be used as a new statistic.

### D.2 EMPIRICAL DISTRIBUTIONS

The distribution of a test statistic must be known analytically or estimable in order to compute p-values. There are a few options for calculating the empirical distribution of a test statistic when using GoF tests with projections. The first possibility is to randomly sample from the unit sphere, project the multivariate data, and then use the corresponding distribution to determine the p-value. However, a single projection may not be particularly informative, and early testing indicated this method lead to slower convergence. Another option is to project along multiple directions and calculate the statistics of these directions. It is then possible to create a *new* statistic from these, for example the average, minimum, or maximum. Unfortunately, this method precludes using the original test statistic distribution or calculating p-values. To remedy this, we create an empirical distribution of this new statistic by repeatedly sampling; an empirical p-value are produced from the test statistic samples.

### D.3 MUTUAL INFORMATION

The mutual information is estimated after the model has been trained by sampling from $\mathcal{N}_{64}(\hat{\boldsymbol{\mu}}, \hat{\boldsymbol{\Sigma}})$ where $(\hat{\boldsymbol{\mu}}, \hat{\boldsymbol{\Sigma}})$ are estimated either during or after training. GoFAE tracks these statistics during training. These samples are decoded, and then encoded. Assuming the encoded data is Gaussian, mutual information may be calculated as $I(\mathbf{Z}; \tilde{\mathbf{Z}}) = \frac{1}{2} \log \frac{|\boldsymbol{\Sigma}_Z| \times |\boldsymbol{\Sigma}_{\tilde{z}}|}{|\boldsymbol{\Sigma}_{z, \tilde{z}}|}$.

### D.4 Further Details on Model Selection

We selected $\lambda$ with grid-search using a training and validation set and an array of $\lambda$ values. Each model was evaluated for p-value uniformity using Algorithm 1. Note that Algorithm 1 takes multiple minibatch (local) GoF p-values and produces a *single* Kolmogorov-Smirnov test statistic and p-value pair for uniformity (a single blue dot in Figs. 5b-5c).

Since univariate GoF tests require projection, there are two sources of randomness that come into play when producing GoF p-values from a data set: 1) shuffling minibatches (line 2 in Algorithm 1), and 2) random projections. Instead of selecting $\lambda$ based on a single KS uniformity p-value, Algorithm 1 was run 30 times. The average of these 30 uniformity p-values was computed, and compared against a pre-specified threshold, which is 0.05 by default. If the average is larger than the threshold, then $\lambda$ is a possible candidate.

There is a region where a variety of $\lambda$ values satisfy the threshold condition of 0.05 (Figs. 5b-5c). Since the loss function, Equation 2, should also have small reconstruction error, the smallest $\lambda$ for which the corresponding average of 30 KS p-values is larger than 0.05 is the chosen hyperparameter and used in the final model. Pseudo-code for the GoFAE pipeline can be seen in Algorithm S1.

---

**Algorithm S1** GoFAE Pipeline: Training Locally and Assessing with Higher Criticism

---

**Require:** Train/Val/Test Set, array of $\lambda$ values, repeats = $R$, threshold
 1: **for** Each $\lambda$ in array **do**
 2:     Use Algorithm 2 on Training Set
 3:     **for** r = 1:R **do**
 4:         Use Algorithm 1 on Training/Validation Set
 5:         Store $KS_{unif}$ p-value
 6:     Compute average $KS_{unif}$ p-value and store
 7: Select smallest $\lambda$ where average $KS_{unif}$ p-value > threshold

---

# E  Additional Experiments

### E.1 Details on Experiments

We trained GoFAE and competing methods on MNIST LeCun et al. (1998) and CelebA Liu et al. (2015). For fair comparisons, we adopted a common architecture for all methods for each dataset as described in Section C.

### E.1.1 MNIST

The MNIST dataset was re-scaled to the unit interval. Training proceeded for 50 epochs using mini-batches of size 128. We specified a 10-dimensional code layer. Competitor models include $\beta$-VAE Higgins et al. (2017), VAE Kingma & Welling (2013), WAE-GAN Tolstikhin et al. (2017), VAE with learned $\gamma$ Dai & Wipf (2019), 2-Stage VAE Dai & Wipf (2019), and AE Bengio et al. (2013). Both VAE and $\beta$-VAE had an initial learning rate of $1e-4$, while VAE with learnable $\gamma$ used $1e-3$.

For our models, Adam optimizer Kingma & Ba (2014) was used for the encoder (learning rate $3e-3$, $\beta = (0.5, 0.999)$) and decoder (learning rate $3e-3$, $\beta = (0.5, 0.999)$). Riemannian SGD with learning rate $5e-3$ was used to constrain $\Theta$ to the Stiefel manifold using a one cycle learning rate scheduler with max learning rate as $1e-3$. Singular value decomposition was used for retracting $\Theta$ back to $\mathcal{M}$ after a parameter update.

### E.1.2 CelebA

The CelebA dataset was pre-processed following the same procedure in Tolstikhin et al. (2017). First, a 140x140 center crop is taken and the image is resized to $64 \times 64$ resolution. For the VAE, the Adam optimizer was used with a learning rate of $1e-4$, and $\beta_1 = 0.5$ and $\beta_2 = 0.999$. The $\beta$-VAE used the Adam optimizer with a learning rate of $1e-4$, $\beta_1 = 0.5$, and $\beta_2 = 0.999$. The VAE with learnable $\gamma$ used the Adam optimizer with a learning rate of $1e-3$, $\beta_1 = 0.5$, and $\beta_2 = 0.999$. The

WAE-GAN also used the Adam optimizer where the encoder and decoder parameters had an initial learning rate of $3e-4$ and the discriminator was set to $1e-3$ with $\beta_1 = 0.5$, $\beta_2 = 0.999$. For the two-stage VAE, we used the trained VAE with learned gamma as stage one. A second VAE was trained using the $\mu, \sigma$ from the first stage as inputs, also with a trainable gamma. The architecture was a three layer dense net with ReLU activations, and the decoder was the inverse, with the last layer being linear.

For our method, which had two components for the encoder, as visualized in Figure (3a) in the main paper. The first component of the encoder used the Adam optimizer with learning rate set to $3e-3$ and $\beta_1 = 0.5$ and $\beta_2 = 0.999$. The decoder also used Adam, with the same learning rates and hyperparameters. The dense layer of the encoding processes is parameterized by $\Theta$, an element of the Steifel manifold $\mathcal{M}$, and is trained with Riemannian SGD with a learning rate of $5e-3$ while making use of the 1-cycle learning rate scheduler with a maximum learning rate of $1e-3$.

As our methods require projection, a $64 \times 64$ dimensional orthonormal basis is sampled and used to project the encoded data.

### E.1.3 CIFAR-10

CIFAR10 is the final dataset, and consists of 60 thousand small images. The training set contains 50 thousand with the remaining going to the test set. After setting a seed, the training set was split into a smaller training set, of 45 thousand, and a validation set with the rest. The architecture used follows from the Model Architecture section. A latent dimension size of $64$ is used. Models are evaluated on the average reconstruction quality. Here, the mean-squared error of randomly selected batches is computed over over a single pass through the test set, and then averaged. This is repeated 30 times, with the mean and standard deviation reported. FID scores for reconstructed and generated data are also computed.

The Adam optimizer was used for the encoder and decoder of all methods. The learning rate was set at $3e-4$, with $\beta_1 = 0.5$ and $\beta_2 = 0.999$. The discriminator for WAE-GAN is the same from the CelebA setup. All models are trained for 200 epochs, and reduce the learning rate when hitting a plateau (*ReduceLROnPlateau* in PyTorch). For this a minimum learning rate was set as $5e-5$, patience of 10, and a factor of $0.2$. For the two-stage VAE, the encoder consisted of 2 dense layers of 32 nodes each with ReLU activation, 64-dimension code layer, and the decoder contained 3 dense layers with ReLU activation except the final layer which was linear. The architecture was explored using the encoded validation set coming from stage 1 (the VAE with learnable gamma). This second stage VAE was trained for 50 epochs, with KL converging to zero quickly. Uniformity for the 2-Stage VAE was assessed using sampling from the test set which was encoded after stage 1 training was complete.

Similar to the models for MNIST and CelebA, the dense layer encoding for the GoFAE models, $\Theta$, is trained with Riemannian SGD. The learning rate is $5e-4$ and uses a 1-Cycle learning rate scheduler with a maximum learning rate of $1e-4$. Each encoded batch is projected onto a randomly sampled $64 \times 64$ orthonormal basis. The test statistic $T$ is applied to each direction and the final statistic (min, average, max) computed.

### E.2 MNIST

In the following sections, we visually assess the quality of GoFAE and competing models. For the GoFAE plots, a "good" regularization coefficient $\lambda > 0$ indicates that the model should (i) fail to reject $\mathcal{H}_0$ at an $\alpha$-level chosen a-priori (close to normal but not overly so), (ii) accurately reconstruct the input, and finally (iii) generate samples which are both qualitatively and quantitatively convincing.

In this section, we present performance of competing and GoFAE methods in Table S2. Reconstruction error, reconstruction FID, generation FID, and the mean (std) of the $KS_{unif}$ p-values are reported.

Additionally, we examine the effects of $\lambda$ with GoFAE-SW model in Figure S1. The GoF test pushes the transformation to become increasingly singular in the sense that the condition number of the covariance matrix becomes increasingly large (Figure S1). As $\lambda$ gets larger, the average KS uniformity statistic also increases, indicating $\mathbb{P}_Y$ appears to be indistinguishable from the Gaussian class. However, as $\lambda$ continues to increase, the condition number of the estimated covariance matrix also increases. Yet, for a small interval of $\lambda$, the mutual information has not completely diminished,

and the KS uniformity suggests $\mathbb{P}_Y$ is still indistinguishable from Gaussian. This visualization provides insights on how the class of Gaussians allows the model to adapt as needed during training.

Table S2: Evaluation of MNIST by MSE, FID scores (Heusel et al., 2017), and samples with p-values from higher criticism.

| Algorithm | MSE ↓ | FID Score ↓ | | Kolmogorov–Smirnov Uniformity Test | | | | |
|---|---|---|---|---|---|---|---|---|
| | | Gen. | Recon. | SW | SF | CVM | KS | EP |
| AE-Baseline | 7.66 | 62.29 | 7.39 | 0.0 (0.0) | 0.0 (0.0) | 0.0 (0.0) | 0.0 (0.0) | 0.0 (0.0) |
| VAE | 13.99 | 16.23 | 8.33 | 0.18 (0.18) | 0.42 (0.30) | 0.07 (0.12) | 0.14 (0.18) | 0.03 (0.06) |
| $\beta(2)$-VAE | 20.98 | 20.54 | 11.52 | 0.38 (0.31) | 0.29 (0.26) | 0.41 (0.29) | 0.45 (0.31) | 0.24 (0.25) |
| C-$\beta$-VAE | 20.94 | 19.21 | 11.38 | 0.48 (0.30) | 0.37 (0.28) | 0.29 (0.26) | 0.42 (0.31) | 0.19 (0.19) |
| WAE-GAN | 9.49 | 15.72 | **7.28** | 0.07 (0.12) | 0.49 (0.29) | 0.09 (0.15) | 0.11 (0.12) | 0.06 (0.12) |
| GoFAE-SW | 7.16 | 16.58 | 7.57 | 0.11 (0.16) | 0.48 (0.23) | 0.06 (0.12) | 0.17 (0.24) | 0.03 (0.05) |
| GoFAE-SF | **7.08** | 18.35 | 7.47 | 0.0 (0.0) | 0.26 (0.27) | 0.0 (0.0) | 0.0 (0.0) | 0.0 (0.0) |
| GoFAE-CVM | 7.67 | **15.56** | 7.71 | 0.14 (0.18) | 0.46 (0.28) | 0.14 (0.22) | 0.12 (0.21) | 0.05 (0.15) |
| GoFAE-EP | 7.54 | 15.85 | 7.84 | 0.25 (0.27) | 0.44 (0.28) | 0.12 (0.23) | 0.17 (0.20) | 0.05 (0.08) |

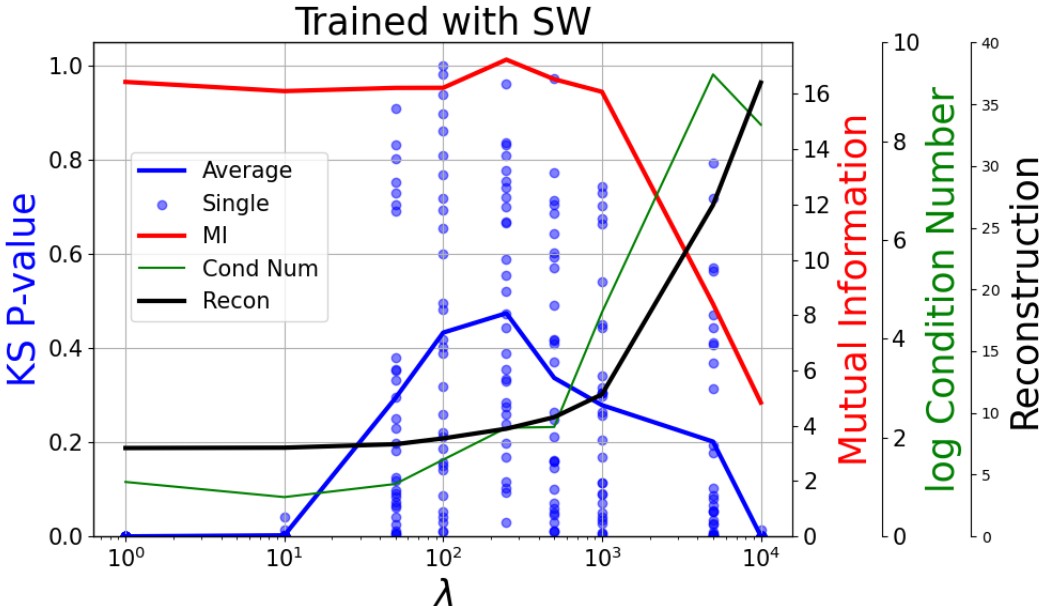

Figure S1: Effects of $\lambda$. We evaluated the uniformity KS test p-values (blue dots), average KS p-values (blue line), mutual information (red line), log condition number (green line) and reconstruction error (black line) under different $\lambda$ for GoFAE-SW.

### E.3    ADDITIONAL PLOTS

In this section, we include additional experimental results for GoFAE and comparison models.

### E.3.1    CELEBA

This section contains images produced from GoFAE and competing GAEs on the CelebA dataset with a latent space dimension of 64. Figure S2 depicts reconstructed faces. In Table 1 in the main paper, we evaluated reconstruction quality with mean-squared error and the Frechet-Inception Distance (FID). The visual quality of the images produced by the GoFAE models are consistent with the low MSE and reconstruction FID observed in Table 1. Figure S3 contains faces generated from the model. The GoFAE models again produced competitive FID scores, with GoFAE-SF producing the lowest (best) of the group. Tables S3 and S4 illustrate the effects of $\lambda$ on KS uniformity test, corresponding p-value, reconstruction error, and test statistics for GoFAE-SW, GoFAE-SF, GoFAE-CVM and GoFAE-KS, respectively.

There are several things to note from the Tables of figures demonstrating the effects of $\lambda$ on the training of GoFAE architectures (Table S3 and Table S4). The first row shows the mutual information and GoF p-value distribution computed from the test set as a function of $\log(\lambda)$. As $\lambda$ increases the models are producing $\mathbb{P}_Y$ that are appearing increasingly indistinguishable from normal. However, beyond a certain $\lambda$, the encoding no longer appears normal, GoFAE-SW clearly depicts this occurring in Table S3, row (a), left pannel. Row (b) tracks the average p-value produced on the test set while training with a particular GoF test statistic. Larger $\lambda$ can easily be seen to focus more on normality as the p-values tend to start at high levels and stay. The reconstruction error in row (c) converges quickly.

Of particular interest in the $\log(\lambda)$ vs KS p-value plots, is the behavior of the average KS p-value; after the average KS uniformity peaks, the mutual information falls rapidly. This clearly illustrates the relationship posited in the experiments section where the model prioritizes posterior matching.

Similar plots for MNIST can be seen in section E.3.2 with the quantitative information regarding reconstruction error, reconstruction FID and generation FID in Table S2. Once again, the GoFAE models produce overall lower MSE than competitor models, while maintaining competitive generation FID scores.

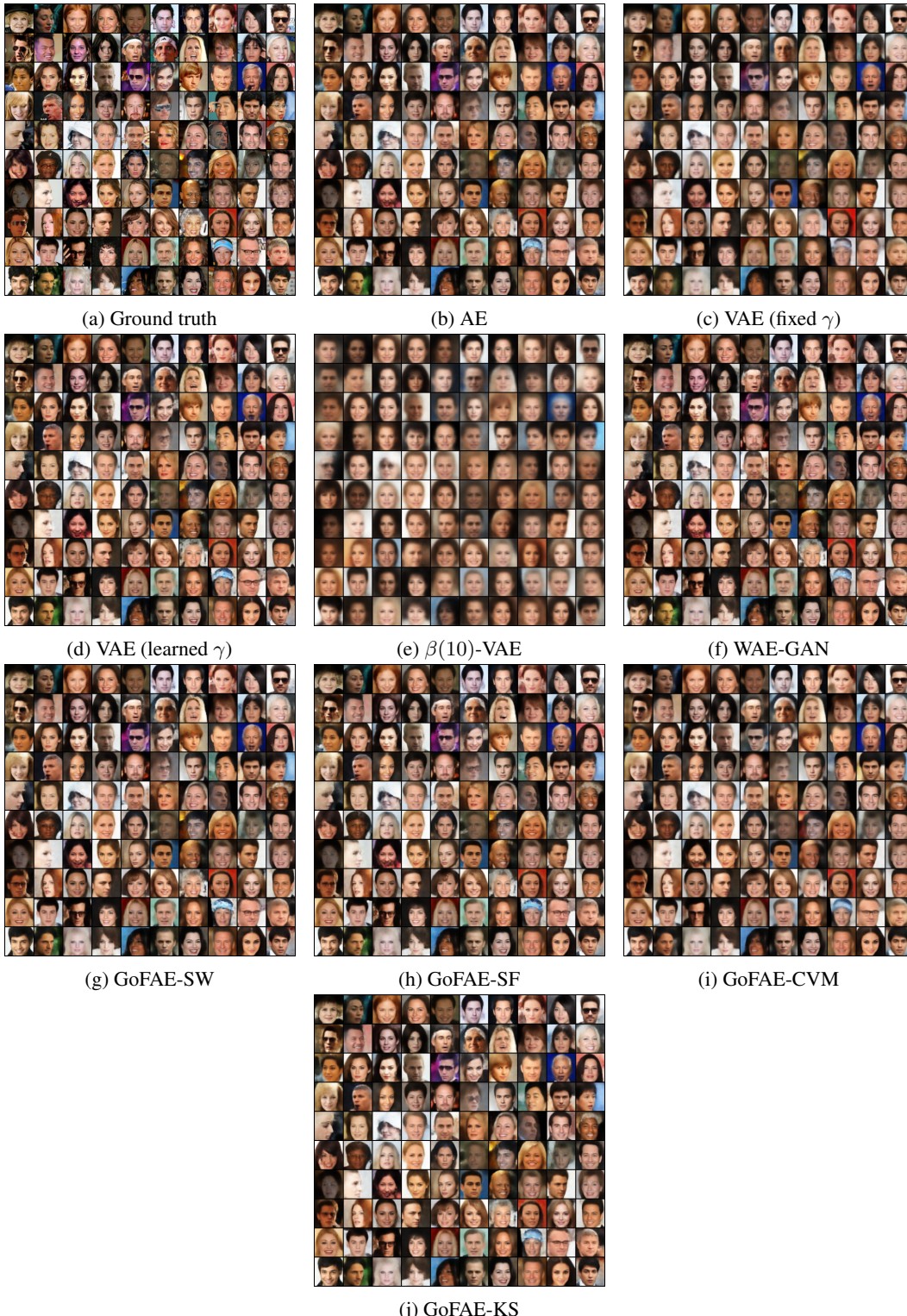

Figure S2: Comparison of reconstruction between competing methods: (b) AE, (c) VAE (fixed $\gamma$), (d) VAE (learned $\gamma$), (e) $\beta$-VAE ($\beta = 10$), (f) WAE-GAN, and our framework: (g) GoFAE-SW, (h) GoFAE-SF, (i) GoFAE-CVM, (j) GoFAE-KS with CelebA.

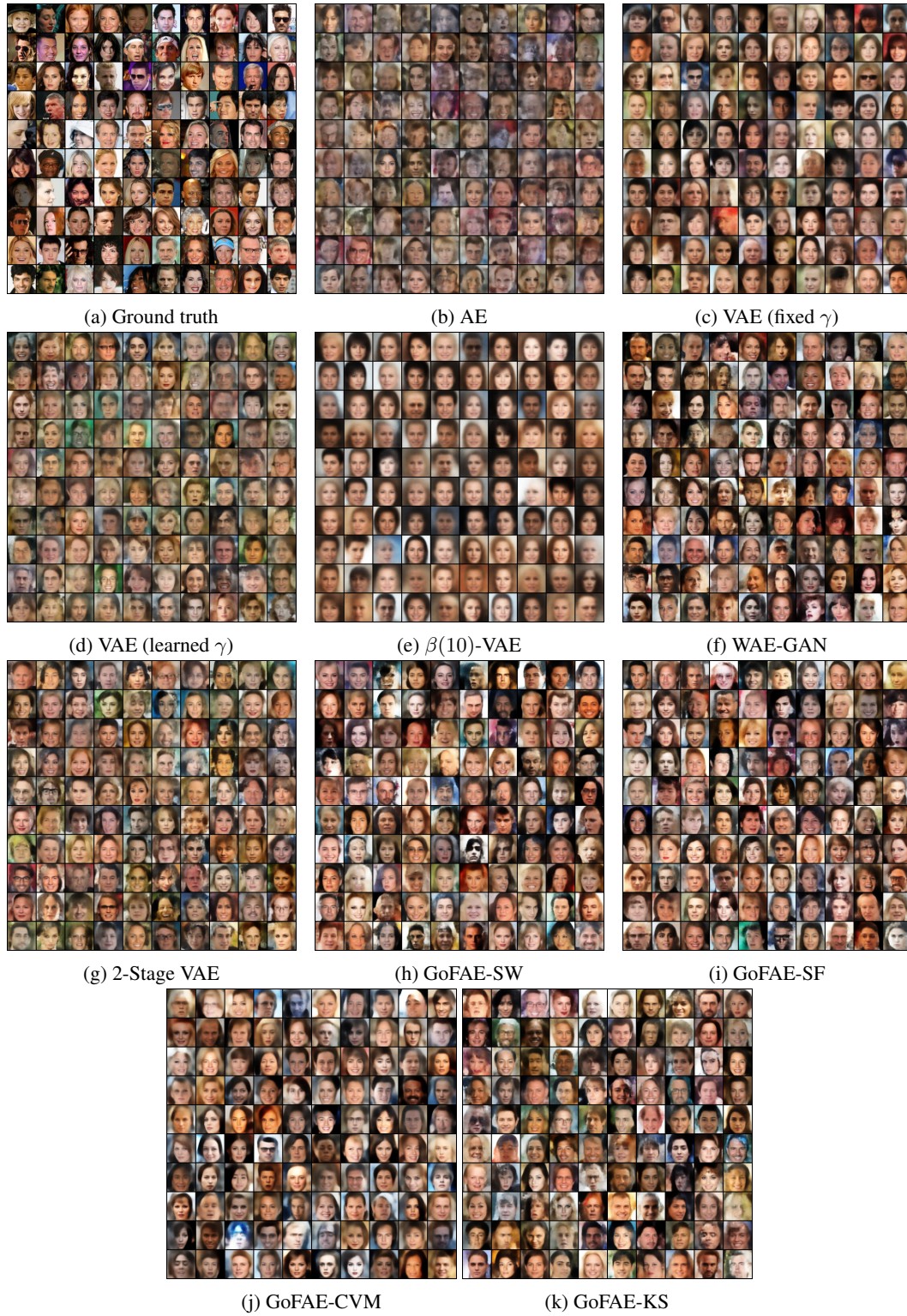

Figure S3: Comparison of generated samples between competing methods: (b) AE, (c) VAE (fixed $\gamma$), (d) VAE (learned $\gamma$), (e) $\beta$-VAE ($\beta = 10$), (f) WAE-GAN, (g) 2-Stage VAE, and our framework: (h) GoFAE-SW, (i) GoFAE-SF, (j) GoFAE-CVM, (k) GoFAE-KS with CelebA.

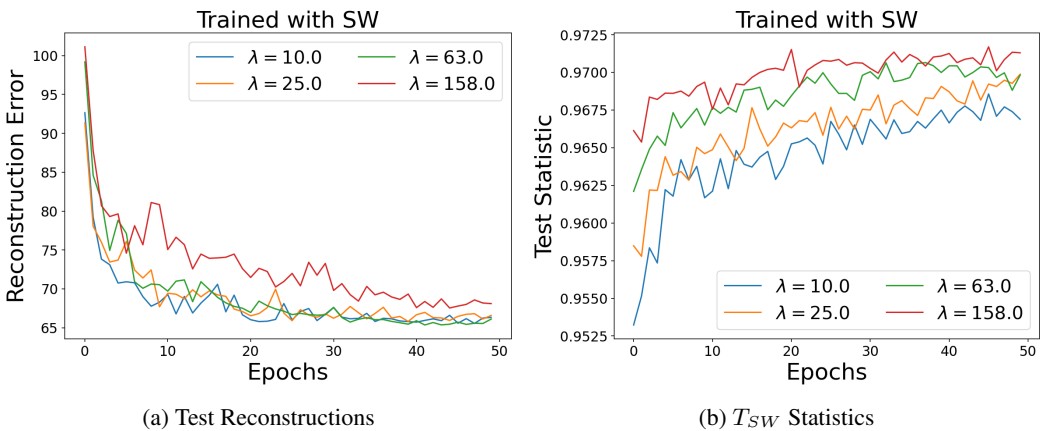

(a) Test Reconstructions

(b) $T_{SW}$ Statistics

Figure S4: Convergence of reconstruction error (a) and $T_{SW}$ (b) for multiple $\lambda$.

Table S3: The effects of $\lambda$ with GoFAE-SW (left) and GoFAE-SF (right) for CelebA. From top to bottom, we have: (a) the normality of GoFAE mini-batch encodings using 30 repetitions of Algorithm 1 for different $\lambda$. (b) p-value changing from epoch 1-50 under varying $\lambda$. (c) Reconstruction error on the testing set with different $\lambda$. (d) Test statistics value on the testing set with different $\lambda$.

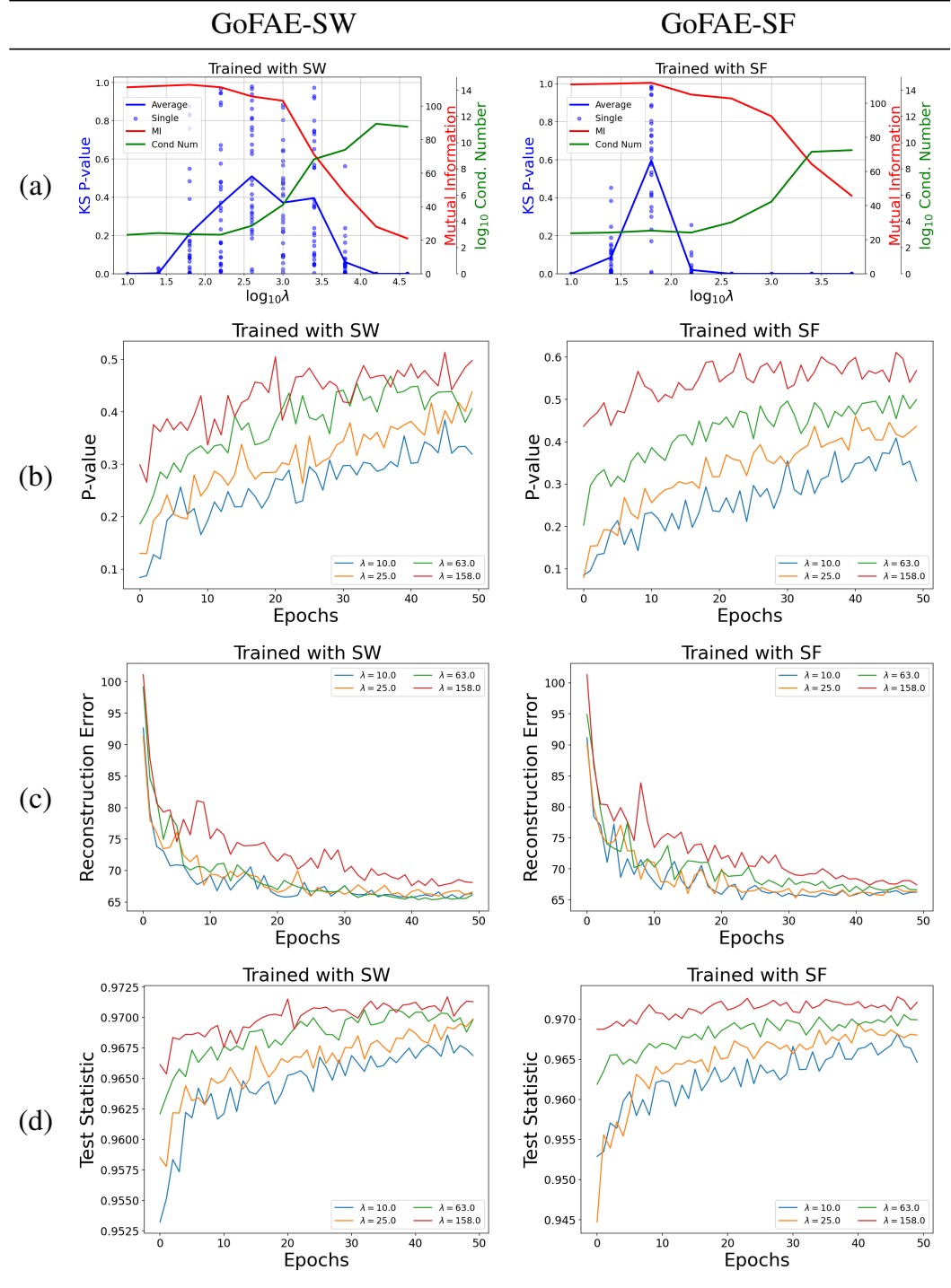

Table S4: The effects of $\lambda$ with GoFAE-CVM (left) and GoFAE-KS (right) for CelebA. From top to bottom, we have: (a) the normality of GoFAE mini-batch encodings using 30 repetitions of Algorithm 1 for different $\lambda$. (b) p-value changing from epoch 1-50 under varying $\lambda$. (c) Reconstruction error on the testing set with different $\lambda$. (d) Test statistics value on the testing set with different $\lambda$.

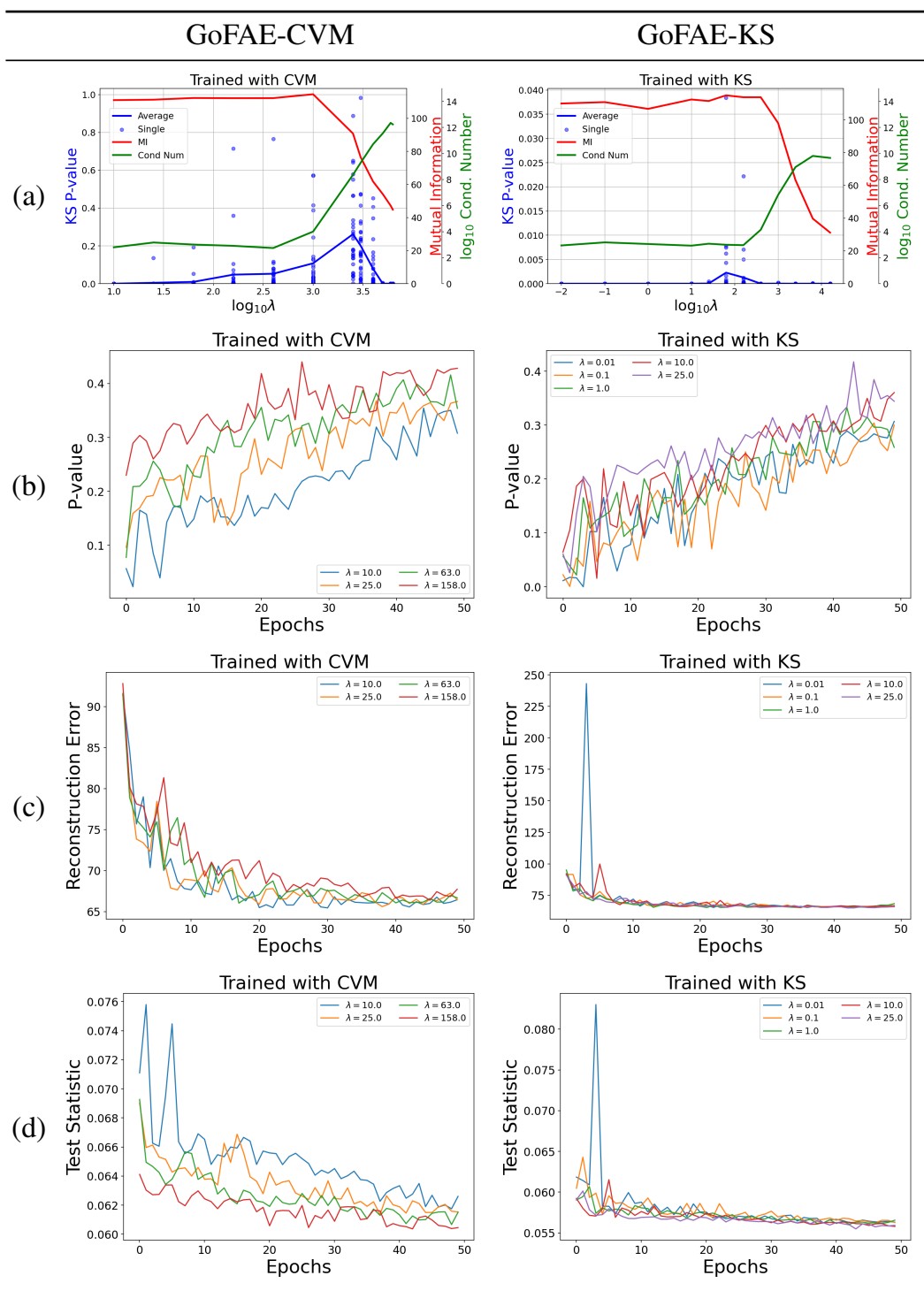

### E.3.2 MNIST

This section includes additional reconstructed and generated samples in Figure S5 and Figure S6. Table S5 and S6 demonstrate the effects of $\lambda$ on KS uniformity test, corresponding p-value, reconstruction error, and test statistics for GoFAE-SW, GoFAE-SF, GoFAE-CVM and GoFAE-KS, respectively. The experiments on MNIST reach the same conclusion as in the CelebA, that our GoFAE methods is competitive in both reconstruction and generation.

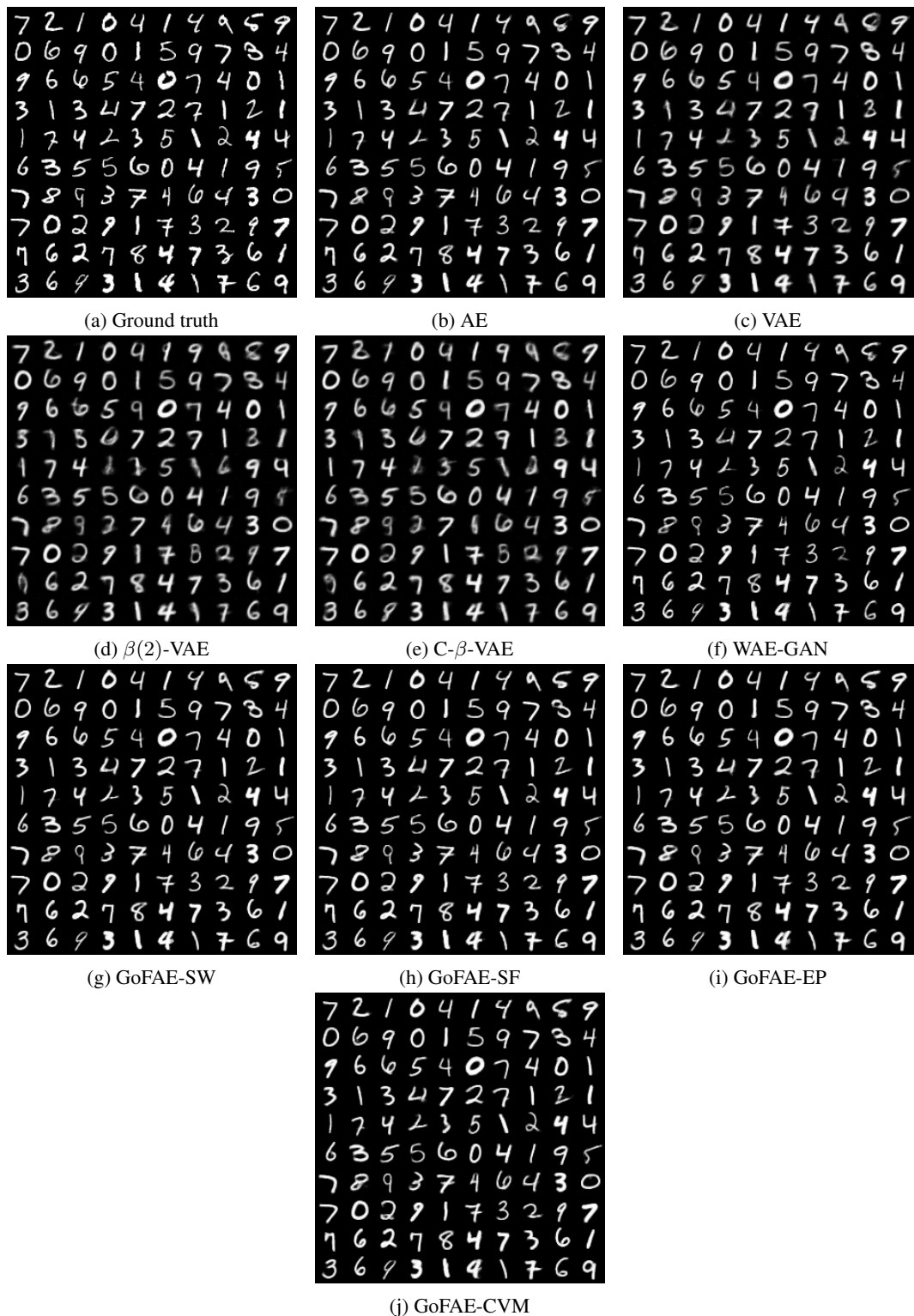

Figure S5: Comparison of reconstruction between competing methods (b) AE, (c) VAE, (d) $\beta$-VAE ($\beta = 2$), (e) C-$\beta$-VAE, (f) WAE-GAN, and our framework: (g) GoFAE-SW, (h) GoFAE-SF, (i) GoFAE-EP, (j) GoFAE-CVM with MNIST.

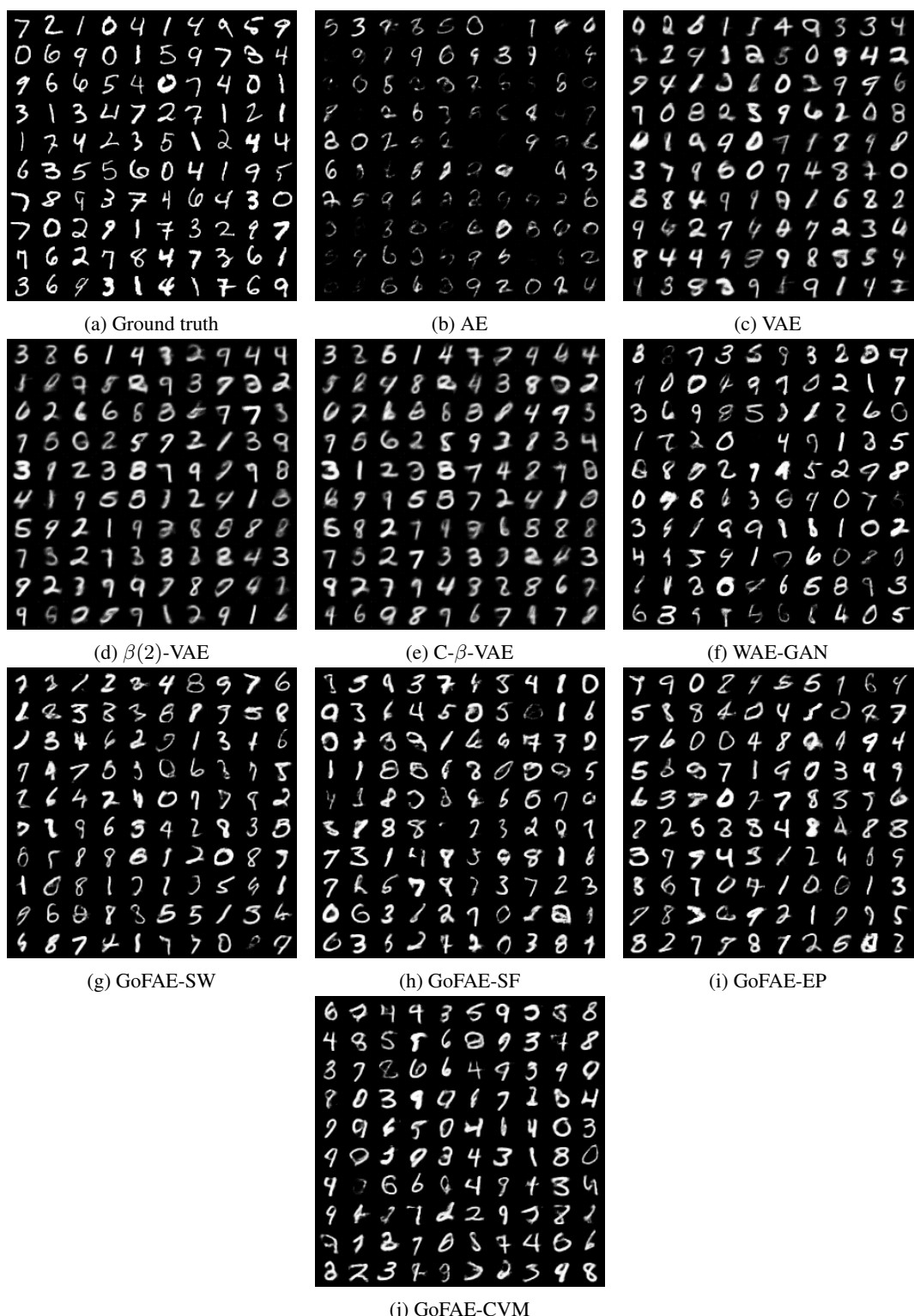

Figure S6: Comparison of generated samples between competing methods (b) AE, (c) VAE, (d) $\beta$-VAE ($\beta = 2$), (e) C-$\beta$-VAE, (f) WAE-GAN, and our framework: (g) GoFAE-SW, (h) GoFAE-SF, (i) GoFAE-EP, (j) GoFAE-CVM with MNIST.

Table S5: The effects of $\lambda$ with GoFAE-SW (left) and GoFAE-SF (right) for MNIST. From top to bottom, we have: (a) p-value changing from epoch 1-50 under varying $\lambda$. (b) Reconstruction error on the testing set with different $\lambda$. (c) Test statistics value on the testing set with different $\lambda$.

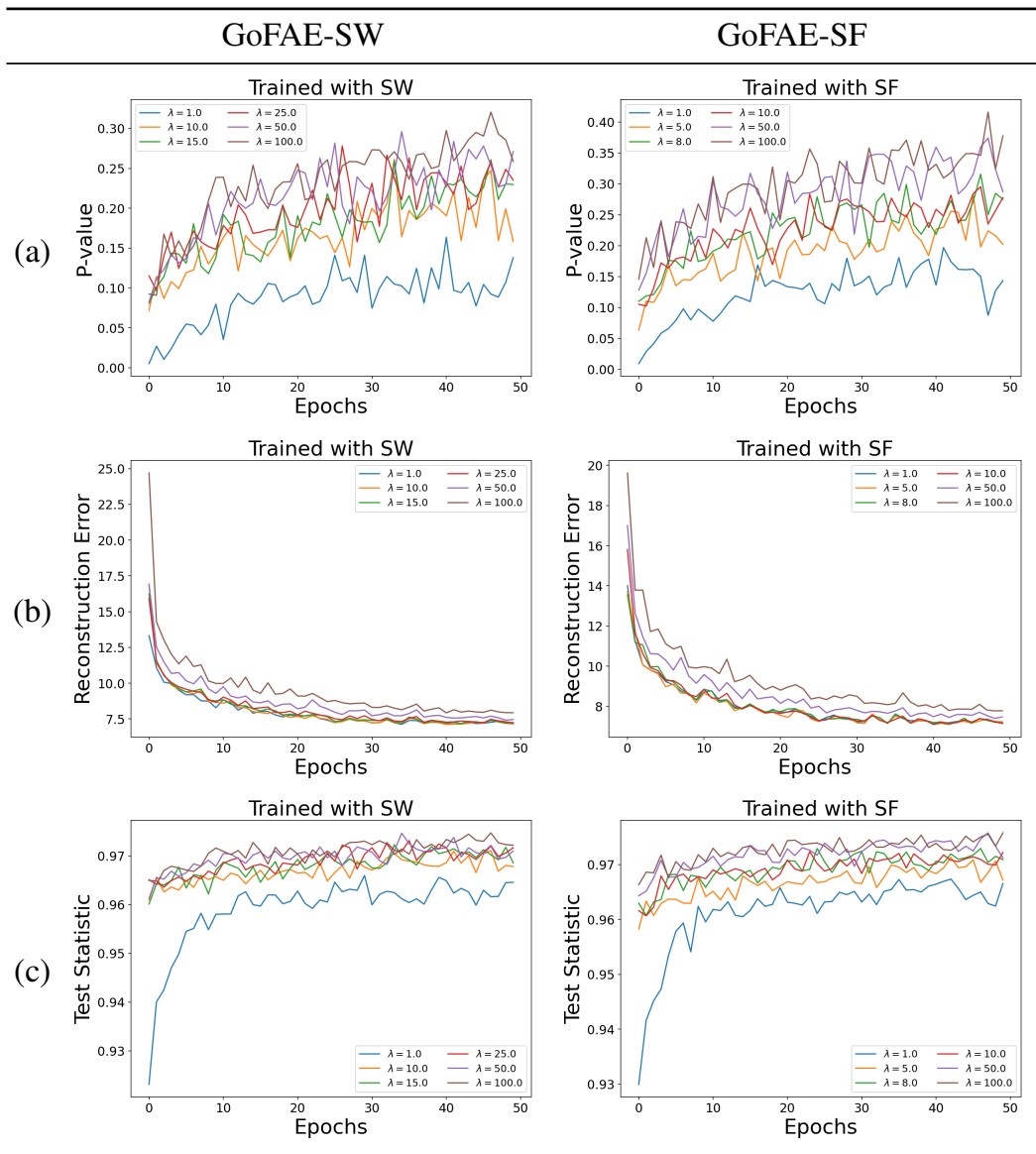

Table S6: The effects of $\lambda$ with GoFAE-CVM (left) and GoFAE-EP (right) for MNIST. From top to bottom, we have: (a) p-value changing from epoch 1-50 under varying $\lambda$. (b) Reconstruction error on the testing set with different $\lambda$. (c) Test statistics value on the testing set with different $\lambda$.

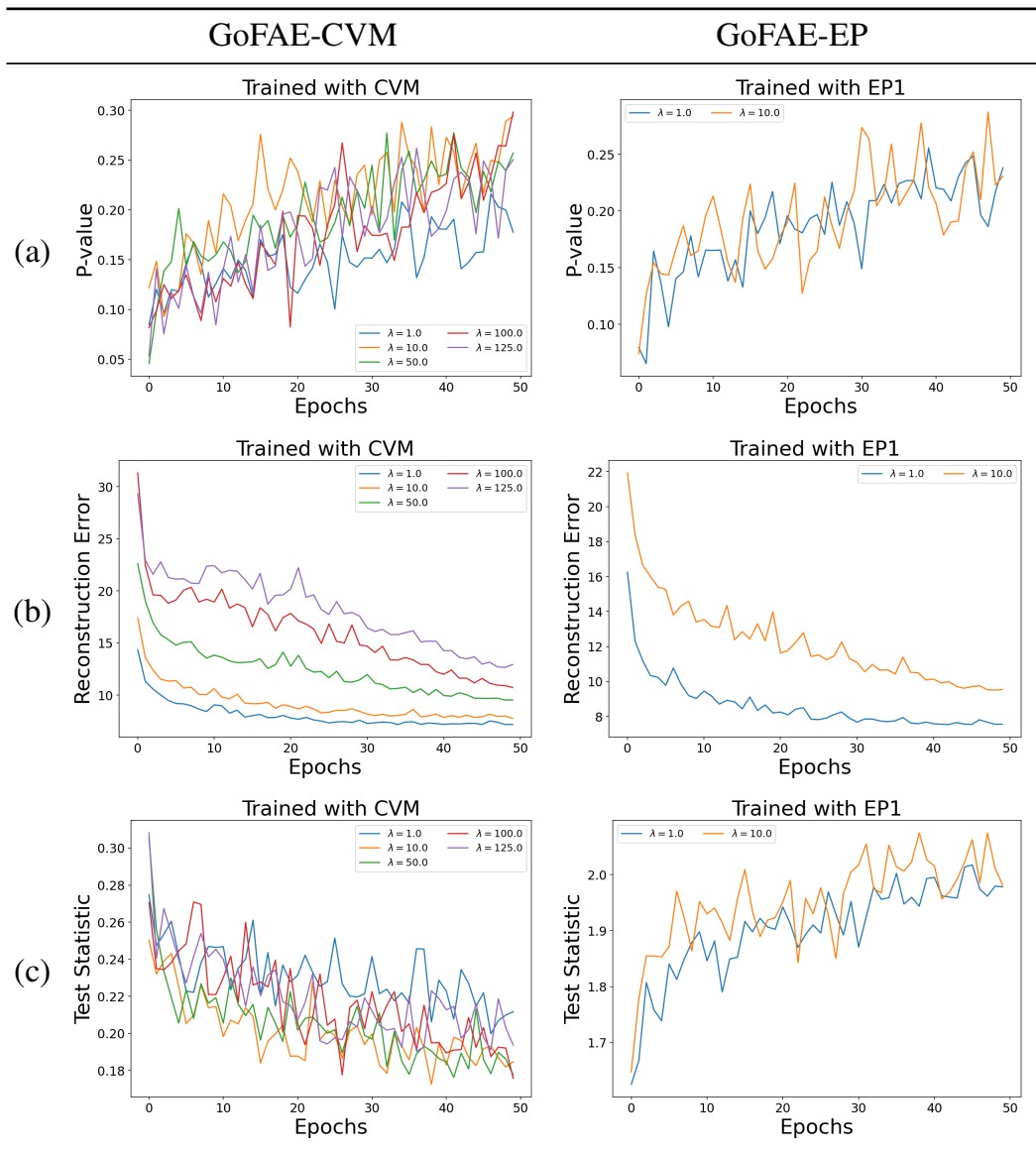

### E.3.3   CIFAR-10

This section includes MSE, FID scores, and p-values from the higher criticism evaluation in Table S7 and reconstructed and generated samples in Figure S7 and Figure S8. Table S8 and S9 demonstrate the effects of $\lambda$ on KS uniformity test, corresponding p-value, reconstruction error, and test statistics for GoFAE-SW, GoFAE-SF, GoFAE-CVM and GoFAE-KS, respectively. The 2-Stage VAE uses the VAE with learned $\gamma$ for reconstruction. The experiments on CIFAR-10 reach the same conclusion as in the CelebA and MNIST, that our GoFAE methods is competitive in both reconstruction and generation.

Table S7: Evaluation of CIFAR-10 by MSE, FID scores, and samples with p-values from higher criticism.

| Algorithm | MSE $\downarrow$ | FID Score $\downarrow$ | |
| --- | --- | --- | --- |
| | | Recon. | Gen. |
| VAE (fixed $\gamma$) | 51.07 (0.04) | 179.75 | 188.55 |
| VAE (learned $\gamma$) | **24.10 (0.01)** | 97.72 | 119.86 |
| 2-Stage-VAE | 24.10 (0.01) | 97.72 | 117.60 |
| $\beta(2)$-VAE | 65.81 (0.07) | 252.98 | 262.55 |
| WAE-GAN | 25.53 (0.00) | 109.55 | 120.30 |
| GoFAE-SW | 24.57 (0.08) | 96.86 | 115.54 |
| GoFAE-SF | 24.69 (0.06) | **97.71** | **115.76** |
| GoFAE-CVM | 24.40(0.06) | 99.76 | 119.51 |
| GoFAE-KS | 24.47 (0.05) | 102.47 | 121.97 |
| GoFAE-EP | 24.74 (0.06) | 106.52 | 124.79 |

| Algorithm | Kolmogorov–Smirnov Uniformity Test | | | | |
| --- | --- | --- | --- | --- | --- |
| | SW | SF | CVM | KS | EP |
| VAE (fixed $\gamma$) | 0.29 (0.28) | 0.28 (0.24) | 0.58 (0.29) | 0.54 (0.30) | 0.48 (0.28) |
| VAE (learned $\gamma$) | 0.00 (0.00) | 0.00 (0.00) | 0.00 (0.00) | 0.01 (0.03) | 0.00 (0.00) |
| 2-Stage-VAE | 0.44 (0.27) | 0.45 (0.28) | 0.48 (0.30) | 0.49 (0.26) | 0.47 (0.28) |
| $\beta(2)$-VAE | 0.43 (0.25) | 0.29 (0.26) | 0.52 (0.33) | 0.46 (0.33) | 0.48 (0.31) |
| WAE-GAN | 0.06 (0.18) | 0.01 (0.03) | 0.41 (0.32) | 0.37 (0.32) | 0.32 (0.28) |
| GoFAE-SW(30) | 0.47 (0.32) | 0.51 (0.28) | 0.37 (0.29) | 0.44 (0.32) | 0.23 (0.22) |
| GoFAE-SF(12) | 0.00 (0.01) | 0.10 (0.16) | 0.01 (0.05) | 0.08 (0.11) | 0.01 (0.01) |
| GoFAE-CVM(5) | 0.03 (0.08) | 0.00 (0.01) | 0.19 (0.21) | 0.31 (0.31) | 0.12 (0.23) |
| GoFAE-KS | 0.00 (0.00) | 0.00 (0.00) | 0.02 (0.04) | 0.10 (0.18) | 0.00 (0.00) |
| GoFAE-EP(5) | 0.43 (0.25) | 0.36 (0.26) | 0.4 (0.25) | 0.53 (0.26) | 0.47 (0.27) |

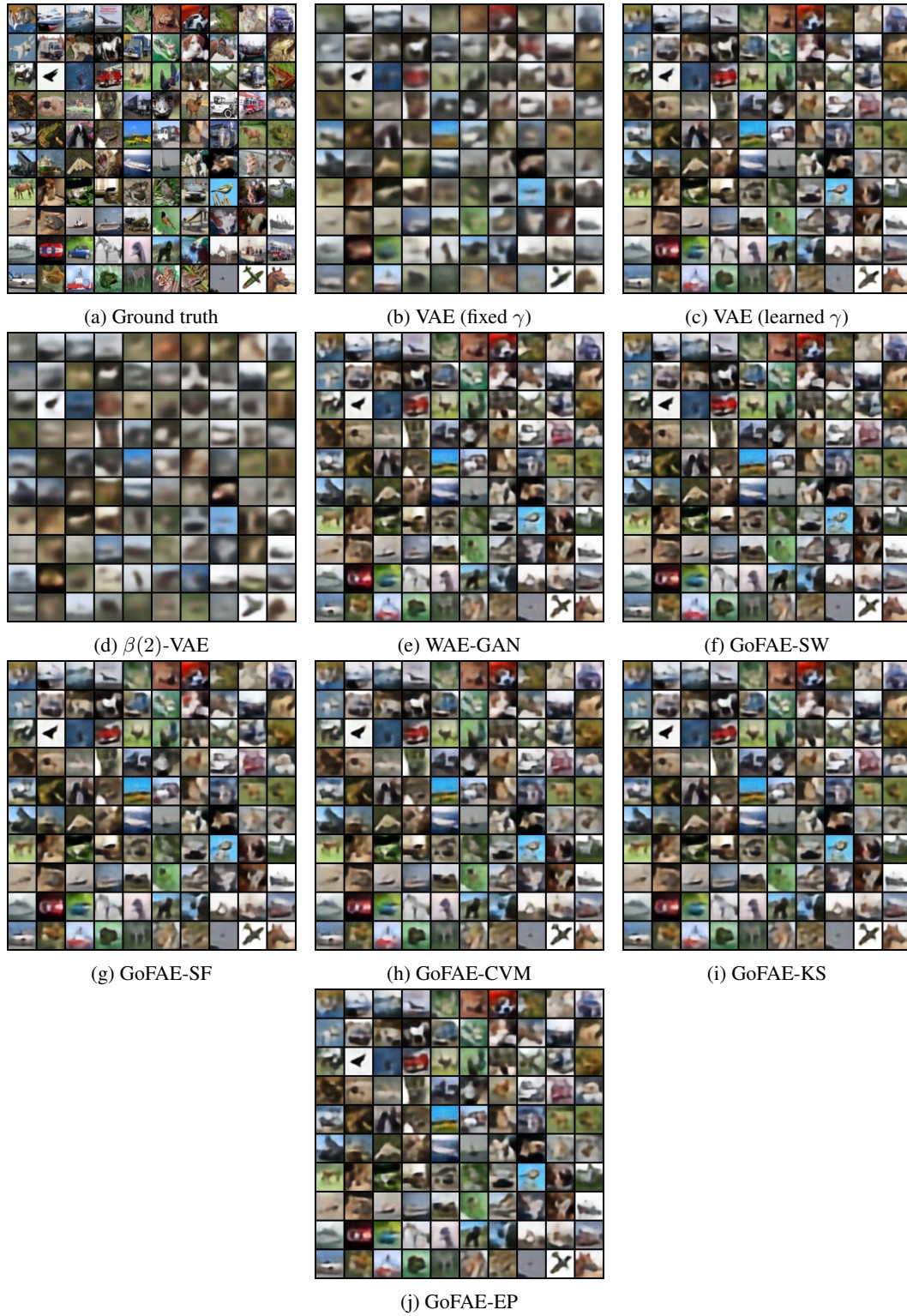

Figure S7: Comparison of reconstructed images between competing methods (b) VAE (fixed $\gamma$), (c) VAE (LEARNED $\gamma$), (d) $\beta$-VAE ($\beta = 2$), (e) WAE-GAN, and our framework: (f) GoFAE-SW, (g) GoFAE-SF, (h) GoFAE-CVM, (i) GoFAE-KS, (j) GoFAE-EP with CIFAR-10.

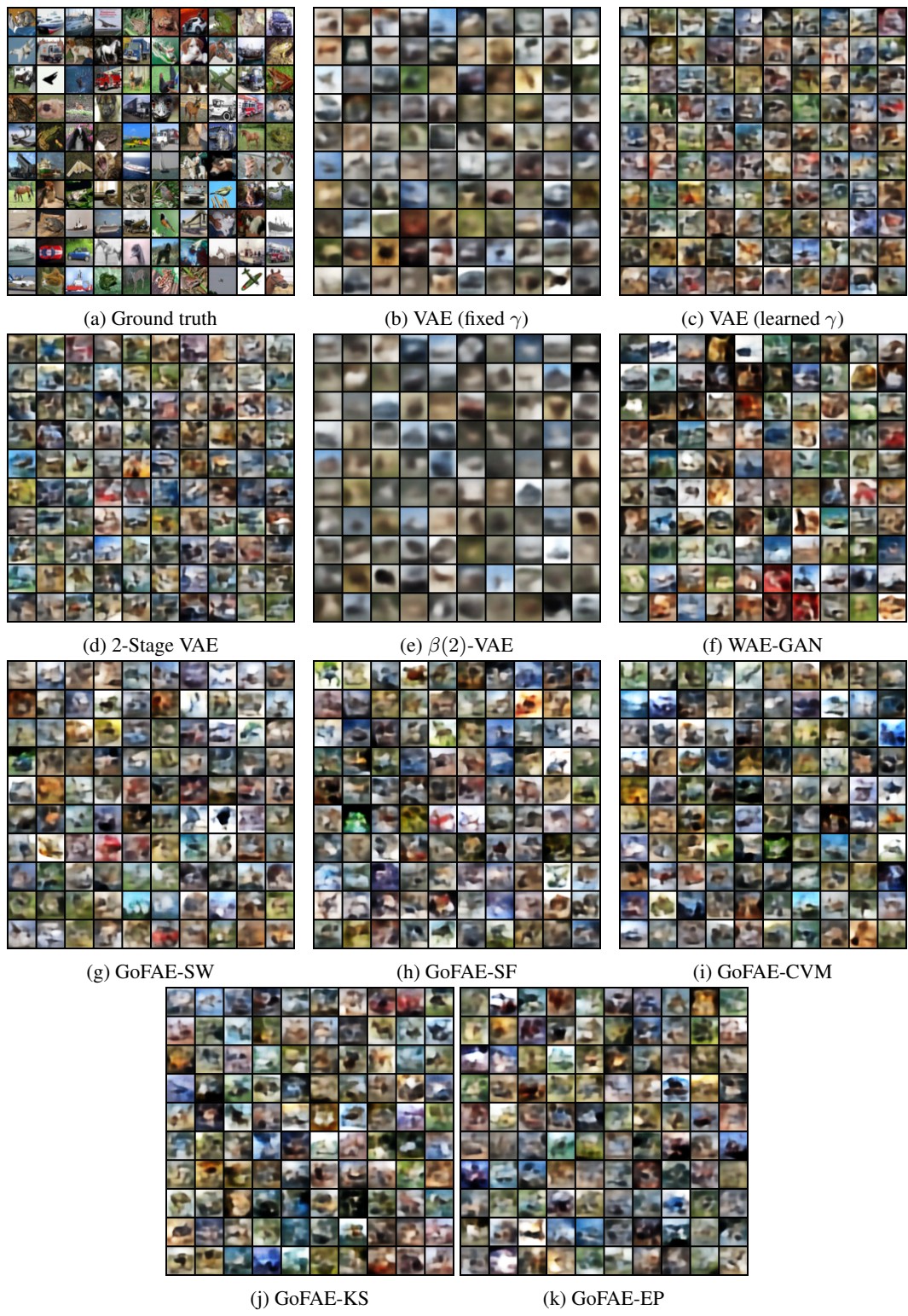

Figure S8: Comparison of generated samples between competing methods (b) VAE (fixed $\gamma$), (c) VAE (LEARNED $\gamma$), (d) 2-Stage-VAE, (e) $\beta$-VAE ($\beta = 2$), (f) WAE-GAN, and our framework: (g) GoFAE-SW, (h) GoFAE-SF, (i) GoFAE-CVM, (j) GoFAE-KS, (k) GoFAE-EP with CIFAR-10.

Table S8: The effects of $\lambda$ with GoFAE-SW (left) and GoFAE-SF (right) for CIFAR-10. From top to bottom, we have: (a) p-value changing from epoch 1-50 under varying $\lambda$. (b) Reconstruction error on the testing set with different $\lambda$. (c) Test statistics value on the testing set with different $\lambda$.

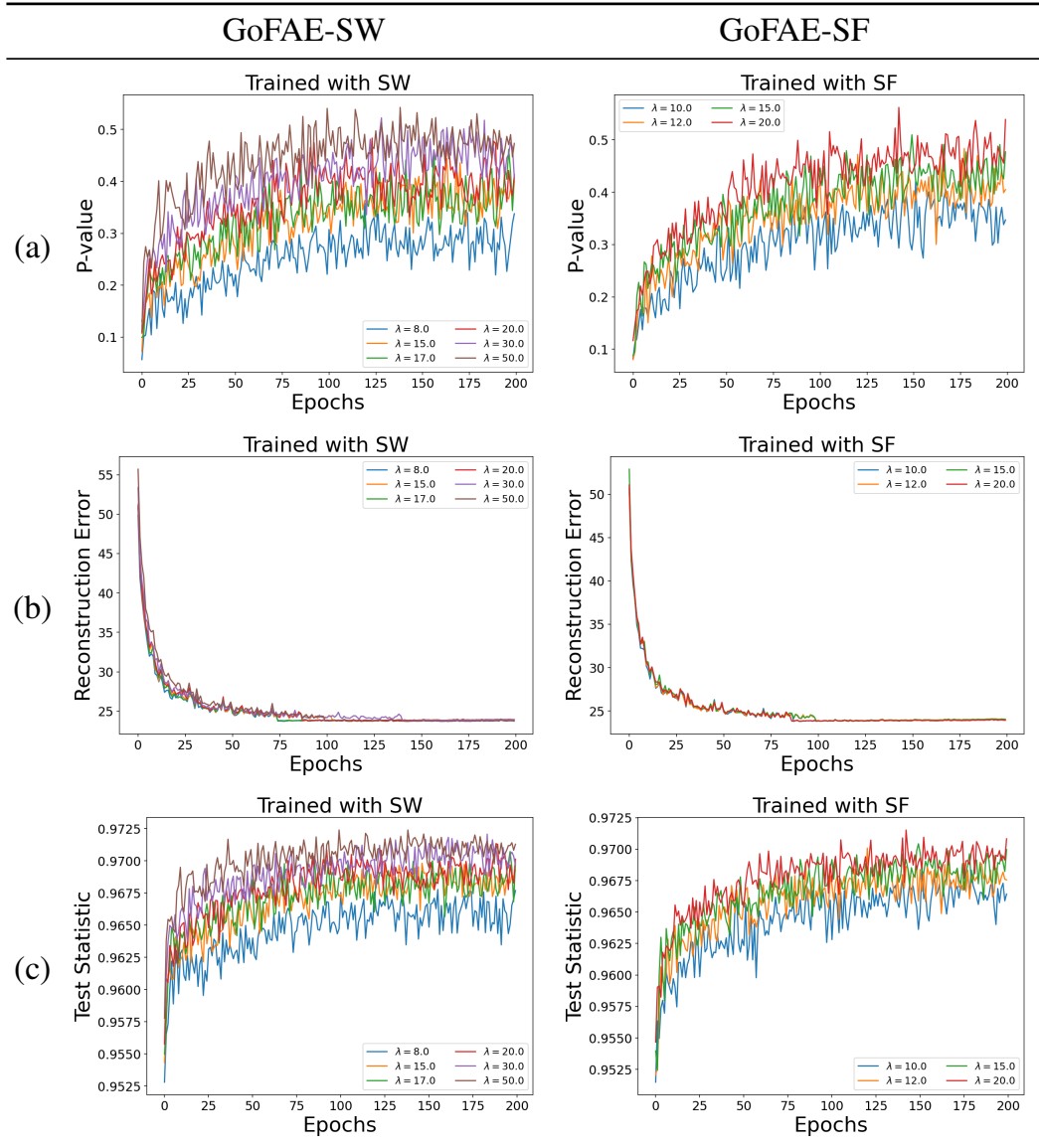

Table S9: The effects of $\lambda$ with GoFAE-CVM (left), GoFAE-KS (middle), and GoFAE-EP (right) for CIFAR-10. From top to bottom, we have: (a) p-value changing from epoch 1-50 under varying $\lambda$. (b) Reconstruction error on the testing set with different $\lambda$. (c) Test statistics value on the testing set with different $\lambda$.

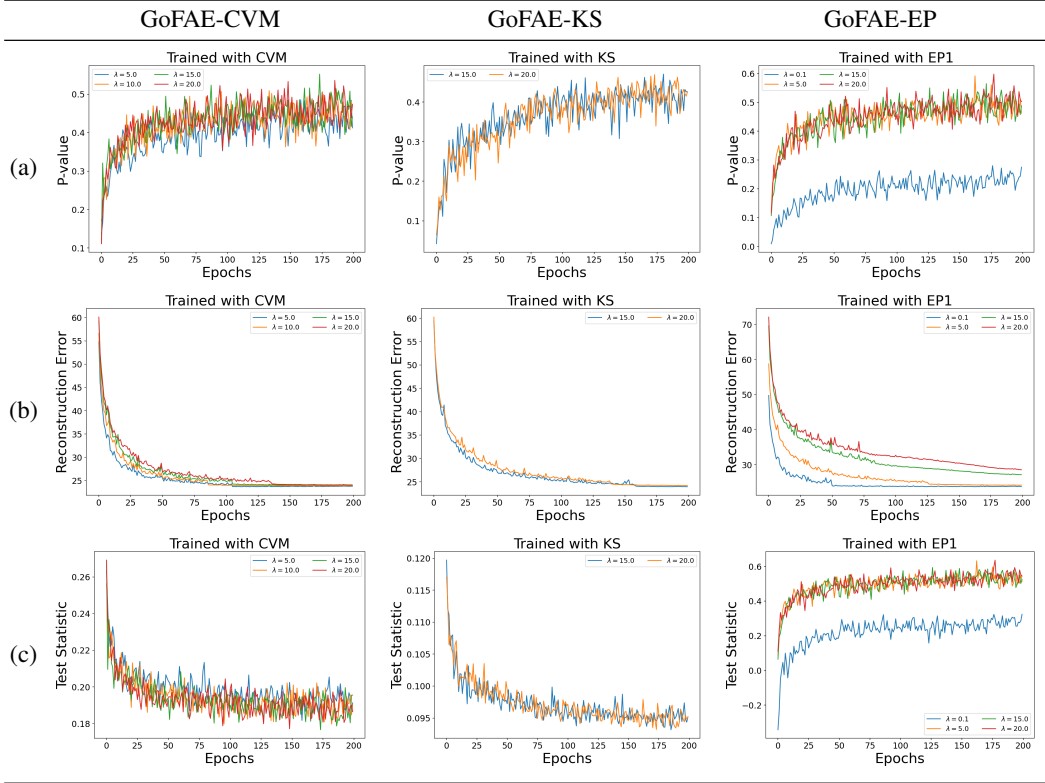

### E.4 CONVERGENCE ANALYSIS FOR COMPETING METHODS

In this section, we include the test set reconstruction error over training epochs for the competing methods for MNIST, CelebA and CIFAR-10 data. Since we adopted a common autoencoder architecture (Section C), reconstruction error follows a similar trend across all models (Figure S9). Since the 2-Stage VAE shares the VAE with learned $\gamma$ as the first stage, we was assessed its convergence in the same manner as the VAE with learned $\gamma$.

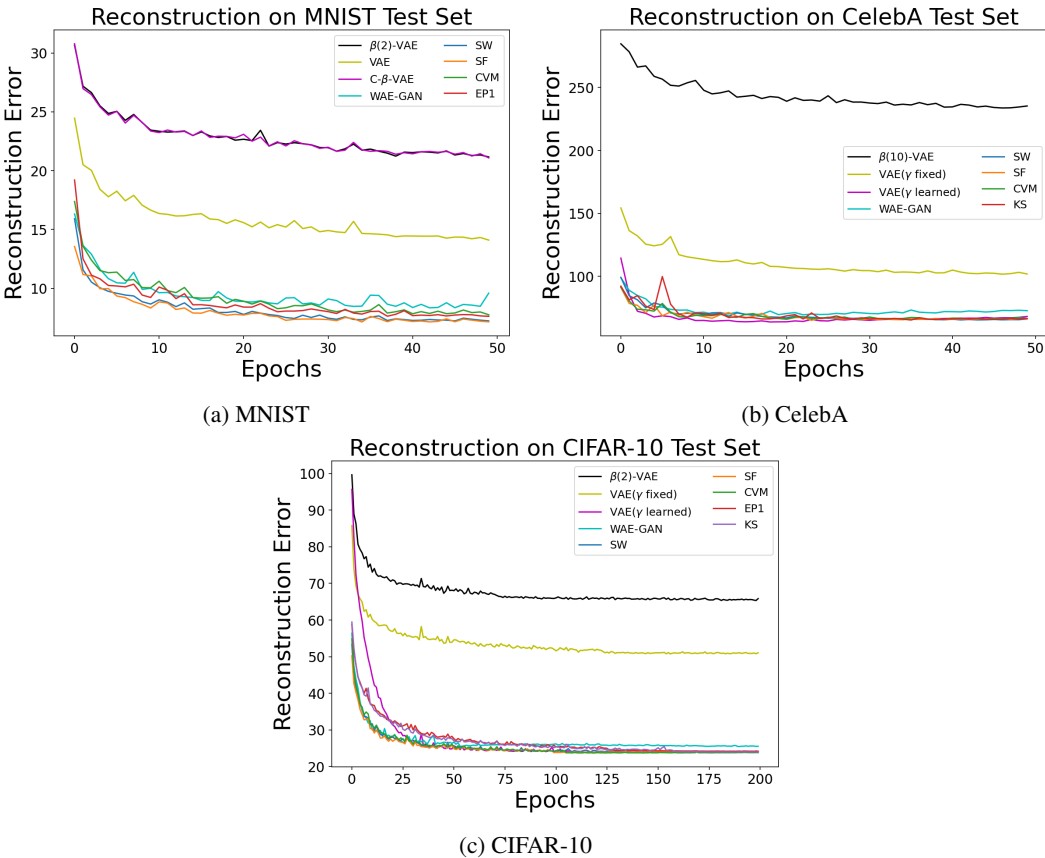

(a) MNIST      (b) CelebA

(c) CIFAR-10

Figure S9: Reconstruction error over training epochs for competing methods.

### E.5 ABLATION STUDY

We proposed Riemannian SGD as a solution to several problems associated with optimizing GoF test statistics. Theoretically, this approach is necessary to prove convergence. However, it remains to be seen whether there is any impact on model performance when the constraints are removed. To explore this, we considered several GoFAE models trained them without RSGD on MNIST and CIFAR-10. We used traditional SGD in place of Riemannian SGD; $\Theta$ is no longer retracted to the Stiefel manifold.

GoFAE models with SW, SF, CVM and EP are retrained using the same architecture and hyper-parameter configuration on MNIST. In place of the orthogonal initialization used for $\Theta$, Kaiming initialization He et al. (2015) was used. Table S2 summarizes the results with the best performing model in bold for each evaluation measure.

GoFAE models with SW, SF, CVM, and EP were also retrained on On CIFAR-10. All architectures and hyper-parameters remained the same as the Riemannian SGD counterparts, except that $\Theta$ which was initialized following Kaiming initialization. Table S11 summarized the results with the best performing model in bold for each evaluation measure. These results show empirical support for

using Riemannian SGD, particularly for generation. The results for reconstruction performance are less clear and would benefit from further experimentation.

Table S10: Comparison of Riemannian SGD (left) and standard SGD (right) for MNIST.

|     | Riemannian SGD | | | Standard SGD | | |
| --- | --- | --- | --- | --- | --- | --- |
|     | MSE | Gen FID | Recon FID | MSE | Gen FID | Recon FID |
| SW  | **7.16** | **16.58** | **7.57** | 7.31 | 19.47 | 7.71 |
| SF  | **7.08** | **18.35** | 7.47 | 7.12 | 20.36 | **7.36** |
| CVM | **7.67** | 15.56 | 7.71 | 7.79 | **15.12** | **7.60** |
| EP  | **7.54** | **15.85** | 7.84 | **7.54** | 16.59 | **7.63** |

Table S11: Comparison of Riemannian SGD (left) and standard SGD (right) for CIFAR-10.

|     | Riemannian SGD | | | Standard SGD | | |
| --- | --- | --- | --- | --- | --- | --- |
|     | MSE | Gen FID | Recon FID | MSE | Gen FID | Recon FID |
| SW  | 24.58 | **117.05** | **100.11** | **23.94** | 119.48 | 101.27 |
| SF  | 24.69 | **115.76** | **97.71** | **23.92** | 118.05 | 101.37 |
| CVM | 24.40 | **119.51** | **99.76** | **23.68** | 120.48 | 100.22 |
| KS  | 24.47 | **121.97** | **102.47** | **24.03** | 122.23 | 103.22 |
| EP  | 24.74 | **124.79** | 106.52 | **24.05** | 124.88 | **105.19** |

