# OpenReview forum: "Auto-Encoding Goodness of Fit"
_ICLR.cc/2023/Conference — ICLR 2023 poster_

### Official Review · Reviewer_5naA · 2022-10-24

**Confidence:** 2
**Correctness:** 3
**Technical Novelty And Significance:** 3
**Empirical Novelty And Significance:** 3
**Recommendation:** 6

**Clarity, Quality, Novelty And Reproducibility:**

The paper is overall clear and high quality in the theoretical part. All results are well motivated and technically supported. The analysis and empirical results in experiments are mostly clear, with some issues as mentioned above. The framework of using these tests to obtain a better latent is novel to my knowledge. The optimization technique is a well-known manifold SGD but the paper has a bunch of novel theoretical analysis towards it when using the specific objective function. Regarding reproducibility, I did not find code for the experiments.

**Strength And Weaknesses:**

I am not an expert in GoF tests so please refer to other reviewers for evaluation of this part.

The paper is clearly written and very well motivated (i.e. having a better latent). It is technically sound with all the theorems supporting the claims or algorithms in the relevant section. The GoF framework for AE is novel and very general in the sense that many different tests can be applied. It also somewhat generalizes $\beta$-VAE when using the likelihood-ratio tests for normality (correct me if I'm wrong). There are also extensive experiments studying different aspects of the proposed method.

There are several weaknesses though. The first is that running statistical tests can be slow, especially when you also need to do backward with those values. It would be nice to have some analysis or experiments on that. The second is that $\lambda$ seems to have a large impact on the results, and therefore the paper needs to expand on how to select this hyperparameter. The third is that individual KS test results (fig 4) have very high variance. This questions the effectiveness of the proposed method unless you sample a huge number of samples for the tests. The forth is that the empirical generation/reconstruction quality is only marginally better than the baseline models, while some baseline methods also have very good latent according to the uniformity test statistics. It would be more convincing and impactful to test on modern VAE architectures and see if there is significant improvement. Finally, the paper seems to evaluate every possible GoF test but cannot provide insight on using which one. The results from different tests are indeed somewhat different, so it is best to have analysis on this.

**Summary Of The Paper:**

The paper aims to force the latent distribution to be close to a normal distribution in AE, similar to the goal of $\beta$-VAE. To do this the paper applies a good-of-fitness test, and optimize the reconstruction error with the test statistics as a regularization term. The paper then proposes a manifold SGD to solve the optimization problem, and gives a theoretical guarantee for convergence. Experiments on several image datasets show the proposed method achieves similar or slightly better generation quality but has a better (close to Gaussian) latent.

**Summary Of The Review:**

I find this paper to be well motivated, technically sound, and empirically interesting. It is definitely an interesting idea to use GoF tests for better latent. However, I'm not expert of GoF tests so I cannot evaluate part of the paper very fairly. Therefore I'd recommend marginally accept at this moment with less confidence.

---

> ### Author Response · Authors · 2022-11-19
> **Response to reviewer**
>
> Dear reviewer,
>
>
>
> Thank you for taking the time to review our manuscript. Please find our responses below.
>
>
>
> **Question in Strength**
>
> > It also somewhat generalizes $\beta$-VAE when using the likelihood-ratio tests for normality
>
> This is an interesting perspective that highlights differences between the two approaches.
>
> - $\beta$-VAE, much like VAE (which optimizes the evidence lower bound), does not guarantee that the aggregated posterior matches the prior.
>
> - LRT is testing on the parameter space of the assumed distribution. GoF is testing on the distribution class.
>
> - The correlation based GoF test assess if the $d_{W_2}$ is small. However, an LRT essentially is on the KL divergence, so it cannot directly assess the $d_{W_2}$.
>
>
>
> **Weaknesses**
>
> > Running statistical tests can be slow, especially when you need to do backward...
>
> As test statistics are closed form and can be computed quickly, as it is only done on a minibatch of data. We do require computing multiple projections and generating an empirical null distribution using Monte Carlo simulation, but the projections can be efficiently computed, and the empirical null need only be computed once prior to training (see the top-level section *Effect of minibatch on uniformity*). Hence backpropagation is quick.
>
> > Lambda seems to have large impact on results, paper needs to expand on how to select this hyperparameter
>
> $\lambda$ indeed has a large impact on the results and is expected. The subsection titled *Effect of $\lambda$ and Mutual Information (MI)* in section 5 and Fig. 5 interpret the impact of $\lambda$. When $\lambda$ is too small, the GoF p-values are close to 0 and indicate that both the (local) minibatches and therefore the (global) data set are not normal (i.e., the global p-value distribution deviates from uniformity). When the coefficient is too large, optimization using GoF tests will ensure minibatches appear normal since the p-values will be close to 1. However, if the encoder is able to learn a parameterization such that every minibatch appears Gaussian, then it has overfit. Nestled between these two extremes is a “Goldilocks zone” where, for a range of lambda values, the p-values appear uniformly distributed. Higher criticism is meant to detect this.
>
> The reason why the uniformity p-values appear to have high variance, is that the probability integral transform also applies to the KS uniformity test itself. Therefore, as lambda increases, not only do the GoF p-values increase, producing uniformly distributed GoF p-values, but the KS uniformity p-values should also be uniformly distributed. When $\lambda$ is too large or small, the GoF p-values become skewed (not uniform) which leads to the KS uniformity p-values dropping back to zero (Fig. 5a). The high variance is precisely what the theory indicates should occur.
>
> > Individual KS test results have high variance. Need for large samples?
>
> Please refer to the previous answer as the variance is directly related to lambda. Running Algorithm 1 for more iterations would produce more precise values of the average which could lead to a better selection of lambda.
>
> > Empirical generation/reconstruction only marginally better...use modern VAE archs
>
> Although the primary purpose of our work is not to simply produce the best images (see top-level response *Empirical results and evaluation*), Table 1 does show more than marginal improvement. Nevertheless, these methods can certainly be applied to architectures with higher capacity.
>
> > Paper seems to evaluate every possible GoF test but cannot provide insight on which...
>
> In the paper we consider a small proportion of possible GoF tests. There is extensive statistical literature regarding univariate and multivariate GoF tests of normality as well as other distributional classes.
>
> It is not possible to make general claims regarding which test is best because we cannot truly assess the statistical power of each GoF test. Assessing power requires the alternative distribution to be specified. When comparing the statistical power of different GoF tests, simulation is commonly done using various specified alternative distributions. GoF tests that have shown strong performance across many alternative distributions are typically the ones used to assess normality on samples coming from an unknown distribution.
>
> However, the Shapiro-Wilk and Shapiro-Francia GoF tests consistently lead to strong evaluation metrics, not only against the competing models, but within the GoF tests used. These tests also happen to be related to the $L_2$-Wasserstein distance which is an upper bound on the distances associated with other tests.
>
> > Code is missing
>
> We commit to releasing the full GoFAE and higher criticism source code if accepted.

---

### Official Review · Reviewer_i5br · 2022-10-24

**Confidence:** 4
**Correctness:** 3
**Technical Novelty And Significance:** 3
**Empirical Novelty And Significance:** 3
**Recommendation:** 6

**Clarity, Quality, Novelty And Reproducibility:**

### Clarity:
The text is easy to follow

### Quality
the text attempts to solve an important issue even if I am not finding the numerical experiments extremely convincing.

### Novelty:
I believe that the approach is novel. I would like to emphasise that I have not kept up to date with the more recent literature on auto-encoders over the past 2 or 3 years. It would be worth it if the authors clarified whether similar approaches (eg. use of test statistics to quantify distance to Gaussian family) have already been proposed.

### Reproducibility
All good.


**Strength And Weaknesses:**

**Strength:**
1. the paper reads well and is straightforward to follow
2. if I am understanding correctly, using a GOF to test whether the aggregated prior $[F(x_1), \ldots, F(x_N)]$ belongs to a class a distribution has not been proposed before in the context of fitting generative auto-encoder. Can the author confirm this?
3. tuning the parameter $\lambda$ is indeed a difficult and important problem in practice and the proposed approach is an interesting step towards solving this problem

**weaknesses**
1. as the authors describes it, there are non-identifiability issues. But deep-neural nets are also extremely non-identifiable, indeed. I have not been able to find the parts where the authors show that a standard gradient descent does lead to instabilities and optimisation issues? Their approach of containing the last layer to orthogonal set of weights is convincing, but may be over-kill if the standard non-constrained parametrisation works reasonably well.
2. It seems like the higher-cricism criteria is quite sensitive to the batch size. If that is correct, I think this should be discussed much more carefully since this appears to be crucial to the tuning of $\lambda$. As a matter of fact, since the tuning of $\lambda$ is really one of the main novelties of the paper, I think that this section should be much expanded. There are several theoretical results (eg. Prop 1 and 2) that do not seem very relevant to the article and could safely be ignored if page-count is an issue.
3. I believe that neither reconstruction error (a basic auto encoder would be fine), neither a good match between the aggregated prior and a Gaussian distribution, are extremely interesting goals in themselves in the setting of the paper. It would seem more interesting to check  whether the encoded samples are useful on downstream tasks (i.e. meaningful compression): is it the case? Can the authors describe what problem they really would like to solve, i.e. what a "meaningful latent representation of the data" is. In some sense, I am not finding the chosen metrics (FID / reconstruction error) meaningful for the paper.


**Summary Of The Paper:**

For a dataset of samples $[x_i]_{I=1}^N$ consider an auto-encoder $(F,G)$ that maps samples to latent representations through an encoder function $F$, i.e. $x_i =F(x_i)$, and a decoder $G$ that maps latent representations to samples, $x = G(z)$. These maps can be stochastic and are typically parametrized by neural networks. This auto encoder is trained by minimising a loss of the type:
$$\text{dist}[x, G \circ F(x)] + \lambda  \cdot T[ (F(x_1), \ldots, F(x_N)))]$$
where $T$ is a statistics that test whether the empirical distribution of the samples $(F(x_1), \ldots, F(x_N))$ belongs to a class of allowable distributions. The parameter $\lambda$ quantifies the strength of the regularisation. Typically, the allowable class of distributions is the Gaussian family. Note that it is more general that enforcing a fixed covariance structure (eg. isotropic) as is often done in many scenarios.

The paper makes several contributions:
1. proposes using a Goodness of Fit to test whether the aggregated posterior belongs to a family of allowable prior (instead of a fixed prior)
2. when the Gaussian family is used, there are non-identifiability issues (eg. rotation/scale/etc...) that can lead to optimisation instabilities. The paper proposes algorithmic ways to mitigate these issues.
3. proposes to using the set of test statistics evaluated on each mini batch for tuning the parameter $\lambda$.

**Summary Of The Review:**

The problem of finding better representations of complex datasets (i.e. dimension reduction with associated semantic meaning) is crucial to many areas of science. Any improvement in this area is likely to have important implications. The text is proposing a new framework for this very purpose in the context of generative auto encoder. While the proposed methodology is certainly interesting, I am not finding the evaluation of the method adequately implemented. I would like to encourage the authors to work on this aspect of this very interesting article.

---

> ### Author Response · Authors · 2022-11-19
> **Response to reviewer**
>
> Dear reviewer,
>
>
>
> Thank you for taking the time to review our manuscript. Please find our responses below.
>
>
>
> **Strength 2 Question**
>
> > using a GOF to test whether the aggregated prior belongs to a class a distribution has not been proposed before in the context of fitting generative auto-encoder. Can the author confirm this?
>
> We use local GoF tests to produce p-values which are passed to the higher criticism mechanism for uniformity testing. To the best of our knowledge, no one has used minibatch p-values coming from a GoF test to assess whether the aggregated posterior belongs to a prior distribution class.
>
>
>
> **Response to Weakness 1**
>
> In Section 4: Identifiability, Singularities, and Stiefel Manifold (beginning with the line “Second, almost...”), we discuss how the gradient and Hessian of an affine invariant GoF test will not be bounded. This means that it is possible for the test statistic value to remain fixed while the gradient can explode. Fig. 4b provides an illustration and Theorem 4 formally states this. If traditional SGD is used, then it is possible for the gradient to explode during model training. Moreover, the affine invariance also leads to an unbounded domain on $\mathbf{\Theta}$. Large weights and gradients are often the culprits when diagnosing model instabilities. Unfortunately, these issues also mean convergence of $\mathbf{\Theta} $ cannot be proved using the standard SGD approach, which means we cannot prove convergence of the model.
>
> Our goal is to use the flexibility provided by GoF tests while avoiding non-identifiability, gradient explosion, and unbounded domain on $\mathbf{\Theta} $. Therefore, we restrict $\mathbf{\Theta} $ to the Stiefel manifold, which justifies the use of Riemannian optimization. We are not claiming that traditional SGD must lead to instabilities and optimization issues. In fact, we conducted a small ablation study where $\mathbf{\Theta} $ was not restricted to the Stiefel manifold, and compared traditional and Riemannian SGD on CIFAR10 data (see Appendix E.5). There are differences, but the model did converge in all of our tests, likely due to the multiple levels of stochasticity (minibatch, sampling, and projection). However, we can contrive scenarios where these issues are apparent through specific initializations, architecture configurations, or dataset and minibatch size. In sum, if traditional SGD is used, we have no assurance that the optimization issues are avoided, and as a result, no convergence proof.
>
>
>
> **Response to Weakness 2**
>
> > It seems like the higher criticism criteria is quite sensitive to the batch size
>
> There may be a misunderstanding here. With respect to batch size, all experiments used a minibatch size of $m=128$ samples. If this comment was instead about the effect of batch size on uniformity, please see the top-level *Effect of minibatch on uniformity* response.
>
> > There are several theoretical results...
>
> Please see our top-level response *Clarity and purpose of Section 2: Propositions 1 and 2*.
>
> **Response to Weakness 3**
>
> > I believe that neither reconstruction error... Can the authors describe what problem they really would like to solve...”meaningful latent representation of the data”
>
> Please see our top-level response *Improved or useful latent representation*.

---

> > ### Comment · Reviewer_i5br · 2022-11-21
> > **Thanks**
> >
> > Thank you for these answers that have clarified a few of my misunderstandings.

---

### Official Review · Reviewer_Fj2Y · 2022-10-25

**Confidence:** 3
**Correctness:** 3
**Technical Novelty And Significance:** 3
**Empirical Novelty And Significance:** 2
**Recommendation:** 6

**Clarity, Quality, Novelty And Reproducibility:**

- Clarity: The paper is hard to follow and writing needs to be improved.
- Quality: The paper is quite technical.
- Novelty: The proposed method is novel.
- Reproducibility: I believe the details provided in the appendix is enough to reproduce the results.

**Strength And Weaknesses:**

## Strength
- The main idea behind the method is interesting and sound.
- The paper is technical.
- The proposed method is theoretically justified.

## Weakness
- Writing needs to be improved. The current draft is very hard to follow with too many (unnecessary) details that hides the main points.
    - Current section 2 mixes background/prior work with motivations. I suggest to have a dedicated related work section to have the prerequisite prepared and use 2 paragraphs to make your main motivation.
    - Way too many acronyms are used.
    - The discussion around figure 2 is good but is presented way too late. I think this is the main intuition this method is based on and should be made clear earlier in the paper.
    - There are too many technical details mentioned when presenting the method. I suggest to simplify the presentation first and add details later on so that readers could get a high-level, accurate overview first before seeing too many details.
- Experiments with real-world data do not focus on evaluating the quality of representation at all. I don't think evaluation purely based on generative quality is enough---if that's the only goal, why don't we use a GAN?

**Summary Of The Paper:**

The paper proposes a method called Goodness of Fit Autoencoder (GoFAE) which uses goodness of fit (GoF) to regularize the posterior at the mini-batch level and selects a regularization coefficient at the global level (a few mini-batches) by making the distribution of p-values from each mini-batch uniform. Empirical results show that GoFAE achieves comparable image quality to existing deep generative models while having a latent space that is more Gaussian-like.

**Summary Of The Review:**

I quite like the paper but it looks to me that the paper is not ready. It needs some good work on improve the presentation (writing) and better evaluation to provide empirical evidence on improved latent representations.

---

> ### Author Response · Authors · 2022-11-19
> **Response to reviewer**
>
> Dear reviewer,
>
>
>
> Thank you for taking the time to review our manuscript. Please find our responses below.
>
>
>
> > Writing needs to be improved...
>
> Please see our first two overall responses on manuscript readability and clarity.
>
> > Current section 2 mixes background/prior work....suggest to have dedicated related work section
>
> We have split Section 2 into 3 parts. The first covers the necessary background, the second focuses on the Wasserstein bounds, and the third makes explicit how the previous bounds can be extended to the case of a distribution class which motivates the use of GoF statistics.
>
> Many concepts that do not neatly fit into a particular area are needed to derive the GoFAE.  We believe a dedicated related work section that combines optimal transport, hypothesis testing, regularization methods, and Riemannian optimization into a single section would be confusing and more difficult to motivate. Presenting and discussing the related material as we move through the paper allows us to more easily motivate and justify the different components of the GoFAE.
>
> > Too many acronyms are used.
>
> Please see our overall response *Readability, notation, terminology, and acronyms* where we describe the removal of several acronyms. The remaining acronyms are used out of necessity.
>
> > The discussion around figure 2...presented too late...intuition should be made earlier
>
> In a previous draft, the figure appeared earlier in the manuscript; however, after editing the manuscript based on reviewer comments, we placed the figure alongside the notations that are necessary to understand it. Also, the newly added Figure 1 provides an overview about the first stage of GoFAE.
>
> > There are too many technical details...
>
> We spent considerable efforts improving readability and clarity of the manuscript (see first two top-level responses). We have relegated many fine details to the appendix and believe that what is left is sufficient to keep the main text precise yet self-contained.
>
> > Experiments with real-world data do not focus on evaluating the quality of representation....why not GAN?
>
> Please see our response in the top-level comment titled *Improved or useful latent representation*.

---

> > ### Comment · Reviewer_Fj2Y · 2022-12-02
> > **Acknowledgement of author response**
> >
> > Thanks for the response and the updated manuscript.
> > I think the paper itself reads much better now and figure 1 helps the understanding of GoFAE as well as showing how it differs from well-established method (and indicates why it might do better).
> >
> > Re. **Improved or useful latent representation**, it maybe helpful to extend the point on "reduced-dimension representation" by having an experiment to answer "what percentage of dimension do GoFAE need to achieve the same FID as a VAE with D dimension". In general, I do agree your point that the current results show some indication that the representations are useful but the paper would benefit from some more direct metric from downstream tasks.
> >
> > Anyway, given the overall improvement, I raised my score to 6.

---

> > > ### Author Response · Authors · 2022-12-08
> > > **Follow up**
> > >
> > > Thank you for your suggestion, which is well taken. We are continuing to study this aspect of the model and believe it is a promising direction for future work.

---

### Official Review · Reviewer_ySEm · 2022-10-28

**Confidence:** 2
**Correctness:** 2
**Technical Novelty And Significance:** 2
**Empirical Novelty And Significance:** 2
**Recommendation:** 5

**Clarity, Quality, Novelty And Reproducibility:**

While this work is novel and original, the clarity of the paper has room for improvement, as mentioned above.

Minor comments:
* The use of p-values. Some tests may only have p-values computable asymptotically, and such tests may not have uniformly-distributed p-values depending on the minibatch size.
* Second line on page 2. GoF tests do not only concern posterior and prior distributions.
* The aggregate posterior $\mathbb{P}_Y$ is a confusing word here, as the authors sometimes denote two things by $F$. One is a vector-valued function and the other is a measure-valued function. The use differs from conventional use. The authors motivate the optimisation problem following the optimal transport framework. Theorem 1 in the Wasserstein auto-encoder paper (Tolstikhin et al., 2018) states that the encoder is in general a measure-valued function of x.
* Line 8 in Section 3.1. Why do we need the expecation $\mathbb{E}[T(F_{\theta}(X)]$? Is the test randomised here? Also the direction of the inequality is not correct?
* The symbol for the test statistic $T$ is overloaded, and it's unclear what $T$ takes as input.
* Line 3 in Algorithm 3: shouldn't G be F?
* Section 4 starts with training specifics using a test statistic $T$ without mentioning a training objective, making the section difficult to read. I would introduce the objective in (2) first, and then mention desiderata/challenges to motivate the proposed optimisation method.
* HT-Trainable. Almost everywhere differentiability seems to be too weak for practical implementations. What happens if we have to evaluate the gradient at a non-differentiable point? Do we define weak derivatives for all non-differentiable points?
* Section 4.2. The minibatch subscript $i$ is bold.



**Strength And Weaknesses:**

The paper addresses interesting problems as above. The proposed optimisation algorithm might be of interest, although the benefit of the optimiser is not substantiated in the main paper (there seems no comparison against other optimisers).


The paper has the following weaknesses:

1. The lack of clarity -- the paper is unfortunately not well-written. I find it particularly challenging to locate the proposed method for model selection, and how it should be used. Although the first experiment provides some information, the lack of a formal description makes it difficult to understand its intended use and therefore the evaluation of the paper.
2. The justification for the proposed selection technique is not strong.
    * There are different measures we could use to choose a model. For example, if we are solely interested in the quality of generators, we can use appropriate metrics for that purpose. As I understand it, Propositions 1 and 2 provide a justification regarding the Wasserstein-2 distance between the generator and model distributions, but these seem to be too indirect (also involve gradient norms). Why is the proposed diagnosis relevant to "good" representations?
    *  According to the first paragraph in Section 2, it appears that what we are interested in the proximity of the aggregated posterior $\mathbb{P}_Y$ *during the training* rather than the end of the training (we minimise the encoder-dependent reconstruction cost subject to the constraint $\mathbb{P}_Y=\mathbb{P}_Z$). The proposed method (Algorithm 1) only seems to indicate how close $\mathbb{P}_Y$ is to the Gaussian at the end of the training. This point makes me wonder how the proposed method is useful. Is the proposed method to be used to modify the regularisation coefficient during the training?


**Summary Of The Paper:**

This paper mainly addresses the following two problems:
1. It introduces an objective function for training generative autoencoders (encoder-decoder networks). The training objective consists of two functions: one represents the reconstruction error, and the other indicates the discrepancy of the latent code distribution with the latent prior (e.g., a standard normal distribution). The authors use a test statistic for normality testing and propose a bespoke SGD method for this objective.
2. It proposes a method to choose a regularisation coefficient (i.e., the relative weight of the normality loss above). The authors use p-values obtained from the test corresponding to the statistic computed from mini-batches. Specifically, the authors use a measure of the uniformity of p-value distribution as a model selection criterion.



**Summary Of The Review:**

The paper is challenging to evaluate due to its lack of clarity.  I would recommend a substantial revision.

---

> ### Author Response · Authors · 2022-11-19
> **Response to reviewer**
>
> Dear reviewer,
>
>
>
> Thank you for taking the time to review our manuscript. Please find our responses below.
>
>
>
> > the benefit of the optimiser is not substantiated…(no comparison against other optimizers)
>
> We are not introducing a new optimizer. Riemannian SGD is an established method, which we indicate using several citations just before the subsection 4.1 header and the convergence results from Bonnabel (2013) in section 4.2. Due to the form that GoF statistics take, there are concerns regarding the optimization process (see Section 4: Identifiability, Singularities, and Stiefel Manifold.). To mitigate these issues Riemannian SGD is the proposed solution. However, the original Riemannian SGD convergence theorem of Bonnabel places a strong condition on the stochastic gradient, which GoF statistics will generally not satisfy. We prove that this condition can be weakened considerably while still providing convergence guarantees (Theorem 6). Without this approach, traditional SGD convergence results are not applicable. Empirically, we performed an ablation study comparing Riemannian SGD with SGD (Tables S9 and S10).
>
> **Response to Weakness 1**
>
> Model selection is done by evaluating higher criticism. The general procedure is described in Section 3.2 and Algorithm 1. The procedure is further explained and evaluated in Section 5: Effect of $\lambda$ and Mutual Information (MI) (first two paragraphs). We added additional clarifying text and pseudo-code to the appendix section D.4.
>
> **Response to Weakness 2**:
>
> The “justification for the proposed selection technique is not strong” was supported by two points:
>
> > For example, if we are solely interested in the quality of generators...
>
> We agree that there are multiple model selection approaches that could be employed. GAE manage a balance between reconstruction and prior matching. While we are interested in the performance of different evaluation measures, our model selection is restricted to ensuring the model sufficiently matches the specified prior as it is part of the theoretical foundation (see Proposition 1 and 2). Additionally, please see our top-level response in the section *Improved or useful latent representation* regarding “good representations”.
>
>
>
> > The proposed method (Algorithm 1) only seems to indicate how close $\mathbb{P}_Y$ is to the Gaussian at the end of the training.
>
> The regularization coefficient is found in a hyperparameter selection step via grid-search for each $\lambda$ (appendix section D.4 has newly added details). However, the GoFAE can be modified to allow the regularization coefficient to change during training. This will affect how convergence is proved and is beyond the scope of the current manuscript.
>
> > ...Is the proposed method to be used to modify the regularisation coefficient during the training?
>
> The newly added text in section D.4 clarifies these points.
>
> **Clarity, Quality, Novelty, Reproducibility**:
>
> > The use of p-values... p-value asymptotics...
>
> Please refer to the overall comment *Effect of minibatch on uniformity*.
>
> > GoF tests do not only concern posterior and prior distributions
>
> The sentence has been revised.
>
> > The aggregate posterior $\mathbb{P}_Y$ is a confusing word...
>
> We have revised the presentation of both the encoder $F$ and decoder $G$ in Section 2, and also revised the notation to better indicate what distribution(s) we are referring to (please see our top-level response in *Readability, notation, terminology, and acronyms*).
>
> > Line 8 in Section 3.1. Why do we need the expectation?
>
> and
>
> > The symbol for the test statistic $T$ is overloaded, it’s unclear what $T$ takes as input.
>
> We have made changes to make this clearer; please see the first paragraph of Section 3. $T$ refers to a test statistic; it is a function that takes an input. When this input consists of random variables (upper-case letters) then we are referring to $T$ as random variable. When the input consists of lower-case letters (realizations or observations) then we are referring to an *observed* test statistic, which we now denote $T^{\star}$ (lower-case $t$ is already used). We are following the standard statistical nomenclature.
>
> > Line 3 in Algorithm 3: Shouldn’t G be F?
>
> Thank you for catching this, it has been fixed.
>
> > Section 4 starts with training specifics using a test statistic $T$ without mentioning a training objective..
>
> We have revised the text based on this suggestion.
>
> > HT-Trainable. Almost everywhere differentiability seems to be too weak for practical applications.
>
> It is non-differentiable on a set of Lebesgue measure 0. In the case that a GoF test is optimal at one of these points, using stochastic gradient descent dictates the chance of landing at one of these non-differentiable points is zero.
>
> > Section 4.2. The minibatch subscript $\mathbf{i}$ is bold.
>
> Thank you, it has been fixed.

---

> > ### Comment · Reviewer_ySEm · 2022-12-02
> > **Acknowledgement of author response**
> >
> > Many thanks to the authors for addressing my comments.
> > The paper's readability has much improved (this was a major concern), which helped my understanding of the paper.
> > I cannot strongly recommend to accept this paper, and my score is set as such.
> > There are some elements introduced in the paper:
> >
> > 1. the use of a family of Gaussians (i.e., the optimization is such that the aggregated posterior is required to be some Gaussian instead of a single Gaussian) for training generative autoencoders,
> > 2. the use of a multivariate normality test (univariate normality tests + ball projection) for training to achieve (a), and
> > 3. the restriction of the last layer of the encoder to the Stiefel manifold.
> >
> > I appreciate that the higher-order criticism provides a way to choose the regularisation coefficient $\lambda$, which could be useful.
> > However, I could not fully comprehend the effect of the above components from the experiments. It seems that this model selection contribution is independent of the above components. I wondered, for example, what happens in practice if we apply the first element above to other autoencoders that do not use approahces 2 and 3 (or what happens
> > The empirical evaluation of the paper appears to have room for improvement.
> >
> > Additional comments:
> > * There are still some undefined symbols, e.g., $\hat{\mathbf{\mu}}$, $\hat{\mathbf{\Sigma}}$.
> > * MI between what?
> > * What distributions (or samples) are used to compute the FID scores?
> > * "This is theoretically supported by Kantorovich duality (Villani, 2008; Patrini et al., 2018)". This requires some assumption on $\mathbb{P}_X$ like not being atomic. Also Monge-Kantrovich equivalence might be a better name.
> > * "If $Y$ has a non-norrmal distribution,...". Lebesque -> Lebesgue. Also the set $\mathcal{S}$ has zero measure with respect to $d$-dimensional Lebesgue measure by definition. Shouldn't this require some condition the distribution on $Y$?
> > * GoF-Trainable's almost everywhere differentiability. Maybe saying something like "for $\otimes^m \mathbf{P}_X$-almost every $\mathbf{X}$..." might clarify what "almost every" means ($\otimes$ denotes the product measure).
> >
> > It appears that the authors justify this using some form of bootstrap (minibatch sampling). This might need some clarification.
> >
> > >  Indeed, some tests only produce an asymptotic p-value. For a large enough sample size, this discrepancy is negligible. In other cases, the test statistic distribution function cannot be obtained analytically, and so p-values are typically estimated empirically by simulation [1]. Since our methods use projection, we compute an empirical null distribution using Monte Carlo simulation; note that this process need only be performed a single time before training for a specific batch size and latent dimension size.

---

> > > ### Author Response · Authors · 2022-12-08
> > > **Further clarifications (part 1)**
> > >
> > > We thank the reviewer for taking another look at the paper, it is greatly appreciated.
> > >
> > > The mentioned typographic errors have been corrected. We have also modified the paper to include further clarifications for the points discussed below.
> > >
> > > > I appreciate that the higher-order criticism...which could be useful.
> > >
> > > Higher criticism is an integral part of the GoFAE. Without higher criticism we do not know if the aggregated posterior is close or far from the chosen prior.
> > >
> > > > However, I could not fully comprehend the components from the experiments: … the use of a family of Gaussians … the use of multivariate normality test …. the Stiefel manifold.
> > >
> > > These components are all bundled together and required to define the architecture. We investigated the effect of $\lambda$ being too small and too large (Figure 5a). When too small, the p-values from the GoF test are skewed right (blue histogram), and when $\lambda$ is too large, the p-values from the GoF test are skewed left (orange histogram). Both of these scenarios indicate that the aggregate posterior does not match the prior. Figures 5b, 5c specifically show the effect of $\lambda$ when using two different GoF tests, Shapiro-Wilk and Cramer von Mises for a range of $\lambda$. Similar plots for other tests can be seen in the Appendix (Figures S1, S2a, S3a).
> > >
> > > We have experimented with the individual components. The baseline AE (no GoF, no Stiefel) is included in Table 1, the GoFAE (GoF and Stiefel) is in Table 1, Appendix S1, and Appendix S6, and the ablation study in Appendix E.5 is GoFAE (GoF without Stiefel). We did not experiment with AE (no GoF but using Stiefel) as this isn’t related to our approach of utilizing GoF testing for evaluating prior matching. The Stiefel manifold is required for our convergence proof.
> > >
> > > > It seems that this model selection contribution is independent of the above components.
> > >
> > > The model selection component is independent in the sense that searching for an appropriate $\lambda$ is not directly part of the (gradient descent) optimization. Nevertheless, it is necessary; please see above point, as well as Appendix D.4.
> > >
> > > > I wondered, for example, what happens in practice if we apply the first element above to other autoencoders that do not use approaches 2 and 3...
> > >
> > > In principle, other divergences exist which are capable of computing proximity between a family of Gaussians and a specific distribution. However, the challenge is rooted in several issues we discussed in the introduction Section 1 paragraph 2. To our knowledge using GoF tests is the only way to test whether a distribution belongs to the class of Gaussians. To be fair, the distribution proximity is framed as a statistical question, so it is natural to take our approach.
> > >
> > > For example, take two common GAE frameworks: VAE and WAE-GAN (using an adversary). The VAE typically minimizes the per-sample discrepancy, $KL(\mathbb{P}\_{Y|X} || \mathbb{P}\_Z)$. Very often the variational approximate posterior is a multivariate (isotropic) Gaussian. If $\mathbb{P}\_Z$ were now replaced with the class of Gaussians, $\mathcal{G}$,  $KL( \mathbb{P}\_{Y|X} || \mathcal{G})$, then this doesn’t make much sense as $\mathbb{P}\_{Y|X}$ is already Gaussian. Please see our first response to 5naA.
> > >
> > > In the case of WAE-GAN, a discriminator is fed samples from the "true" distribution (prior) and samples from the "fake" (encoded) distribution $F\_{\\#}\mathbb{P}\_X$. Instead of per-sample discrepancy, the WAE approach seeks to match the aggregated posterior to the prior, often times a multivariate (isotropic) Gaussian. If the prior distribution is now replaced with a class of Gaussians, $\mathcal{G}$, then it is not clear what should be sampled from the prior (which is a class of distributions) and given to the discriminator.
> > >
> > > The VAE and WAE-GAN approaches are incompatible with using a Gaussian class as a prior. We included Figure 1 to help visualize the difference between the VAE, WAE, and our approach, the GoFAE. Nevertheless, it is an interesting question and one we have thought about.
> > >
> > > With regards to the Stiefel manifold, since distribution proximity testing is naturally addressed by using GoF tests for multivariate normality, then the Stiefel manifold is necessary for proving convergence (traditional SGD results are not applicable). Without the Stiefel manifold, it is not clear how to address the optimization problems discussed in Section 4: Identifiability, Singularities and Stiefel Manifold.
> > >
> > > >  There are still some undefined symbols $(\hat{\mu}, \hat{\Sigma})$.
> > >
> > > In order to generate new data, we require estimates of $\mu$ and $\Sigma$; these estimates are $(\hat{\mu}, \hat{\Sigma})$. We follow the statistical convention of putting hat above estimated quantities, but we have added additional clarification in the latest version.

---

> > > > ### Author Response · Authors · 2022-12-08
> > > > **Further clarifications (part 2)**
> > > >
> > > > > MI between what?
> > > >
> > > > We are computing the mutual information between $Z \sim \mathbb{P}\_Z$, and $\tilde{Z} \sim F\_{\\#}G\_{\\#}\mathbb{P}\_Z$. An estimate of the mutual information between $Z$ and $\tilde{Z}$, $I(Z, \tilde{Z})$, is computed between the set $\\{z_i\\}\_{i=1}^N$ and $\\{\tilde{z}_i\\}\_{i=1}^N$, and is discussed in the final paragraph of Section 5: Effect of $\lambda$ and Mutual Information (MI). We did include a discussion in Appendix D.3, but have clarified how mutual information is computed in the latest version.
> > > >
> > > > > What samples are used to compute FID scores?
> > > >
> > > > We used 10,000 samples reconstructed from the test set for computing the reconstruction FID score (Table 1 “Recon” under FID Score). We also generated 10,000 samples, by sampling from $\mathcal{N}(\hat{\mu}, \hat{\Sigma})$ and passing these through the decoder. These generated images are what the FID score is computed on.
> > > >
> > > > > This requires some assumption on $\mathbb{P}_X$ like not being atomic. Also Monge-Kantorovich equivalence might be a better name.
> > > >
> > > > We have made this assumption explicit in the latest version of the manuscript and have changed Kantorovich duality to Monge-Kantorovich equivalence.
> > > >
> > > > > Shouldn’t this require some condition on the distribution of $Y$?
> > > >
> > > > No assumption is needed. This general result is the main conclusion of the paper by Shao and Ming [1], which we have cited within our manuscript.
> > > >
> > > > > GoF-Trainable's almost everywhere differentiability. Maybe something like...
> > > >
> > > > This is a good point. To help with readability, we added the following sentence above the two GoF conditions. “We suppose that with probability one under $\mathbb{P}_X$ the following two conditions are satisfied.”
> > > >
> > > > > It appears that the authors justify this using some form of bootstrap (minibatch sampling). This might need some clarification.
> > > >
> > > > The previous response is referencing how the p-values for the minibatch GoF test are computed. There is no bootstrapping occurring; it’s simulation. This simulation is what produces the empirical null distribution of the test statistic. This is discussed in Appendix Sections D.1 and D.2.
> > > >
> > > > [1] Shao, Yongzhao, and Ming Zhou. "A characterization of multivariate normality through univariate projections." Journal of multivariate analysis 101.10 (2010): 2637-2640.

---

> > > > ### Comment · Area_Chair_ETAq · 2022-12-12
> > > > **AC to reviewer ySEm and authors**
> > > >
> > > > > There are still some undefined symbols $(\hat{\mu}, \hat{\Sigma})$.
> > > >
> > > > @Authors, I believe, as part of the question, the reviewer ySEm is also asking for the definitions of $\mu$ and $\Sigma$ (i.e., not just what hat $\hat{\cdot}$ means). Expanding the answer here will help further clarify this point. Thank you.

---

> > > > > ### Author Response · Authors · 2022-12-13
> > > > > **Authors to AC and reviewer ySEm**
> > > > >
> > > > > We have clarified the sentence using $(\hat{\boldsymbol{\mu}}, \hat{\boldsymbol{\Sigma}})$ in the most recent version of the manuscript. We have also specified the algorithm for generating these samples in pseudocode in section E of the Appendix.
> > > > >
> > > > > Old version:
> > > > >
> > > > > > Finally, we used the estimated $(\hat{\boldsymbol{\mu}}, \hat{\boldsymbol{\Sigma}})$ for each model to draw $N=10^4$ samples $\\{\mathbf{z}\_i\\}\_{i=1}^N$ and generate images $\\{G\_{\boldsymbol{\phi}}(\mathbf{z}\_i) \\}\_{i=1}^N$.
> > > > >
> > > > > New version:
> > > > >
> > > > > > Our method uses the class of Gaussians, $\mathcal{G} = \\{ \mathcal{N}(\boldsymbol{\mu}, \boldsymbol{\Sigma}): \boldsymbol{\mu} \in \mathbb{R}^d, \boldsymbol{\Sigma} \in \mathbb{R}^{d \times d} \\}$ as a prior, where $(\boldsymbol{\mu}, \boldsymbol{\Sigma})$ are the parameters of a Gaussian distribution denoting the mean and covariance matrix, respectively. When a model is finished training, we assume $F\_{\\#}\mathbb{P}\_X \in \mathcal{G}$ and use estimates of the mean and covariance $(\hat{\boldsymbol{\mu}}, \hat{\boldsymbol{\Sigma}})$ for each GoFAE model to generate samples. Specifically, we drew $N=10^4$ samples $\\{\mathbf{z}\_i\\}\_{i=1}^N$ to generate images $\\{G\_{\boldsymbol{\phi}}(\mathbf{z}_i) \\}\_{i=1}^N$.

---

### Author Response · Authors · 2022-11-19
**Top-level response**

Dear Reviewers,

We appreciate the critical suggestions from the reviewers, which have already improved our manuscript. We are excited to see encouragement for our work, including: finding the problem interesting (ySEm, Fj2Y, i5br, 5naA), recognizing its importance (i5br), the novelty of the approach (ySEm, Fj2Y, i5br, 5naA), generality of the method (i5br, 5naA), the theoretical motivation, justification and technical soundness (Fj2Y, i5br, 5naA), and the writing quality (i5br, 5naA).

We have spent significant efforts in addressing the comments.  Please find our responses to several shared concerns below. Responses for individual reviewer’s comments are made within their respective sections.

**Readability, notation, terminology, and acronyms** (ySEm, Fj2Y)

Based on reviewer feedback, we adjusted terminology and rewrote parts of sections 1 and 2.

* First, in our submission, we use upper-case letters for random variables and lower-case for realizations. Any deviations from this are explicitly stated. We added text to be explicit about this in the background within Section 2, and when defining GoF test statistics in Section 3.

* Second, we switched to pushforward notation to unambiguously denote the measurable spaces we are referring to in our definitions.

* Third, we improved general readability by removing the acronyms for hypothesis test (HT), optimal transport (OT), probability integral transform (PIT), and singular value decomposition (SVD). We redefined acronyms that are less commonly used in the literature (but commonly used in the manuscript) in each section where they are used (e.g., HC for higher criticism). We also removed abbreviations where possible.

**Clarity and purpose of Section 2: Propositions 1 and 2** (ySEm, Fj2Y, i5br)

We reorganized Section 2 into the theoretical preliminaries and motivation for the rest of the paper. Section 2 establishes the theoretical foundation of our GoFAE framework based on $L_2$-Wasserstein distance bounds using a specific prior distribution $\mathbb{P}_Z$. We extend this setup to the case where an entire distribution *class* is used instead of a single prior. To make the distinction clearer, we’ve included a new figure (Figure 1) that compares the latent space priors and encoding distributions of the VAE, WAE, and GoFAE, and adjusted Figure 2 by explicitly representing Gaussian distributions with different covariance matrix ranks within a Gaussian class prior $\mathcal{G}$. Compared with the VAE and WAE that use fixed priors, the GoFAE implicitly selects a $\mathbb{P}_Z$ from the entire class.

To compute the dissimilarity between a sample of data and a distribution class $\mathcal{G}$, we use GoF hypothesis test statistics. The GoF tests we have chosen to use are either based on Wasserstein distance or dominated by it. This section shows where and how GoF tests fit in, which leads to later topics like higher criticism and Riemannian optimization. We added some sentences at the end of Section 2 to more explicitly state how Propositions 1 and 2 can be extended to a distribution class.

**Effect of minibatch on uniformity**

(ySEm, i5br)

Indeed, some tests only produce an asymptotic p-value. For a large enough sample size, this discrepancy is negligible. In other cases, the test statistic distribution function cannot be obtained analytically, and so p-values are typically estimated empirically by simulation [1]. Since our methods use projection, we compute an empirical null distribution using Monte Carlo simulation; note that this process need only be performed a single time before training for a specific batch size and latent dimension size. This procedure avoids appealing to asymptotics by providing empirical p-values that approximate the exact p-values. There are some standard restrictions on minibatch size. If the minibatch size is too small, the usual problems associated with small minibatches arise, e.g. gradient estimation becomes noisy and can lead to slow convergence. If the minibatch size is too large, issues associated with testing on many samples arise. There is a tendency to reject $\mathcal{H}_0$ for a large sample size, as the large sample tends to magnify negligible differences [2]. Thus, the learned model may be oversensitive to insignificant deviations in the tails of the distribution.

[1] Mecklin, Christopher J., and Daniel J. Mundfrom. "A Monte Carlo comparison of the Type I and Type II error rates of tests of multivariate normality." Journal of Statistical Computation and Simulation 75.2 (2005): 93-107.

[2] Van der Vaart, Aad W. Asymptotic statistics. Vol. 3. 2000.

---

> ### Author Response · Authors · 2022-11-19
> **Top-level response part 2**
>
> **Improved or useful latent representation** (Fj2Y, i5br)
>
> Autoencoder-based representation learning is built around the manifold hypothesis: high-dimensional data often lies near a low-dimensional manifold [1,2]. The definition of an “improved” or “useful” latent representation depends on the context of the problem and specific task [3]. In this work, we evaluate *improved* or *useful* in a similar manner as previous work [4,5] based on image reconstruction and generation (MSE, FID, and image samples) across three experimental datasets of various complexity. Further, we include (1) explicit statistical hypothesis tests evaluating Gaussianity of the latent representation; (2) an analysis of the effect of $\lambda$ on the p-value distribution of batches, mutual information, and the condition number of its covariance matrix; and (3) evidence that the GoFAE can adapt as needed to a reduced-dimension representation through an analysis of the spectrum of singular values. All three of these analyses are crucial for downstream tasks that assume normality of the encoding and a well-specified latent dimension. Evaluating the level of representational improvement for other tasks, like clustering or supervised learning, is an interesting idea, but outside the scope of the present work.
>
> [1] Bengio, Yoshua, Aaron Courville, and Pascal Vincent. "Representation learning: A review and new perspectives." IEEE transactions on pattern analysis and machine intelligence 35.8 (2013): 1798-1828.
>
> [2] Rifai, Salah, et al. "The manifold tangent classifier." Advances in neural information processing systems 24 (2011).
>
> [3] Tschannen, Michael, Olivier Bachem, and Mario Lucic. "Recent advances in autoencoder-based representation learning." arXiv preprint arXiv:1812.05069 (2018).
>
> [4] Dai, Bin, and David Wipf. "Diagnosing and Enhancing VAE Models." International Conference on Learning Representations. 2018.
>
> [5] Tolstikhin, Ilya, et al. "Wasserstein Auto-Encoders." International Conference on Learning Representations. 2018.
>
>
>
>  **Empirical results and evaluation** (Fj2Y, i5br, 5naA)
>
> We would like to emphasize that our manuscript’s main focus is to understand the theoretical aspects of optimizing GoF statistics based on subsamples of the data and how this can be used to better address the reconstruction and posterior matching tradeoff. Furthermore, we aimed to test if higher-criticism can successfully use the information provided by (local) GoF hypothesis tests to appropriately select $\lambda$ and ensure the model is neither over nor under regularized. Beyond the theoretical results, we included a variety of qualitative and quantitative evaluations on three experimental datasets where reconstruction and generation are only a part of a more robust benchmarking and results exploration.
>
> Specifically, the reconstruction and p-value uniformity (coming from the test set) criteria provide evidence that the GoFAE model is *fit* correctly. FID scores suggest that the GoFAE model is *functioning* correctly by enabling the sampling from something close to $\mathbb{P}_X$. These criteria are also used when comparing generative autoencoders as they also seek to minimize reconstruction while matching a Gaussian prior. Mutual information provides an indication that the encoding process does not substantially remove information about the data space. Lastly, the condition number (in Figs. 5b-5c and appendix) provides an indication about the intrinsic dimension (see Section 3: The Benefits of GoF Testing in Autoencoders) of the data, something that other generative autoencoder methods either cannot do, or require two-stages. These multiple criteria are used to assess the GoFAE method holistically and are covered in Section 5 and the appendix. Our goal was never to only focus on generative quality; there are plenty of fantastic methods that already do this.
>
> Finally, we would like to highlight the fact that all models were trained with the same architecture for a fair comparison. We *only* use p-values from a training and validation set for the higher-criticism model selection step and can show strong performance across a variety of criteria.
>
>
>
> **Selection of Lambda, hyperparameter, KS variance** (ySEm, 5naA)
>
> See the response under 5naA (second weakness)
>
>
>
> **Clarity** (ySEm, Fj2Y)
>
> The reviewers pointed out several linguistic and typographic errors, all of which have been addressed.
>
>
>
>
> Best,
>
> “Auto-Encoding Goodness of Fit” Authors

---

### Decision · Program_Chairs · 2023-01-20

**Decision:**

Accept: poster

**Justification For Why Not Higher Score:**

While the paper's core proposal is theoretically sound, there were concerns on weak experiments (but still informative).

**Justification For Why Not Lower Score:**

The use of p-values of a goodness-of-fit test as a proxy to quantify how close the aggregated posterior is to the prior is unique, which may open up new directions. This and other contributions are worth showing to the community.

**Metareview: Summary, Strengths And Weaknesses:**

This submission considers the problem of learning a (probabilistic) auto-encoder. Main contributions of this submission are

1. Proposes using minibatch p-values coming from a goodness-of-fit test to assess whether the aggregated posterior belongs to a prior distribution class (class of Gaussian distributions). Uses uniformity testing.
2. Restricts parameters to the Stiefel manifold and appropriately proposes to use Riemannian optimization.

While there were concerns raised regarding missing crucial explanations, and writing being hard to follow (ySEm, Fj2Y, 5naA), these have since been resolved by appropriate rebuttals and revisions. Other concerns raised were on experiments (5naA, i5br) which appear to be weak in terms of comparison to existing baselines. All reviewers nonetheless agree that the proposal method is interesting, theoretically grounded and novel. The use of p-values of a goodness-of-fit test as a proxy to quantify how close the aggregated posterior is to the prior is unique, which may open up new directions.

Owing to these merits from the theoretical side, I recommend that the paper be accepted.

Suggestions to the authors to further improve the paper:
1. Consider further simplifying the presentation.
2. Consider further experiments beyond verifying FID scores. For instance, investigate "what percentage of dimension do GoFAE need to achieve the same FID as a VAE with D dimension" as suggested by reviewer Fj2Y.


**Note From Pc:**

if the above contains the word "oral" or "spotlight" please see: "oral" presentation means -> notable-top-5% and "spotlight" means -> notable-top-25%. As stated in our emails, we are disassociating presentation type from AC recommendations

**Summary Of Ac-Reviewer Meeting:**

N/A